# A frequency-amplitude coordinator and its optimal energy consumption for biological oscillators

Bo-Wei Qin [1,2✉], Lei Zhao [1,3] & Wei Lin [1,2,4,5✉]

Biorhythm including neuron firing and protein-mRNA interaction are fundamental activities with diffusive effect. Their well-balanced spatiotemporal dynamics are beneficial for healthy sustainability. Therefore, calibrating both anomalous frequency and amplitude of biorhythm prevents physiological dysfunctions or diseases. However, many works were devoted to modulate frequency exclusively whereas amplitude is usually ignored, although both quantities are equally significant for coordinating biological functions and outputs. Especially, a feasible method coordinating the two quantities concurrently and precisely is still lacking. Here, for the first time, we propose a universal approach to design a frequency-amplitude coordinator rigorously via dynamical systems tools. We consider both spatial and temporal information. With a single well-designed coordinator, they can be calibrated to desired levels simultaneously and precisely. The practical usefulness and efficacy of our method are demonstrated in representative neuronal and gene regulatory models. We further reveal its fundamental mechanism and optimal energy consumption providing inspiration for biorhythm regulation in future.

---

[1] School of Mathematical Sciences, Fudan University, 200433 Shanghai, China. [2] State Key Laboratory of Medical Neurobiology and MOE Frontiers Center for Brain Science, Institutes of Brain Science, Fudan University, 200032 Shanghai, China. [3] The GLOBE Institute, University of Copenhagen, Copenhagen, Denmark. [4] Shanghai Center for Mathematical Sciences, 200438 Shanghai, China. [5] Center for Computational Systems Biology of ISTBI, LCNBI, and Research Institute of Intelligent Complex Systems, Fudan University, 200433 Shanghai, China. ✉email: boweiqin@fudan.edu.cn; wlin@fudan.edu.cn

Plenty of living organisms as well as synthetic biological networks have their capacity to generate rhythmic processes involving molecules, cells, and tissues[1–5]. For instance, neuron spiking, cell reproduction, hormone secretion, protein synthesis, and heartbeat are common periodic activities in the human body. These processes are usually regulated by single or multiple biological oscillators exhibiting various frequencies and amplitudes (Fig. 1). The former ones would result in distinct functional consequences[6], while the latter ones abstract expression levels meeting body demands for protein, hormone, energy, to name a few[7]. More importantly, they control the identity and intensity of a signal being critical to internal information transduction[8]. Therefore, their coordinations are essential to physiological behaviors, such as sleeping, feeding, and mood[9–13].

Oscillators with anomalous frequency or amplitude may disrupt biorhythm and lead to sleep and metabolism disorders or even diseases[14–16]. Fairly recent perspectives also addressed that circadian rhythmicity in our body can be leveraged to develop chronotherapy and to account for the mechanism behind it[17,18]. Harmonizing administration of agent with associated biological target can improve its efficacy and reduce toxicity. Thus, flexible biological oscillators have a wide range of benefits for healthy sustainability. As a consequence, designing practical coordinators

to acquire desirable frequency or/and amplitude becomes a growing and significant issue.

The theoretical study on biorhythm dates back to 1960s when Winfree studied frequency exclusively in his pioneering work[19]. Since then, the adjustment of frequency via synchronization were thoroughly studied using reformulated Kuramoto model[20]. In contrast, there are few studies focusing particularly on amplitude. Over the past decades, frequency and amplitude modulations of biological oscillators has attracted more and more attentions, among which coordinating amplitude or frequency independently (Fig. 1a, b) is particularly significant.

For biological models, the dynamics depend highly on their network motifs (pattern of positive/negative interactions) and significantly attract plenty of attentions[21–23]. Therefore, investigating the mechanisms between the motif and frequency–amplitude coordination becomes a natural direction. Auto-regulations via endogenous interactions were ubiquitously studied[24–28]. Prior works also suggested that the tunability of frequency and amplitude can be enhanced by positive feedback loops, while the independent coordinations may not be achieved if only negative feedback loops appear[29–31]. Besides, a very recent work[32] studied the dual-feedback oscillator and the repressilator as well as the modifications of their network architectures. The frequency and amplitude of a gene

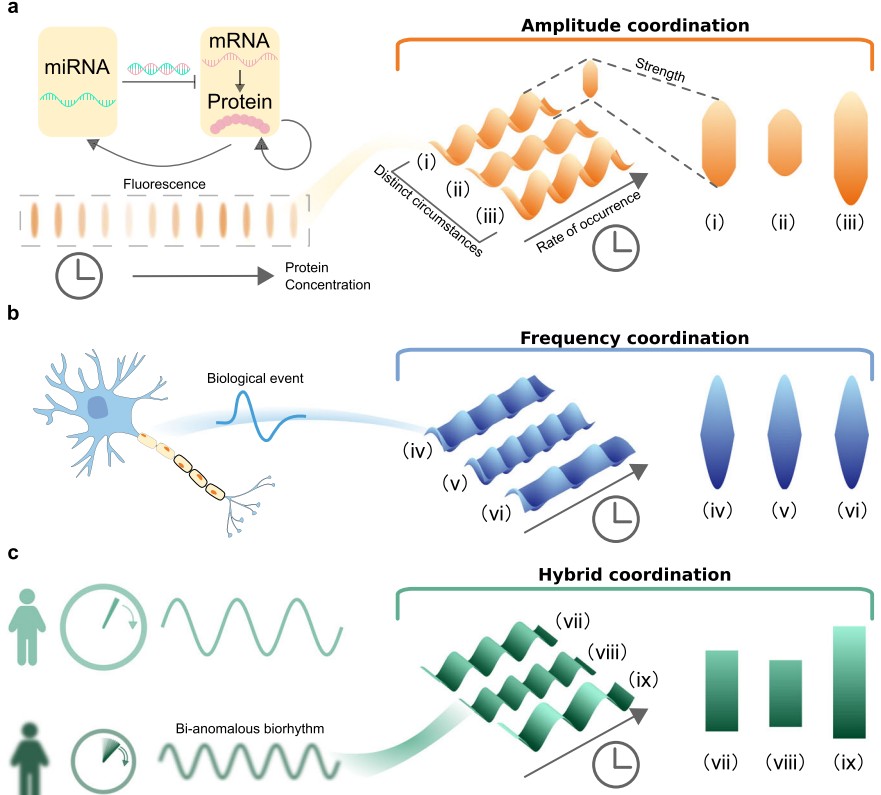

**Fig. 1 Independent and hybrid coordinations in representative biological models. a** Amplitude coordination. The rhythmic biological processes exist in genome-scale transcriptional and translational processes such as protein–microRNA (miRNA) circuitry. The protein concentration in a specific gene-regulatory network changes throughout the day. The variations under distinct circumstances are sketched by the heterogeneous wave-like pattern (i)–(iii) on the right. The projections of the waves on the rightmost represent their amplitudes. The amplitude is varied under (independent) amplitude coordinations. Meanwhile, the frequencies of these processes are not changed. **b** Frequency coordination. Periodic biological event is generated due to neuronal excitability exhibiting diverse frequencies under distinct circumstances (right). Under (independent) frequency coordinations, the rates of occurrence can be modulated in both ways (acceleration or deceleration) while the amplitude (intensity of the event) remains invariant. **c** Hybrid coordination. A biorhythm disorder may include both frequency and amplitude disruption, which further influences internal signal transduction. With hybrid coordination, the two quantities are modulated simultaneously, and they are interchangeable with one another. Compared with pattern (vii), (viii) has a higher frequency and a lower amplitude. An opposite situation is illustrated as pattern (ix). Biological events in the space may occur under different boundary conditions. The amplitude projections shown in the rightmost of (**a**–**c**) are typical diagrams for the Robin, Dirichlet, and Neumann boundary condition, respectively.

outside the main oscillator are "physically" decoupled by re-designing the oscillator, thus they can be modulated independently in extended ranges.

Actually, the regulations of frequency and amplitude of the components inside an oscillator are also significant. Moreover, besides the network topology, the intensities of endogenous interactions affect frequency and amplitude decisively, but there are few related works focusing on this direction[33]. Frequency and amplitude can be coordinated by making a slight intervention on these interactions. This can be realized either internally by varying the system parameters (e.g., degradation rate, synthesis rate, strength of a stimulus, etc.) or externally by a feedback controller (e.g., computer-based microfluidic devices[34,35]). More importantly, the underlying mechanisms for coordinations as well as their energy consumptions also depend highly on these interactions, which have not yet been uncovered. In this article, we present a universal computational framework for precisely designing such an intervention coordinator. We will try to answer the following questions: What specific form of an intervention should be made (i.e., enhancing or mitigating the original inter-actions)? How is its intensity and energy consumption? Additionally, with our well-designed coordinator, both frequency and amplitude can be regulated precisely and concurrently (Fig. 1c).

To get a deeper insight into the biological oscillators, dynamical systems techniques based on computational and mathematical modeling are usually applied[21,36–41]. For systems in small scales, spatial movements including molecular and ion diffusion cannot be neglected when modeling because they would cause heterogeneity patterns (Fig. 1a, b). By dividing space into several discrete parts, we can use agent- or compartment-based models (i.e., ordinary differential equations) to simulate such a system[42]. However, these models still have their limitations. We may oversimplify a system if the spatial division is coarse, and the large number of components would increase computational consumption hugely. Contrarily, partial differential equation (PDE) is a more appropriate model to describe systems with variations in both continuous space and time. Especially, reaction–diffusion (R–D) systems are commonly used to investigate dynamics at the molecular, cellular, and tissue levels[43–47]. It is also well known that Alan Turing used both discrete and continuous models in his seminal work on morphogenesis[48]. Moreover, R–D systems can describe simple but significant random walk processes[49]. The Hopf bifurcation plays a pivotal role in appearance of a rhythmic oscillation. It refers to a transition from a quiescent state to an oscillatory state[50–53]. In R–D systems, it yields spatial, periodic oscillations[54,55]. Owing to diffusive effects and diverse boundary conditions, such rhythmic oscillations may be spatially non-homogeneous (Fig. 1a, b), for which frequency and amplitude coordinations still remain to be explored. It is then reasonable and significant to start our study with the Hopf bifurcation in R–D systems.

In this work, we computationally design a universal coordinator intervening linear interactions for modulating both frequency and amplitude. It is feasible for oscillations arising from the Hopf bifurcation in generic two-component R–D systems with the conventional boundary conditions [e.g., the Neumann boundary condition (NBC), the Dirichlet boundary condition (DBC), and the Robin boundary condition (RBC), see Supplementary Note 1 for details]. The basic idea is to extract and normalize the frequency and amplitude by utilizing the center manifold and normal form theories[54,56,57] and to link them with the intensities of linear interactions. In such a way, we are able to design a useful coordinator leveraging its linear regime [e.g., the Michaelis–Menten (MM) function]. In two representative biological models (Fig. 1a, b), a gene-regulatory network and a neuronal model, we demonstrate the efficacy and practical usefulness

of our framework. In light of the estimations computed via our approach, oscillations far from the quiescent state can be modulated as well. The underlying mechanisms and energy consumptions are also discussed, which might be helpful for biological regulation in future.

## Results

**Coordinating a "cancer network" by MM regulations.** As a first example, we consider the cyclic dynamics in a gene-regulatory network involving a microRNA (miRNA) cluster and a protein module[58]. It is called a "cancer network" because the miRNA behaves as an oncogene or tumor suppressor depending on the protein concentration. Its mathematical model abstracts the interaction among the transcription factors and a miRNA cluster[58] whose dimensionless diffusive model is given as (see Supplementary Note 8 for more details on the model)

$$\begin{aligned}
\frac{\partial \phi}{\partial t} &= d_{\mathrm{p}} \frac{\partial^2 \phi}{\partial x^2} + \frac{1}{\epsilon} \left[ \alpha' + \left( \frac{\kappa \phi^2}{\Gamma_1' + \phi^2 + \Gamma_2' \mu} \right) - \phi \right], \\
\frac{\partial \mu}{\partial t} &= d_{\mathrm{m}} \frac{\partial^2 \mu}{\partial x^2} + 1 + \phi - \mu,
\end{aligned} \quad (1)$$

where $\phi = \phi(x, t)$ and $\mu = \mu(x, t)$ represent the dimensionless level of protein and miRNA, respectively. Here we include two diffusive terms (second-order differentiation with respect to the spatial variable $x$) into the equations to describe the unbiased molecular diffusion. With appropriate parameters (Supplementary Table 3), the model possesses a constant quiescent state $(\phi_0, \mu_0)$ that undergoes the Hopf bifurcation at $\epsilon = \epsilon^*$ yielding rhythmic oscillation when $\epsilon < \epsilon^*$ (Fig. 2).

To achieve frequency and amplitude coordinations, we add two terms $F_1(\phi, \mu)$ and $F_2(\phi, \mu)$ into the first and second equation to regulate the dynamics of protein and miRNA, respectively. They act as instantaneous regulations on $\phi$ and $\mu$ as follows

$$\begin{aligned}
F_1(\phi, \mu) &= f_{11} M(\phi - \phi_0, K_{11}) + f_{12} M(\mu - \mu_0, K_{12}), \\
F_2(\phi, \mu) &= f_{21} M(\phi - \phi_0, K_{21}) + f_{22} M(\mu - \mu_0, K_{22}),
\end{aligned} \quad (2)$$

where $M(x, K) = x/(x + K)$ is a MM regulation and $f_{ij}$ indicate the intensities. The MM regulation was recently shown to be practical and useful in implementing the proportional control in biological molecules[59]. Here we take into account all possible regulations: two self-regulations ($\phi \to \phi$ and $\mu \to \mu$) and two cross-regulations ($\phi \to \mu$ and $\mu \to \phi$). We need to point out that the regulations in Eq. (2) incorporate the coordinate translations because the dynamics considered in our case oscillates around the quiescent state $(\phi_0, \mu_0)$ and we desire to leverage the information on their displacement from the quiescent state. Actually, such a translated regulation can be divided into two typical MM functions

$$M(x - x_0, K) = \frac{x - x_0}{x - x_0 + K} = \frac{x}{x + \bar{K}} - \frac{x_0}{\bar{K}} \cdot \frac{\bar{K}}{x + \bar{K}}, \quad (3)$$

with $\bar{K} = K - x_0$.

For the given Michaelis constants ($K_{ij} = 5$ in our case), we select appropriate regulation intensities $f_{ij}$ to coordinate independently the frequency or amplitude of the oscillation near the Hopf bifurcation ($\epsilon = 0.08 < \epsilon^*$). By varying the intensities at distinct time, we eventually accelerate the oscillation with a doubled frequency. Concurrently, both amplitudes of protein and miRNA concentrations are kept near constants (Fig. 2a–d). Moreover, with other well-designed intensities, the independent amplitude coordination is also successfully performed (Supplementary Fig. 3 and Movie 5).

Depending on the levels of protein module $\phi$, the miRNA cluster is classified as oncogenes or tumor suppressors[58]. When $\phi$ approximately lies between 2.75 and 3.75, the region is labeled as a cancer zone where hyper-proliferation occurs. Accordingly, the

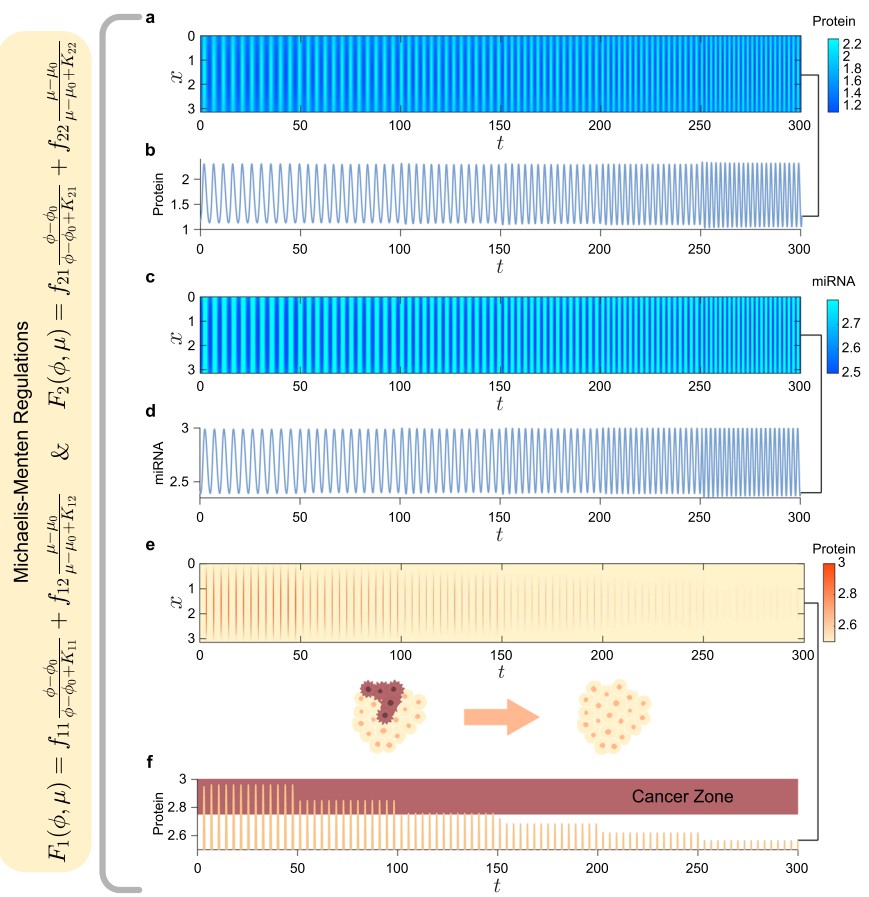

**Fig. 2 Independent frequency and amplitude coordinations in the "cancer network" using Michaelis–Menten regulations. a** The time course of protein concentration in the "cancer network" [Eq. (1)] when $\epsilon = 0.08$. When $t < 50$, no coordinator is applied to the system. Starting from $t = 50$, Eq. (2) with $K_{ij} = 5$ is applied as a coordinator. At time labels $t = 50, 100, 150, 200$, and $250$, the intensities $f_{ij}$ are varied. **b** The time course of protein concentration at $x^\star = \pi/2$. **c** The time course of miRNA level corresponding to **a. d** The time course of miRNA level at $x^\star = \pi/2$. **e** The time course of protein concentration in the "cancer network" when $\epsilon = 0.05$. The amplitude is gradually and independently suppressed (the color becomes lighter) by varying the coordinator at distinct time labels. The brown area is the cancer zone with high probability of oncogenesis classified by the level of protein (>2.75). The amplitude suppression prevents the oscillation from the entry into the cancer zone. For the corresponding time course of the miRNA cluster, see Supplementary Fig. 2. **f** The time course at $x^\star = \pi/2$. (For the parameters and intensities of the coordinator used in **a–f** see, respectively, Supplementary Table 3 and 4).

probability of oncogenesis is increased. We find that the oscillation stays away from the cancer zone (i.e., $\phi < 2.75$) if it is close to the quiescent state (Fig. 2a, b). But, when it keeps growing, its amplitude increases and the peak enters the cancer zone. For instance, the peak of the oscillation (when $\epsilon = 0.05$ without a coordinator) shown in Fig. 2e, f ($t < 50$) lies in the cancer zone. To prevent the oscillation from the entry of the cancer zone, it is possible to use our coordinator with appropriate intensities $f_{ij}$ to suppress its amplitude. For this purpose, we design appropriate coordinators and apply them at different time to decrease the amplitude gradually ($50 < t < 300$ in Fig. 2e, f). Eventually, the protein concentration stays away from the cancer zone. Thus, the miRNA cluster is no longer classified as oncogenes. Evidently, our designed coordinator is feasible for coordinating both frequency and amplitude. Let us now introduce its theoretical background and a universal computational framework for determining the intensities.

**Endogenous linear interactions and their interventions**. To investigate the dynamics near the Hopf bifurcation of every biological diffusive model akin to Eq. (1), we can always translate the quiescent state to the origin and write the equations into a generic form (see "Methods" for a detailed description):

$$\underbrace{\frac{\partial \mathbf{u}(x,t)}{\partial t}}_{\text{Time evolution}} = \underbrace{\mathbf{D}(\epsilon)\frac{\partial^2 \mathbf{u}(x,t)}{\partial x^2}}_{\text{Spatial diffusion}} + \underbrace{\mathbf{A}(\epsilon)\mathbf{u}(x,t)}_{\text{Linear interaction}} + \underbrace{\mathbf{g}(\mathbf{u}(x,t),\epsilon)}_{\text{Nonlinear interaction}}. \qquad (4)$$

Such a system possesses two self-interactions and two cross-interactions, all of which can be divided into two parts: the linear interactions $\mathbf{Au}$ and the nonlinear ones $\mathbf{g}$. The matrix $\mathbf{A}$ is associated with the Jacobian matrix and can be acquired in any computational model[21]. It represents the endogenous linear interactions, which consists of four coefficients $a_{ij}$ $(i, j = 1, 2)$ representing associated interaction ($u_j$ to $u_i$). The sign of $a_{ij}$ indicates the role played by the component $j$ (activator or inhibitor), see below. Their magnitudes represent the intensities that are determined by system parameters, such as degradation rate, synthesis rate, Michaelis constant, etc. These parameters are usually different from system to system (see Supplementary Note 9 for more details on the linear interactions).

For a given biological system, the endogenous linear interactions always exist in its computational model and have dominant

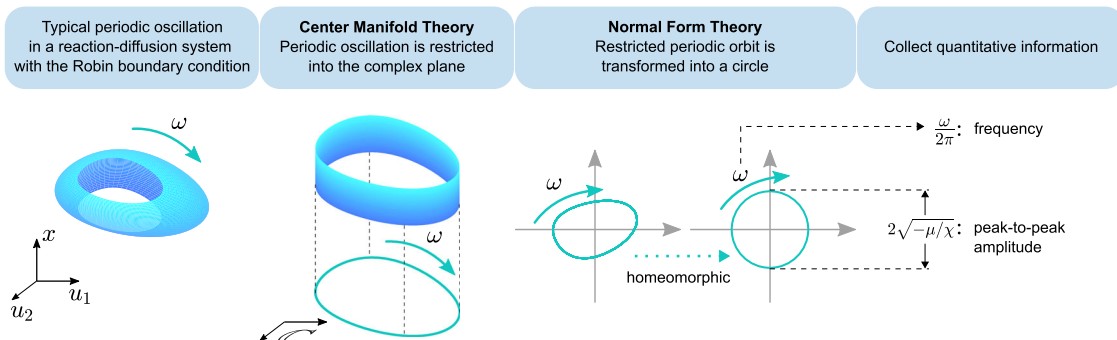

**Fig. 3 The process for quantifying frequency and amplitude of a periodic oscillation.** A typical periodic oscillation with two components ($u_1$ and $u_2$) in a R–D system with non-uniform distribution is shown in the leftmost panel. The diffusive (heterogeneity) effect along the spatial variable $x$ can be clearly seen. By applying the center manifold theory, such an effect is eliminated (see Supplementary Note 5 for details). Then the restricted periodic oscillation is spatially homogeneous (invariant along $x$-axis) and finally can be projected into a complex plane which is of dimension two, as shown in the second panel. By applying the normal form theory, the periodic orbit is converted to a circle (the third panel), whose frequency [$\omega/(2\pi)$] and amplitude ($2\sqrt{-\mu/\chi}$) are finally collected (the last panel).

impacts on the behavior of the oscillation near the quiescent state. Specifically, the intensities of these interactions determine the frequency and amplitude decisively. When translating the coordinated system, our proposed MM regulations [Eq. (2)] are also shifted to a regular MM function. For instance, $M(\phi - \phi_0, K_{11})$ is translated to $M(\Phi, K_{11})$ with $\Phi = \phi - \phi_0$ representing the displacement of the oscillation from the quiescent state. Near the Hopf bifurcation, the quantity $\Phi$ is relatively small. The MM function is therefore in its linear regime, that is,

$$M(\Phi, K) = \frac{\Phi}{\Phi + K} \approx \frac{1}{K}\Phi \quad \text{when} \quad |\Phi| \ll 1. \tag{5}$$

Consequently, near the Hopf bifurcation, the introduced MM regulation is analogous to a linear proportional control making interventions on the original linear interactions, thereby our proposed coordinator is feasible for coordinating the frequency and amplitude.

**A universal design policy for the coordinator.** To design a feasible coordinator, we need to know how the intensities of the MM regulations relate to the frequency and amplitude. Here we apply the center manifold and the normal form theories to accomplish the task (Fig. 3). In the following computational framework, we consider the linear regime of the MM function where the feedback coordinator for Eq. (4) is written as

$$\mathbf{Fu} = \begin{bmatrix} f_{11} & f_{12} \\ f_{21} & f_{22} \end{bmatrix} = \begin{bmatrix} u_1 \\ u_2 \end{bmatrix}. \tag{6}$$

Note that the intensities $f_{ij}$ in Eqs. (6) and (2) differ by a given Michaelis constant $K_{ij}$. Considering solely the linear regime is sufficient to provide accurate estimation of the intensities in Eq. (2). Comparing with the nonlinear MM functions, the linear terms significantly simplify the subsequent calculations. Another advantage of investigating the linear terms is that it is applicable for other cases where the MM regulations may not be used. For instance, when coordinating a neuronal model, it is possible to integrate the linear terms as additional stimulus current or power supply. The result for Eq. (6) is also suitable for other nonlinear functions possessing a linear regime, such as the sinusoidal signal/current

$$\sin(au_1 + bu_2) \approx au_1 + bu_2 \quad \text{when} \quad |u_1| \text{ and } |u_2| \ll 1, \tag{7}$$

and other generalized MM functions

$$\frac{u_1}{u_1 + K_1 u_2 + K_2} \approx \frac{1}{K_2} u_1 \quad \text{when} \quad |u_1| \text{ and } |u_2| \ll 1,$$
$$\frac{u_1}{u_1 + K_1 u_1^2 + K_2} \approx \frac{1}{K_2} u_1 \quad \text{when} \quad |u_1| \ll 1. \tag{8}$$

We now present the computational framework that includes the following steps.

We first perform a persistence analysis (see "Methods") before moving into detailed computations. It guarantees the existence of the oscillation after intervention. This gives us Theorem 1, which provides the first criteria ($f_{22} = -f_{11}$) for designing the coordinator. It indicates that the two self-interactions of a feasible coordinator must be opposite to one another. Thus, a negative feedback loop is established providing the potential to produce rhythmic oscillation in a biological system[29], reasoning that it ensures the occurrence of the Hopf bifurcation leading to a cyclic oscillation. In practical applications, a stable oscillation is of great significance and interest. Therefore, we also perform a stability analysis to guarantee that the oscillation is still stable after implementing the coordinator (see "Methods").

The second step is to link the intensities $f_{ij}$ with the frequency and amplitude. Therefore, we exploit the normal form of the Hopf bifurcation[57] utilizing the center manifold and the normal form theories. For a periodic oscillation, the Poincaré normal form is always found as (see "Methods" for more details)

$$\dot{w} = \lambda w + \eta w^2 \bar{w} + \mathcal{O}(|w|^4), \quad w \in \mathbb{C}, \tag{9}$$

where $\lambda = \mu + i\omega$ is the complex eigenvalue corresponding to the Hopf bifurcation and $\eta$ is the normal form coefficient whose real part, denoted by $\chi$, is the first Lyapunov coefficient. It is the simplest form preserving the information and is usually applied to study qualitative behaviors. Here we extract the quantitative information (frequency and amplitude) from the normal form. It serves as a bridge from the original oscillation to the coordinated one. Therefore, those information are helpful for designing the coordinator. The quantifying procedure is illustrated in Fig. 3.

The third step is to modulate both components ($u_1$ and $u_2$) consistently. We notice that they always share the same frequency, whereas their amplitudes could be distinct. As a consequence, the variations made by the coordinator may be inconsistent, in the sense that the amplitude of $u_1$ (e.g., protein concentration) is increased or decreased while that of $u_2$ (e.g., miRNA concentration) is invariant or vice versa. Such a result may be unexpected for a given biological system. To avoid this

circumstance, we analyze both components when computing the normal form (see "Methods"). We surprisingly find that the modulations on both amplitudes are commensurate with one another if we have:

$$\frac{f_{21}}{f_{12}} = \frac{a_{21}}{a_{12}}. \tag{10}$$

This becomes the second criteria for the design policy. Reasonably, it incorporates the endogenous cross-interactions and their interventions. The ratio of two amplitudes is affected by the cross-intensities. To keep this ratio unvarying, the intensities of the interventions must also follow a fixed ratio determined by the intrinsic ones (i.e., $a_{21}/a_{12}$).

In the final step, we derive two algebraic equations for the coordinator. From Eq. (9), the frequency and amplitude are extracted and approximated as $\omega/(2\pi)$ and $2\sqrt{-\mu/\chi}$, respectively. Note that $\mu$ is independent of the intervention coefficients, whereas $\omega$ and $\chi$ are expressed in terms of $f_{11}$ and $f_{12}$, i.e., $\omega = \omega(f_{11}, f_{12})$ and $\chi = \chi(f_{11}, f_{12})$. Before going any further, it is worth pointing out that the approximated amplitude is independent of $x$ (i.e., the spatial variable). In spite of this, the coordinator designed here still works for the entire space, because the spatially heterogeneous effect is eliminated during the computation (see Supplementary Notes 5 and 6 for more details) and it does not affect the final coordination.

We then denote by $\omega_0$ and $\chi_0$ the two values for the original oscillation [i.e., $\omega(0, 0)$ and $\chi(0, 0)$], and $\omega_c$ and $\chi_c$ the coordinated ones. To accomplish the coordination at the desired frequency and amplitude, we only need to solve two algebraic equations: $\omega_c/\omega_0 = r_F$ and $\sqrt{\chi_0/\chi_c} = r_A$, where $r_F$ and $r_A$ indicate, respectively, the fold changes of the frequency and amplitude with regard to the original quantities. For instance, to coordinate frequency independently, we set $r_A = 1$ and an unrestricted $r_F$. Analogously, we coordinate amplitude independently by setting $r_F = 1$ and a free $r_A$. The frequency and amplitude are coordinated simultaneously if neither of $r_F$ and $r_A$ is one. Generically, the equations afford an accurate prediction for designing the coordinator (Supplementary Fig. 5). For higher accuracy, we numerically search more appropriate values close to the predicted ones.

Following the above procedure, we design feasible coordinators for the independent frequency and amplitude coordinations in the "cancer network" (see Supplementary Note 8 for computational details). As mentioned before, Eq. (6) provides accurate estimation of Eq. (2). To verify this fact, we also compare the efficacy of the linear coordinator and that of the MM regulations. Our proposed nonlinear coordinator is indeed well approximated by the linear coordinator even if the oscillation is relatively far from the quiescent state (Supplementary Fig. 4).

**Independent coordination in a neuronal model**. To further demonstrate the efficacy of our framework, we also apply it to the FitzHugh–Nagumo (F–N) system, a representative model simulating biological neuron[60–62] (see "Methods"). The parameters we used under distinct boundary conditions and other information are provided in Supplementary Table 5. Following the procedure introduced before, we design some feasible linear coordinators such that the frequency of a periodic oscillation is gradually increased or decreased under different boundary conditions (Fig. 4a, f, g and Supplementary Figs. 6 and 7). Simultaneously, the amplitudes of both components are almost kept as constants. For more examples of independent frequency coordination, see Supplementary Figs. 8–10 and Movies 1–3.

Given a fixed fold change $r_F$, some computed intervention intensities should be abandoned as their magnitudes are too large for practical applications. Besides, to guarantee that the

coordinated periodic oscillation is still stable, we require the coordinator to satisfy two stability conditions: the negativity of the first Lyapunov coefficient $\chi$ and the positivity of another measure index $c_0(k_i)$ (see Theorem 2 in "Methods"). Accordingly, a fixed $r_F$ yields two coordinators $(f_{11}, f_{12})$, where $f_{11}$ are identical and the two values of $f_{12}$ are symmetric to a constant.

We present in Fig. 4b–d the relations between the intervention intensities and the ratio of frequencies. The two coefficients show an almost linear relation. When $\omega_0/\omega_c < 1$ (i.e., $r_F > 1$), the large magnitudes of $f_{11}$ and $f_{12}$ imply that a more intense coordinator is needed for acquiring a higher frequency. As a comparison, it is much easier to decrease the frequency in the sense that the required interventions are relatively moderate.

Analogously, we also accomplish the independent amplitude coordinations with the proposed approach. Under different policies, the oscillation is gradually amplified (for both components) while the frequency remains unchanged (Fig. 5a, b). See Supplementary Movie 4 and Figs. 13, 14, and 16 for more instances. We find that the amplitude fold change $r_A$ has a lower bound (e.g., about 0.723 for the NBC case), below which the oscillation cannot be coordinated successfully (Fig. 5c, d). In other words, the periodic oscillation cannot be suppressed to an arbitrarily small size. This is caused by the instability of the oscillation. Although we find some coordinators for every $r_A < 1$, they may violate Theorem 2 (see "Methods"). Consequently, the oscillation becomes unstable. To analyze the lower bound, we investigate the minimum of the index $c_0(k_i)$, whose negativity implies instability (see Supplementary Fig. 15). In the unstable region (Fig. 5c, d), the oscillation cannot be observed since it would converge to a different state (possibly a steady state) or undergo a finite time blowup.

As claimed before, one feature of our method is the capability to modulate the amplitude in the entire spatial domain by almost identical fold change $r_A$. Specifically, we illustrate in Fig. 5e, f an adequate example. Apparently, for both components ($V$ and $W$), the amplitudes are increased for all spatial variable $x$ (the vertical axis) indicating that the spatially non-homogeneous property does not influence the effect of our coordinator.

**The optimal energy consumption of a coordinator**. In practice, the consumption of a coordinating policy for sustaining an oscillation is of great significance. Therefore, an index $\mathcal{E}$ is always introduced to measure the performance of a specific control policy[63], see "Methods" for the one used in our problem. Figure 4e shows the energy consumption of independent frequency coordination in the F–N system with NBC. It is clear that there is a big difference between the two policies. Usually, the blue one is the optimal one that we may use because its consumption is lower. It is worth pointing out that, throughout this work, we only illustrate the coordinations attained by the optimal coordinator. A clearer energy consumption of the optimal frequency coordinator for the F–N model is provided in Supplementary Fig. 11 from which we deduce that the coordinator consumes more energy for acquiring a decreased frequency (i.e., $\omega_0/\omega_c > 1$). In Fig. 5d, the energy consumption for the optimal independent amplitude coordination is also given. Apparently, the magnitudes of the index $\mathcal{E}$ are exceedingly different for the two cases in the sense that the amplitude coordination consumes more energy than the frequency coordination. Moreover, increasing and decreasing the amplitude yield analogous energy consumptions. In Fig. 6c, the optimal energy consumptions for the four cases are indicated according to their magnitudes.

**Coordinating the "bigger" oscillation**. Until now, our framework are verified on distinct oscillations in the two models, most

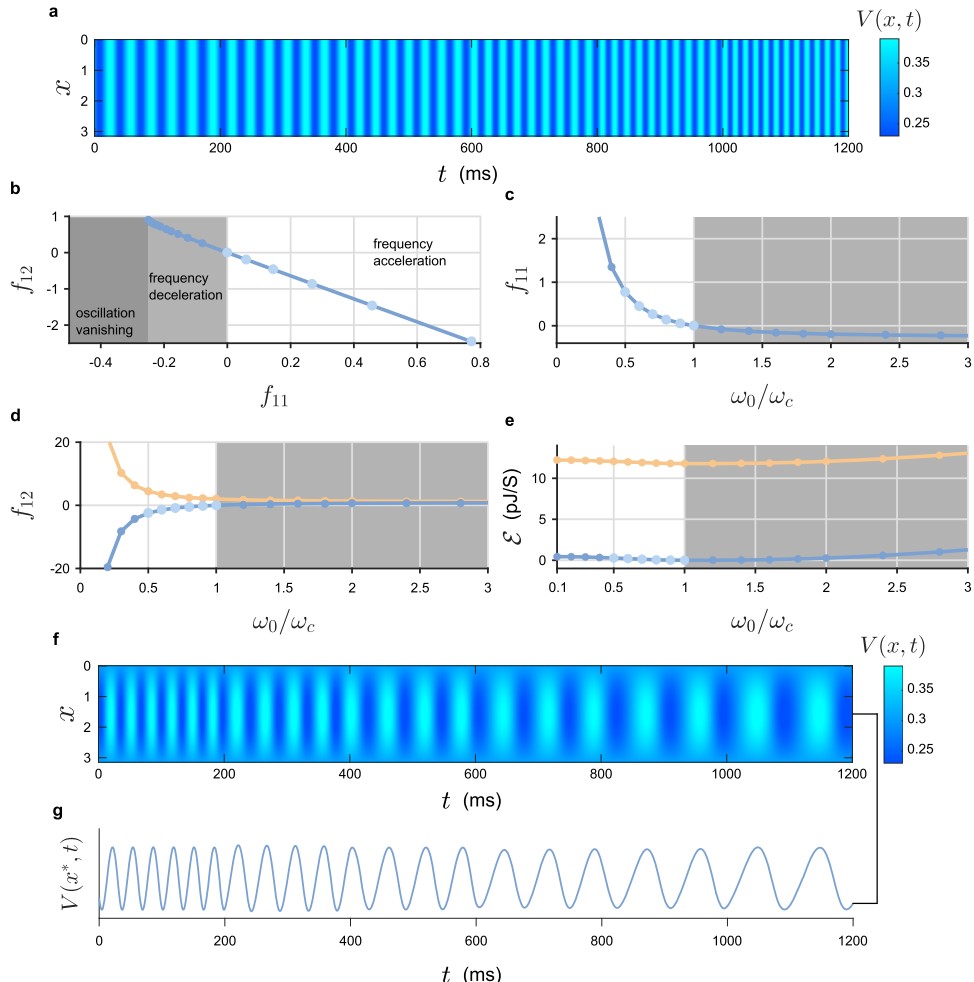

**Fig. 4 Independent frequency coordination in the neuronal model. a** The time course of the periodic oscillation in the F–N model with NBC. We achieve gradually and independently a twofold increase for the frequency by varying the coordinator at time labels $t = 200, 400, 600, 800$, and 1000. The colors (light and dark blue) of the stripes suggest a near constant amplitude. **b** The relation between $f_{11}$ and $f_{12}$ of an optimal coordinator (the one with minimum energy consumption). The solid curve and circles represent theoretical and numerical results, respectively. The oscillation is accelerated (respectively, decelerated) when $f_{11} > 0$ (respectively, $f_{11} < 0$). When $f_{11}$ approaches the critical value (approximately $-0.25$), the period of the oscillation tends to infinity. The oscillation eventually disappears in the darkest region. The coordinating policies applied in **a** as time increases are highlighted (from left to right) with light blue circles. **c**, **d** The values of $f_{11}$ and $f_{12}$ for obtaining different frequencies. Two distinct strategies are shown in different colors. Note that the horizontal axis indicates the value of $\omega_0/\omega_c$ (i.e., $1/r_F$) for readability. **e** The average energy consumed by a coordinator to sustain an oscillation at desired frequency. The two colors correspond to the two policies shown in **d**. **f** Independent frequency coordination (threefold decrease) in the F–N neuronal model with DBC. The frequency is gradually and independently decreased by varying the coordinator at each time labels $t = 200, 400, 600, 800$, and 1000. **g** The time course at $x^\star = \pi/2$.

of which are the oscillations near the Hopf bifurcation with relatively small amplitudes. Our method, based on dynamical systems tools, provides a reliable design policy for coordinating such oscillations. In practice, an oscillation far from the quiescent state with greater amplitude is, however, more likely to exist and needs to be coordinated. For such an oscillation, our approach provides an estimation of a feasible coordinator. Mostly, it may not be accurate enough to obtain desired frequency or amplitude because they are also affected by the nonlinear terms as the oscillation grows bigger. But, we can vary the four intensities $f_{ij}$ and search a more appropriate combination numerically in the neighborhood of our estimations. Compared with seeking a feasible coordinator in the entire space $\mathbb{R}^4$, the numerical investigation based on proposed approach significantly reduces the computational consumption. As two examples, we perform the independent coordinations on "bigger" oscillations in both "cancer network" (Fig. 2e, f) and F–N system (Supplementary Fig. 16).

**Independent coordinator is an amplifier or a damper.** Though we have successfully performed independent coordinations under distinct circumstances, there is still no overview picture on the coordinator configurations. That is, it is not apparent whether the four interventions in Eq. (6) are positive or negative, whereas this information could be significant for understanding the underlying mechanisms. For this purpose, we ignore the exact values of the intensities and focus only on their signs, positive (activator) or negative (inhibitor) (Fig. 6). We observe that, for both models, in order to acquire an increased frequency or amplitude, the required intervention coefficients follow the endogenous ones in the same direction (i.e., $f_{ij} \cdot a_{ij} > 0$), while we have $f_{ij} \cdot a_{ij} < 0$ for a decreased frequency or amplitude. Specifically, the designed coordinator is a reversible version of the endogenous interactions for suppressing either frequency or amplitude. In summary, the designed independent coordinator is just like a two-way dial, which can be turned up or down. When serving as an amplifier, it

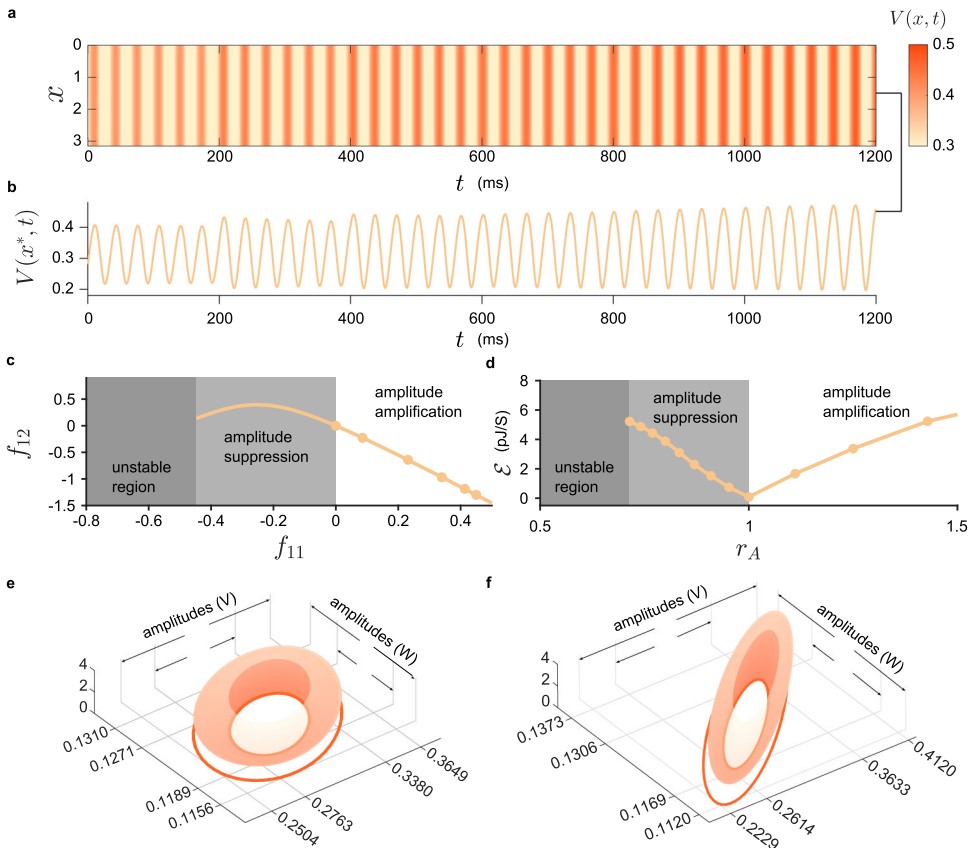

**Fig. 5 Independent amplitude coordination in the F-N model. a** The time course of $V$ for the periodic oscillation in the F-N model with NBC. The amplitude is independently increased by varying the coordinator at each time labels $t = 200, 400, 600, 800,$ and 1000. The gaps between each stripes remain invariant. See Supplementary Fig. 12 for the time course of $W$. **b** The time course at $x^* = \pi/2$. **c** The relation between $f_{11}$ and $f_{12}$ of an optimal coordinator for the independent amplitude coordination. The circles (from left to right) represent the policies applied in **a** as time goes on. When $f_{11}$ is greater (respectively, less) than zero, the amplitude of the oscillation is increased (respectively, decreased). A critical value exists (approximately $r_A = 0.723$), below which the oscillation turns out to be unstable. **d** The average energy consumed by a coordinator to sustain an oscillation at desired amplitude. When suppressing an oscillation (the light gray region), the energy consumption grows rapidly as the amplitude becomes smaller. Contrarily, the energy growth is not significant for magnifying the amplitude (white region). **e**, **f** A typical example of amplitude amplification in the F-N model with RBC. **e**, **f** show, respectively, the original oscillation and the one with magnified amplitude in $(x, V, W,)$-space. The dark circles are the projections of parametric orbits $(V, W)$ onto the $(V, W)$-plane with the greatest amplitude. In each panel, two examples of amplitudes are marked by arrows. The smaller one corresponds to $x = 0$ while the larger one to the dark circle.

accelerates and reinforces the reaction in a system raising the frequency or amplitude. Otherwise, the coordinator serves as a damper slowing down or mitigating the reaction (akin to breaks of a car).

**Hybrid coordination and energy transfer**. Besides modulating frequency or amplitude of a biological oscillator independently, sometimes a hybrid coordination is also needed to meet a certain demand (e.g., signal transduction). In this way, the frequency and amplitude can be interchanged with one another. We can design a coordinator for any values of $r_F$ and $r_A$ as long as the two algebraic equations have a solution and the coordinated oscillation remains stable. Therefore, the hybrid coordination can be easily accomplished by our approach. Figure 7d shows the region of hybrid coordination bounded by the two curves for the independent coordinations (the neuronal model with zero flux at the boundary). Note that the configurations for independent frequency and amplitude coordinations (Fig. 7f, g) obey the mechanisms presented in Fig. 6. For an oscillation moving in the hybrid region, its frequency and amplitude are coordinated in a reverse way. An example for the hybrid coordination is shown in Fig. 7a–c. The oscillation in the beginning ($t < 200$) has already experienced a frequency coordination. It has the same amplitude

and higher frequency ($r_F = 5/3$) as the original oscillation. As time goes on, it undergoes a hybrid coordination along the green curve (circles from left to right) in Fig. 7e. With such a sequence of coordinations, the amplitude is increased (up to $r_A = 5/3$), whereas the frequency is gradually decreased until it reaches the same amount as the original one ($1000 < t < 1200$). We also computed the energy consumed by the hybrid coordinators (Fig. 7h, i). Along the hybrid coordination, more energy are required from sustaining higher frequency to higher amplitude, because an oscillation with higher amplitude consumes more energy than the one with higher frequency. If the hybrid coordination is performed in an opposite way, some energy can be released. This finding is in accordance with the results shown in Fig. 6c.

## Discussion
Over the past decades, it was widely recognized that frequency and amplitude variations are fundamental processes in biological oscillators. While their precise coordinations via either internal regulation or external intervention have great benefits for health, the underlying mechanisms and the coordinating strategies are rarely studied. Especially, there are few works focusing on concurrent regulation of both quantities rather than a single one and

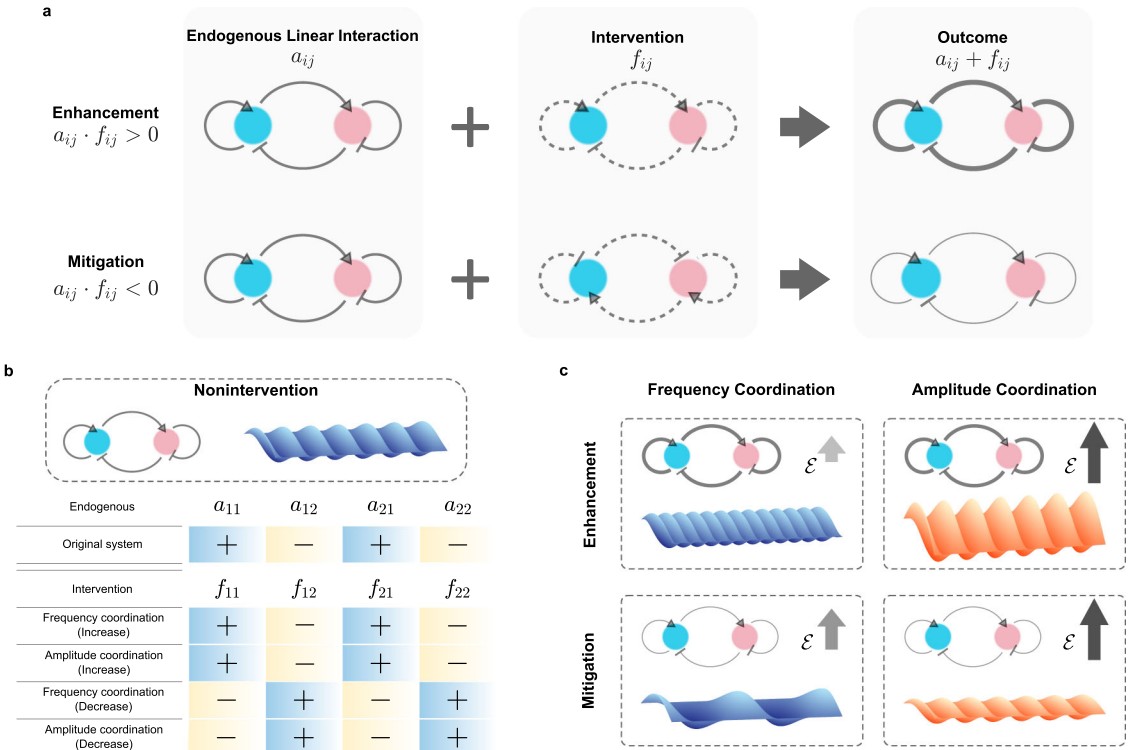

**Fig. 6 The underlying mechanisms and energy consumptions of independent frequency and amplitude coordinations. a** The optimal coordinator [the one with minimum energy consumption $\mathcal{E}$, see "Methods" for its definition] for independent frequency and amplitude coordination follows the two rules. The four linear interactions $a_{ij}$ (arrow: activation; I-shaped: inhibition) between two components (blue and red circles) are intervened by the linear regime of the coordinator [Eq. (6)] with intensities $f_{ij}$. For both independent frequency and amplitude coordination, all the four interactions are either enhanced (thicker curves) or mitigated (thinner curves) simultaneously. Thus, the coordinator can be viewed as either an amplifier or a damper. **b** The non-intervened oscillation in the neuronal model Eq. (14) (blue: $V$; red: $W$). All four configurations for independent frequency and amplitude coordination are tabulated below. Only the signs of the linear interactions and interventions are taken into account. If $f_{ij}$ (or $a_{ij}$) is positive, then component $j$ can be regarded as an activator to $i$, otherwise, an inhibitor. It can be observed that all the intervention intensities follow the same sign as that of the corresponding endogenous interaction, if an increased frequency or amplitude is desired (the case of enhancement in **a**). On the contrary, for acquiring a decreased frequency or amplitude, the designed intervention exhibits as a reversible version of the endogenous one (the case of mitigation in **a**). **c** The four intervened examples for each case given in **b**. The energy consumption for each case is sketched by arrows on the right of the motif. A higher and darker arrow suggests a greater energy consumption.

the intensities of the motif are usually ignored. Through this work, we accomplish this task with a well-designed coordinator intervening the linear interactions. We find that the interventions of linear interactions play dominant roles for the coordinations on both frequency and amplitude. Also, with two concrete and successful examples, we demonstrate that the designed coordinator are heavily dependent on the endogenous linear interactions. According to required outcomes, it is considered as either an amplifier or a damper (Fig. 6). This unprecedented finding may lead us to a better understanding of the underlying mechanism of biological oscillations. Of course, it is a challenge to address the questions: Is our discovery a universal principle? Is there any other mechanism behind frequency and amplitude coordinations? They are beyond the scope of the present article, which could be important directions for future study. The proposed coordinating strategy can be regarded as either an internal or an external intervention. Therefore, our discovery provides significant information for not only the steering of traditional biological models but also the designing of flexible artificial bio-systems in future.

Compared with the recent work focusing on the re-design of specific oscillators[32], our approach pays attention to the oscillator itself and the intensities of its linear interactions. On the one hand, we focus on coordinating the frequency and amplitude of the components inside the oscillator rather than using a main

oscillator to control an outside gene[32]. Therefore, the two works have practical usefulness under distinct circumstances. If the main oscillator is more important and needs to be modulated, then our framework is appropriate to accomplish the task. On the other hand, our designed coordinator is able to precisely acquire desirable frequency and amplitude. It provides the possibility of precise coordinations. Though it has not been verified experimentally in this work, it may be realized in future by a well-designed computer-based microfluidic device. For various biological systems, such as the neuronal model and the gene-regulatory network considered in this work, the endogenous interactions are always extractable, if its corresponding computational model is given (or discovered via machine leaning from data[64,65]). Therefore, the proposed strategy can be applied to those systems straightforwardly. This work serves as a necessary foundation for precise biorhythm regulation. It may fill the gap between experimental data and theoretical coordinating strategies making a potential contribution for precision medicine.

In the field of dynamical systems, the center manifold and the normal form theories are usually used to carry out qualitative analysis, specifically, to investigate the direction of the Hopf bifurcation and the stability of associated periodic oscillation[57]. Here we show that they are also powerful for coordinating biological oscillators. We exploit their roles for analyzing quantitative information of rhythmic oscillations. In such a way, the

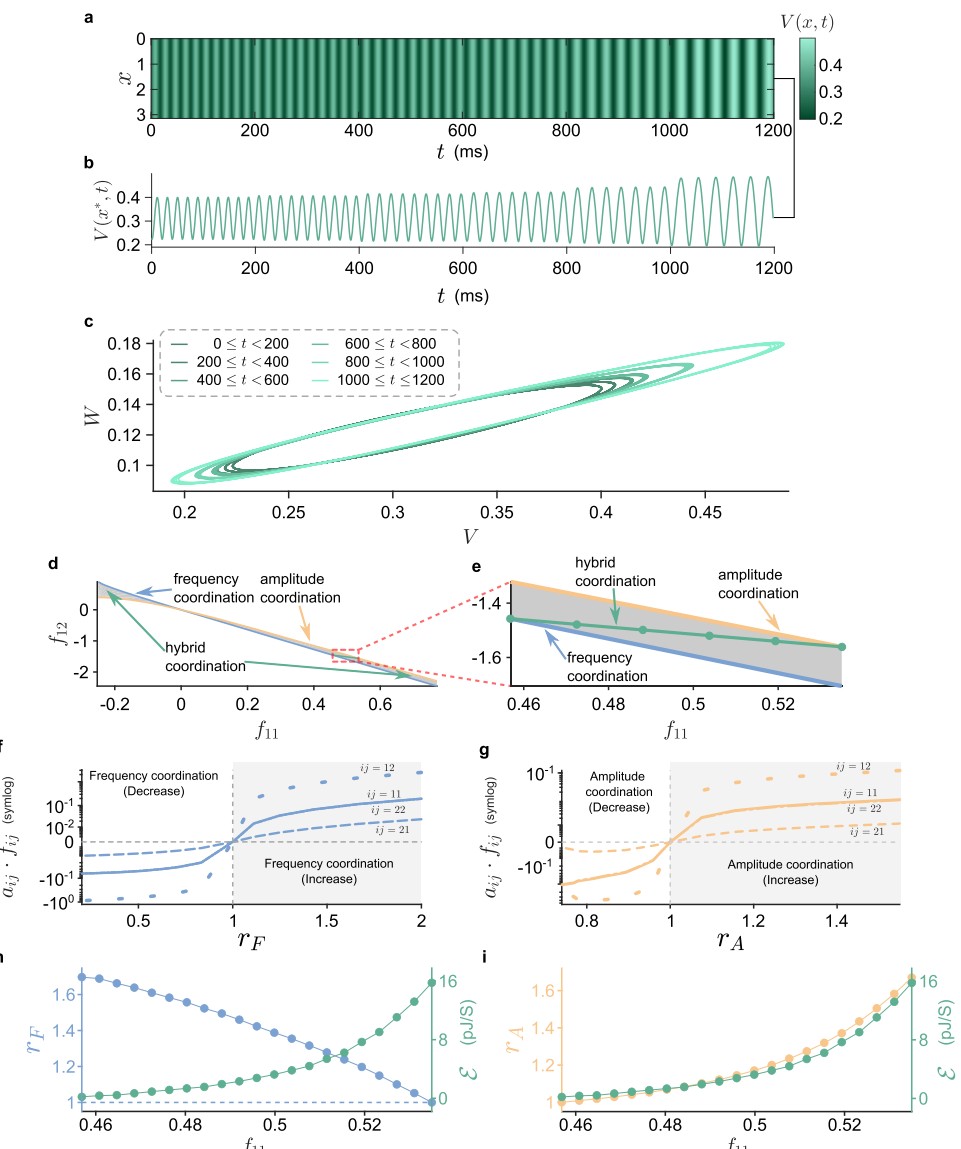

**Fig. 7 Hybrid coordination and energy transfer. a** Time course of the periodic oscillation in the F–N neuronal model (zero flux at the boundary) under hybrid coordination. Both frequency and amplitude are gradually and simultaneously coordinated. As time goes on, the frequency and amplitude undergo a 5/3-fold decrease (wider gaps) and increase (brighter colors), respectively. Different coordinators are applied at distinct time labels $t = 200$, 400, 600, 800, and 1000. For the corresponding time course of $W$, see Supplementary Fig. 17. **b** The time course at $x^* = \pi/2$ in **a**. **c** The phase portrait corresponds to the temporal profile in **b**. The larger the circle, the greater the amplitude. See Supplementary Movie 6 for the trajectories in each time interval. **d** The relations between $f_{11}$ and $f_{12}$ of an optimal coordinator for the independent frequency and amplitude coordination are shown, respectively, by the blue and yellow curves. The gray area between the two curves is labeled as hybrid coordination region, where the frequency and amplitude can be coordinated and interchangeable with one another. **e** A zoom-in view at the red rectangle in **d**. The oscillation shown in **a** is coordinated along the green curve. The policies applied as time goes on are highlighted by circles from left to right. **f**, **g** The optimal configurations for the four cases tabulated in Fig. 6b (the curves for $ij = 11$ and $ij = 22$ are not visually distinguishable). For an increased (respectively, decreased) frequency (blue) or amplitude (yellow), we have $a_{ij} \cdot f_{ij} > 0$ (respectively, $<0$). **h**, **i** The frequency (blue) and amplitude (yellow) variation ($r_F$ and $r_A$) together with the average energy (green) consumed by hybrid coordination along the green curve in **d**.

intervention intensities and desired frequency or amplitude are connected, whereby simple criteria are derived for designing coordinators. Consequently, we can alter the frequency and amplitude independently or simultaneously. With these tools, we also show that the diffusive effects are preserved from the original oscillation during the coordination so that our approach is applicable for the entire spatial domain. In fact, biological processes following certain rules are dynamical systems described by either discrete or continuous mathematical models. Besides the Hopf bifurcation yielding rhythmic oscillation, there are other bifurcations accounting for various biological phenomena[53].

Analogously, dynamical systems tools can be applied for certain biological coordinating tasks, such as controlling spatial patterns induced by Turing instability[66] and excitability related to canard phenomenon in biological systems with multiple time scales[67], to name a few.

The present approach sheds some light upon the tunability of heterogeneity biological oscillators and their energy consumption. Though we only consider two-component systems in this work, our systematic approach can be readily adapted to the one with more components following the idea that we can decouple and extract the information of frequency and amplitude via the center

manifold and the normal form theories. For instance, the Kim–Forger model describing the circadian rhythm regulated by BMAL1:CLOCK transcription factor and PER:CRY complex[53] can be investigated with the present approach. In this model, the cyclic oscillation also arises from the Hopf bifurcation. More components imply more intervention coefficients. Therefore, we would have more choices to design coordinators for such systems. More importantly, the energy consumption may be further reduced or follows a different rule compared to the one in Fig. 6. Remember that the amplitude coordination is restricted in some range due to the instability of the oscillation. With diverse coordinating policies, this range may also be extended.

Although we show that the linear regime of a well-designed coordinator works well for both "small" and "big" oscillations in the two models, its efficacy may be less than satisfactory for systems with greater nonlinear coupling effect. Also, a coordinator sometimes cannot be instantly implemented due to delay. Therefore, besides the linear interventions, we can also take advantages of nonlinear interventions (e.g., the Hill function of multimer) or the one with time lag to improve our results in future. On the other hand, an oscillator over a two- or three-dimensional spatial domain could be considered and some boundary control techniques[68] are also possible for coordinating such oscillations.

## Methods

**Generic R–D model.** The periodic oscillation arising from the Hopf bifurcation are studied in this work. Thus, for every R–D system investigated here, it possesses a constant stationary solution (i.e., homogeneous quiescent state). By appropriate transformation and Taylor expansion, this solution is moved to the origin and the system is written as Eq. (4).

In this PDE, $\mathbf{u} = \left[ u_1, u_2, \cdots, u_n \right]^\top \in \mathbb{R}^n$ and is well defined on $(x, t) \in [0, \pi] \times [0, +\infty)$, $\epsilon$ is the bifurcation parameter, $\mathbf{D}(\epsilon) = \mathrm{diag}[d_1(\epsilon), d_2(\epsilon), \cdots, d_n(\epsilon)]$ is the non-zero diagonal matrix consisting of non-negative diffusive coefficients, $\mathbf{A}(\epsilon)$ is an $n \times n$ real matrix, and $\mathbf{g}(\mathbf{u}, \epsilon)$ includes nonlinear terms that satisfies $\mathbf{g}(\mathbf{0}, \epsilon) = \mathbf{0}$. Particularly, we focus on the case of $n$, the number of components, as 2 throughout the work. The methods proposed in this work can be extended straightforwardly to a system with more components or spatial variables.

**Coordinator configuration.** The coordinator used in the computational framework is Eq. (6). For simplicity, we define a linear operator as $\mathcal{L}(\epsilon) := \mathbf{D}(\epsilon)\partial^2/\partial x^2 + \mathbf{A}(\epsilon) + \mathbf{F}$. The coordinated system is then written concisely as

$$\frac{\partial \mathbf{u}}{\partial t} = \mathcal{L}(\epsilon)\mathbf{u} + \mathbf{g}(\mathbf{u}, \epsilon). \tag{11}$$

**Eigenvalues of the operator** $\mathcal{L}$. For all the three boundary conditions considered in this work, the Laplacian operator in $\mathcal{L}$ possesses countably infinitely many eigenvalues $k_i^2$ ($k_i \geq 0$, $i \in \mathbb{N}_0$), and they are always ordered as $0 \leq k_0^2 < k_1^2 < k_2^2 < \cdots$ [69]. Further, for each $k_i$, we deduce the eigenvalue problem $\mathbf{L}(k_i)\boldsymbol{\psi}_{k_i} = \lambda_{k_i}\boldsymbol{\psi}_{k_i}$, where $\mathbf{L}(k_i) := (-k_i^2\mathbf{D} + \mathbf{A} + \mathbf{F})$ and $\boldsymbol{\psi}_{k_i}$ is the eigenvector corresponding to $\lambda_{k_i}$. Then, every $\lambda_{k_i}$ is also an eigenvalue of $\mathcal{L}$ and can be solved from the characteristic equation given by

$$\lambda_{k_i}^2 + c_1(k_i)\lambda_{k_i} + c_0(k_i) = 0, \tag{12}$$

where

$$c_1(k_i) = (d_1 + d_2)k_i^2 - a_{11} - a_{22} - f_{11} - f_{22},$$
$$c_0(k_i) = d_1 d_2 k_i^4 - [d_1(a_{22} + f_{22}) + d_2(a_{11} + f_{11})]k_i^2$$
$$+ (a_{11} + f_{11})(a_{22} + f_{22}) - (a_{21} + f_{21})(a_{12} + f_{12}).$$

For the details on the corresponding eigenvector, see Supplementary Note 2.

**Persistence and stability analysis.** Depending on the particular parameters and boundary conditions, the periodic oscillation arising from the Hopf bifurcation would be either stable or unstable[57]. First, to guarantee its existence with or without a coordinator at the same bifurcation parameter, we have the following proposition (see Supplementary Note 3 for more details).

**Proposition 1.** For Eq. (11), there exists $\epsilon^* \in \mathbb{R}$ such that both original system (i.e., $\mathbf{F}$ vanishes) and coordinated one undergoes the Hopf bifurcation when $\epsilon = \epsilon^*$.

As the Hopf bifurcation occurs at the same critical value for any $\mathbf{F}$, we deduce the theorem below (see Supplementary Note 4 for a detailed proof).

**Theorem 1.** The Hopf bifurcation of $\mathbf{u} \equiv \mathbf{0}$ in Eq. (11) always occurs at the same system parameters if and only if the matrix $\mathbf{F}$ satisfying $f_{11} + f_{22} = 0$.

Second, to guarantee that the Hopf bifurcation yields a stable oscillation, we also have the following proposition (see Supplementary Note 3 for more details).

**Proposition 2.** When $\epsilon = \epsilon^*$, the operator $\mathcal{L}$ has a unique pair of purely imaginary eigenvalues yielding the Hopf bifurcation. All other eigenvalues strictly lie in the left-half complex plane.

We now analyze the spectrum of $\mathcal{L}$ and determine $\epsilon^*$. According to Proposition 2, the roots of Eq. (12) must be non-zero and their real part are non-positive (for all $k_i$). Then, a simple manipulation (Supplementary Note 4) implies the following theorem.

**Theorem 2.** $c_0(k_i)$ is strictly positive for all $k_i$.

Generically, the quiescent state $\mathbf{u} \equiv \mathbf{0}$ undergoes infinitely many Hopf bifurcations. That is, for each $k_i$, there is a critical value such that Eq. (12) has a pair of purely imaginary roots. Consequently, there exists a sequence of periodic oscillations arising from each Hopf bifurcation. It is then natural to raise the question: Which oscillation should we coordinate its frequency and amplitude? Thanks to the following theorem (proved in Supplementary Note 4), the only case that we need to investigate is $i = 0$. All other Hopf bifurcations yield unstable oscillations that is of less interest in practice. We finally have the two theorems below.

**Theorem 3.** Assume that the stationary solution $\mathbf{u} \equiv \mathbf{0}$ of Eq. (11) undergoes a Hopf bifurcation satisfying Proposition 2. Then the unique pair of purely imaginary eigenvalues of $\mathcal{L}$ must be the roots of Eq. (12) with $i = 0$.

**Theorem 4.** The critical value $\epsilon^*$ for the Hopf bifurcation of $\mathbf{u} \equiv \mathbf{0}$ in Eq. (11) satisfies $c_1(k_0) = 0$ [in Eq. (12)], if the arising periodic solution is stable.

**Normal form and quantitative information.** Having analyzed the stability of periodic oscillation, we then compute its normal form, from which its frequency and amplitude are extracted. We do this by applying the projection method[57,70]. According to Theorem 3, the pair of purely imaginary eigenvalues is solved from Eq. (12) when $i = 0$. For readability, we drop the subscript and represent the two eigenvalues as $\lambda$ and $\bar{\lambda}$. Then, for $\epsilon$ sufficiently close to $\epsilon^*$, they can be written as

$$\lambda(\epsilon) = \mu(\epsilon) + i\omega(\epsilon), \quad \bar{\lambda}(\epsilon) = \mu(\epsilon) - i\omega(\epsilon) \quad \text{with} \quad \omega(\epsilon) > 0, \tag{13}$$

where

$$\mu(\epsilon) = \frac{1}{2}\left[a_{11} + a_{22} - k_0^2(d_1 + d_2)\right],$$
$$\omega^2(\epsilon) = -(a_{12} + f_{12})(a_{21} + f_{21}) - \xi^2,$$

and

$$\xi = f_{11} + \frac{1}{2}\left[a_{11} - a_{22} + k_0^2(d_2 - d_1)\right].$$

Following the standard procedure given in Supplementary Note 6, we finally obtain the Poincaré normal form given in Eq. (9) from which the frequency and amplitude are finally quantified.

**Modulating both components.** To approximate the amplitude of the first component $u_1$ from the normal form, we carefully determine the coefficient of the eigenvector corresponding to $\lambda$ (see Supplementary Note 6 for details). Also, if we want to implement a commensurate variation on the second component $u_2$ simultaneously, the transformation acted on $u_2$ should also be considered. Otherwise, as stated in "Results", the amplitude of $u_2$ could be increased or decreased, whereas that of $u_1$ is invariant, or vice versa. Our computation [Supplementary Eq. (S17)] shows that the approximation for the amplitude of $u_2$ includes a coefficient $\sqrt{-(a_{21} + f_{21})/(a_{12} + f_{12})}$. We finally deduce Eq. (10) [Supplementary Eq. (S18)] to eliminate this influence.

**The F–N system and its coordination.** The R–D version of the F–N model is given by

$$\frac{\partial V}{\partial t} = d_1 \frac{\partial^2 V}{\partial x^2} + V(V - \theta)(1 - V) - W + I,$$
$$\frac{\partial W}{\partial t} = \epsilon d_2 \frac{\partial^2 W}{\partial x^2} + \epsilon V - \epsilon \gamma W, \tag{14}$$

where all system parameters are positive and $\epsilon$ is the bifurcation parameter. With appropriate parameter values, it possesses a positive stationary solution denoted by $(V, W) = (V_0, W_0)$. Before investigation, we first convert the system into the form of Eq. (11). The formulation of the coordinator together with the detailed computations can be found in Supplementary Note 7.

**The index for assessing energy consumption.** For different biological models, distinct definitions may be introduced to assess the energy consumption. For the

diffusive neuronal model considered here, we adopt the classical definition used in the cable model[71]. Accordingly, the energy at time $t*$ induced by the stimulus current $I$ in a diffusive neuronal model is defined as

$$H = \int_0^{t^*} \int_0^\pi IV(x,t)\mathrm{d}x\mathrm{d}t. \qquad (15)$$

Since we consider the periodic oscillation in this work, in order to make a fair comparison, we take the average energy consumption into account. It is defined as

$$\bar{H} = \frac{1}{T} \int_0^T \int_0^\pi IV(x,t)\mathrm{d}x\mathrm{d}t, \qquad (16)$$

where $T$ is the period of the considered oscillation. To implement our designed coordinator, the linear terms $f_{11}(V - V_0) + f_{12}(W - W_0)$ (Supplementary Note 7) can be integrated as an additional stimulus current $I_c$. Our aim is to assess the energy consumption of the coordinator alone rather than the entire system. We therefore refer to an analogous power consumption index used in controlling the activity of the F–N neuron[72]. Consequently, we finally define the index for our purpose as follows

$$\mathcal{E} = |\bar{H}_c - \bar{H}_o| = \frac{1}{T} \left| \int_0^T \int_0^\pi \left[ (I_c + I_o)V_c - I_o V_o \right] \mathrm{d}x\mathrm{d}t \right|, \qquad (17)$$

where $I_o$ is the original constant stimulus current and $V_o$ and $V_c$ are the membrane potential in the original and controlled system, respectively. This index evaluates the average energy consumption of the designed coordinator.

As for the other biological models such as the "cancer network" considered in this work, the energy can be evaluated by the average $L_2$-norm of the designed coordinator over one period, which yields the following index

$$\mathcal{E} = \frac{1}{T} \int_0^T \| \mathbf{Fu} \| \ \mathrm{d}t, \qquad (18)$$

where $T$ is the period of oscillation and $\| \cdot \|$ is the $L_2$-norm induced by the inner product [Supplementary Eq. (S4)].

We introduce $L_2$-norm here because it is a generic definition for the assessment of a control policy[73] including the one used in a biological network with protein–protein interactions[63]. Moreover, the biorhythm can be regarded as a cyclic signal for information processing whose energy is also conventionally defined as the integral of $L_2$-norm[74]. Therefore, it is appropriate for the "cancer network". Of course, there could be other definition for a different model. It is worth pointing out that the two definitions introduced here can be computed in practice once the signal of a biorhythm (e.g., action-potential or expression level) is measured.

**Numerical simulations**. All the numerical simulations are performed using PDE solver *pdepe* in MATLAB R2019a (The Mathworks), and both absolute and relative tolerance are set to 1e−8.

**Reporting summary**. Further information on research design is available in the Nature Research Reporting Summary linked to this article.

## Data availability
The authors confirmed that all relevant data are included in the paper and/or its Supplementary Information. Source data are provided with this paper.

## Code availability
Code is provided in the link given in the next sentence to replicate the results shown in this paper. The codes for numerical simulations and analytical computation of the normal form are available at https://github.com/boweiqin/FMAM-in-biological-oscillators-PDE-.git.

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

# ARTICLE

38. Mogilner, A., Wollman, R. & Marshall, W. F. Quantitative modeling in cell biology: what is it good for? *Dev. Cell* **11**, 279–287 (2006).

39. Csikász-Nagy, A. & Mura, I. in *Cell Cycle Oscillators. Methods in Molecular Biology*, Vol. 1342. (eds Coutts, A. & Weston, L.) 59–70 (Humana Press, 2016).

40. Amemiya, T., Shibata, K., Du, Y., Nakata, S. & Yamaguchi, T. Modeling studies of heterogeneities in glycolytic oscillations in HeLa cervical cancer cells. *Chaos* **29**, 033132 (2019).

41. Lopatkin, A. J. & Collins, J. J. Predictive biology: modelling, understanding and harnessing microbial complexity. *Nat. Rev. Microbiol.* **18**, 507–520 (2020).

42. Rajapakse, I. & Smale, S. Emergence of function from coordinated cells in a tissue. *Proc. Natl Acad. Sci. USA* **114**, 1462–1467 (2016).

43. Hatzikirou, H., Deutsch, A., Schaller, C., Simon, M. & Swanson, K. Mathematical modelling of glioblastoma tumour development: a review. *Math. Models Methods Appl. Sci.* **15**, 1779–1794 (2005).

44. Kondo, S. & Miura, T. Reaction-diffusion model as a framework for understanding biological pattern formation. *Science* **329**, 1616–1620 (2010).

45. Llopis, P. M. et al. Spatial organization of the flow of genetic information in bacteria. *Nature* **466**, 77–81 (2010).

46. Terry, A. J., Sturrock, M., Dale, J. K., Maroto, M. & Chaplain, M. A. J. A spatio-temporal model of notch signalling in the zebrafish segmentation clock: conditions for synchronised oscillatory dynamics. *PLoS ONE* **6**, 1–18 (2011).

47. Eliaš, J. & Clairambault, J. Reaction–diffusion systems for spatio-temporal intracellular protein networks: a beginner's guide with two examples. *Computat. Struct. Biotec.* **10**, 12–22 (2014).

48. Turing, A. M. The chemical basis of morphogenesis. *Philos. Trans. R. Soc. Lond. Ser. B Biol. Sci.* **237**, 37–72 (1952).

49. Codling, E. A., Plank, M. J. & Benhamou, S. Random walk models in biology. *J. R. Soc. Interface* **5**, 813–834 (2008).

50. Ospeck, M., Eguíluz, V. M. & Magnasco, M. O. Evidence of a Hopf bifurcation in frog hair cells. *Biophys. J.* **80**, 2597–2607 (2001).

51. Izhikevich, E. M. *Dynamical Systems in Neuroscience: The Geometry of Excitability and Bursting* (The MIT Press, 2007).

52. Murayama, Y. et al. Low temperature nullifies the circadian clock in cyanobacteria through Hopf bifurcation. *Proc. Natl Acad. Sci. USA* **114**, 5641–5646 (2017).

53. Tyson, J. J. & Novak, B. A dynamical paradigm for molecular cell biology. *Trends Cell Biol.* **30**, 504–515 (2020).

54. Haragus, M. & Iooss, G. *Local Bifurcations, Center Manifolds, and Normal Forms in Infinite-Dimensional Dynamical Systems* (Springer-Verlag, 2011).

55. Dong, Y., Li, S. & Zhang, S. Hopf bifurcation in a reaction–diffusion model with Degn–Harrison reaction scheme. *Nonlinear Anal. Real. World Appl.* **33**, 284–297 (2017).

56. Guckenheimer, J. & Holmes, P. *Nonlinear Oscillations, Dynamical Systems and Bifurcations of Vector Fields* (Springer-Verlag, 1983).

57. Kuznetsov, Y. *Elements of Applied Bifurcation Theory* (Springer-Verlag, 2004).

58. Aguda, B. D., Kim, Y., Piper-Hunter, M. G., Friedman, A. & Marsh, C. B. MicroRNA regulation of a cancer network: consequences of the feedback loops involving miR-17-92, E2F, and Myc. *Proc. Natl Acad. Sci. USA* **105**, 19678–19683 (2008).

59. Chevalier, M. et al. Design and analysis of a proportional-integral-derivative controller with biological molecules. *Cell Syst.* **9**, 338–353 (2019).

60. FitzHugh, R. Impulses and physiological states in theoretical models of nerve membrane. *Biophys. J.* **1**, 445–466 (1961).

61. Nagumo, J., Arimoto, S. & Yoshizawa, S. An active pulse transmission line simulating nerve axon. *Proc. IRE* **50**, 2061–2070 (1962).

62. Rocsoreanu, C., Georgescu, A. & Giurgiteanu, N. *The FitzHugh-Nagumo Model: Bifurcation and Dynamics* (Springer, 2000).

63. Li, A., Cornelius, S. P., Liu, Y.-Y., Wang, L. & Barabási, A.-L. The fundamental advantages of temporal networks. *Science* **358**, 1042 (2017).

64. Brunton, S. L., Proctor, J. L. & Kutz, J. N. Discovering governing equations from data by sparse identification of nonlinear dynamical systems. *Proc. Natl Acad. Sci. USA* **113**, 3932–3937 (2016).

65. Champion, K., Lusch, B., Kutz, J. N. & Brunton, S. L. Data-driven discovery of coordinates and governing equations. *Proc. Natl Acad. Sci. USA* **116**, 22445–22451 (2019).

66. Strier, D. E. & Dawson, S. P. Turing patterns inside cells. *PLoS ONE* **2**, 1–4 (2007).

67. Wechselberger, M., Mitry, J. & Rinzel, J. In *Nonautonomous Dynamical Systems in the Life Sciences. Lecture Notes in Mathematics*, Vol. 2102 (eds Kloeden, P. & Pötzsche, C.) 89–132 (Springer International Publishing, 2013).

68. Krstic, M. & Smyshlyaev, A. *Boundary Control of PDEs* (Society for Industrial and Applied Mathematics, 2008).

69. Grebenkov, D. S. & Nguyen, B.-T. Geometrical structure of Laplacian eigenfunctions. *SIAM Rev. Soc. Ind. Appl. Math.* **55**, 601–667 (2013).

70. Yi, F., Wei, J. & Shi, J. Bifurcation and spatiotemporal patterns in a homogeneous diffusive predator–prey system. *J. Differ. Equ.* **246**, 1944–1977 (2009).

71. Ju, H., Hines, M. L. & Yu, Y. Cable energy function of cortical axons. *Sci. Rep.* **6**, 29686 (2016).

72. Li, F. Simulating the electric activity of FitzHugh-Nagumo neuron by using Josephson junction model. *Nonlinear Dyn.* **69**, 2169–2179 (2012).

73. Lewis, F. L. & Syrmos, V. L. *Optimal Control.* 2nd edn. (Wiley, 1995).

74. Grami, A. *Introduction to Digital Communications* (Academic Press, 2016).

## Acknowledgements

We thank the anonymous reviewers for their valuable and constructive comments that helped us to improve the work. B.-W.Q is supported by the National Natural Science Foundation of China (No. 12001110), by the China Postdoctoral Science Foundation (No. 2020M670965) and by the Shanghai Postdoctoral Excellence Program (No. 2019173). W.L. is supported by the National Key R&D Program of China (No. 2018YFC0116600), by the National Natural Science Foundation of China (Nos. 11925103 and 61773125), and by the STCSM (Nos. 18DZ1201000, 19511132000, 19511101404, and 2021SHZDZX0103).

## Author contributions

B.-W.Q. and W.L. conceived idea; all authors, B.-W.Q., L.Z., and W.L. designed and performed research; B.-W.Q. and W.L. analyzed data and wrote the paper.

## Competing interests

The author declares no competing interests.
