## [Peer Review File · Nature Communications]

Reviewers' Comments:

Reviewer #1:

Remarks to the Author:

Manuscript: NCOMMS-21-12746

Title: A frequency-amplitude coordinator and its optimal energy consumption for biorhythm

Authors: Bo-Wei Qin, Lei Zhao, and Wei Liu

Overview

Qin et al.'s manuscript is one of the few studies on the important topic of simultaneous regulation of frequency vs. amplitude in biological clocks. The work is well-grounded and innovative thanks to the derivation of its main results in the framework of reaction-diffusion systems encompassing FitzHugh-Nagumo lattices and cancer networks. The quality of the figures is good.

The work remains nonetheless of inherently technical interest and arguably more pertinent to be published in a more specialized journal such as, for example, an APS Physics Review Journal. Perhaps the material could be reshaped to convey a more impactful message to a broader audience such as that expected from Nature Communication's readers. However, I foresee insurmountable issues for this purpose, including a complete rewriting of the article and a substantial reshaping of the text. Overall, the manuscript is generally poorly written and embraces the wrong choice of omitting essential modeling details. It is often hard to understand what the authors refer to and the logical flow of their study. I am elaborating these aspects in the following brief list of suggestions for improvement. In its current version, I am afraid I am discouraging publication in NC.

Suggestions for improvements.

At the Introduction level, I think you could rephrase your reasoning, emphasizing first practical examples of frequency vs. amplitude regulation matter. And also give some ideas of where their combined regulation could be of practical importance. Then, develop the theoretical canvas, linking clearly different positive/negative feedback mechanisms and their interplay with frequency/amplitude regulation. For example, in the current version of the manuscript, it is not clear where you take inspiration for distinction from endogenous vs. exogenous mechanisms. Finally, I found it hard to link what energy is in your framework. I assume, of course, that you are referring to the concept from a classical Systems' point of view. Still, it would be much more helpful to clearly relate what you define as energy with what is measurable in practice in the examples of biological systems you consider.

Concerning the exposition of your procedure in "A Universal policy for the coordinator," I would consider rewriting it entirely. You chose to omit the essential equations and use symbols that are not clearly understandable if not framed into equations. Moreover, you present critical results without explaining all the logical steps. That is detrimental to your text readability.

Personally, I would envisage a classic structure: 1. Introduction, 2. Procedure outline: (potentially broken down into 2-3 steps, based on figure 3, with the essence of your analytical results. 3. Energy characterization, including better definitions and explanations. 4. Applications: again divided into three subsections: 4.1. FHN (frequency regulation only); 4.2. Amplitude regulation in the gene network; 4.3. Hybrid regulation in the cancer network. 5. Discussion/Conclusions.

Figure 1. Why not fix the figure shape on the right and instead use figure size for amplitude and color-coding for frequency. What is the application of hybrid frequency-amplitude regulation?

Figure 2. This figure could actually be elaborated, showing examples of frequency vs. amplitude regulation in all possible four configurations if available (F-A ++/ +- / -+ / - -).

Figure 3. Should precede Figure 2.

Figure 4 vs. Figure 5: I do not understand why you don't keep the same order of exposition between the two figures (akin to Figure 6 w.r.t. Figure 4).

Figure 6. Why not consider plotting panel C in symlog scale or in a way that does not require zooming in but rather shows both regions on the left and the right sufficiently well?

It would be very nice to extend this figure with graphs accounting for the possible configurations just tabulated in Figure 2.

A final remark concerns your Online Methods. Propositions (sometime, more appropriately, Lemmas) should be demonstrated accordingly. Conclusions are more properly Theorems or Corollaries, and once again should be properly proved. I could not find clear proofs of your statements, except for Corollary 3, neither in your Online Methods section, nor in your Supplementary Material.

Reviewer #2:

Remarks to the Author:

The manuscript "A frequency-amplitude coordinator and its optimal energy consumption for biorhythm" by Qin, Zhao and Lin describes a novel biorhythm control device ("coordinator"). The coordinator design consists of four feedback loops to modulate both frequency and amplitude simultaneously. Importantly, two of these four feedbacks may be endogenous biorhythm circuits, and the authors illustrate how to design coordinators to precisely modulate endogenous circuits. The coordinator relies on the center manifold and normal form theories to derive parameters that achieve a desired amplitude and frequency (within the stability limits of the system). They focus on two published models of endogenous biorhythm circuits: neuronal activation and a protein-miRNA circuit. In addition, the authors quantify the energy consumed by different coordinator policies that achieve the same output, allowing facile selection of the best policy. Their framework is both elegant and rigorous, and though it has not yet been tested experimentally, it has great potential for guiding future interventions of endogenous oscillators.

Comments to be addressed in a revision:

1. The author's figures and proofs are clear and intuitive. However, their sentences are often incomplete. Many sentences begin with conjunctions (such as "because" and "whereas") and should be joined with the previous sentence.
2. The legend for Figure 1b is confusing. They illustrate a network where a miRNA regulates a protein (by regulating its mRNA abundance), and that protein regulates both itself and the miRNA. They describe this as a "protein-mRNA interaction"; perhaps "protein-miRNA feedback" or "protein-miRNA circuitry" would be clearer.
3. In Figure 5g legend the authors incorrectly cite reference 9 instead of 59 (cancer network). They refer to protein-mRNA oscillation in the cancer network, but reference 59 describes a cancer network consisting of a miRNA cluster and proteins (E2F and MYC). The authors should clarify if the reported quantity represents this miRNA cluster.
4. We note that the authors focus on two relatively simple endogenous biorhythm circuits. Could the authors comment on more complex oscillators that control important biorhythms (CLOCK/BMAL, COOLAIR/COLDAIR)? Can the authors predict whether their coordinator can be applied to such networks?
5. We recommend the authors make their Matlab and Maple code public upon publication. This would greatly facilitate application of their design process to other biorhythm circuits.
6. Units of time are not specified in any figures or legends.

Reviewer #3:

Remarks to the Author:

The authors state that to understand the spatio-temporal dynamics of many biological phenomena one needs to consider both amplitude and frequency of these dynamics. They then argue that the amplitude is often ignored, and in particular that both amplitude and frequency are often not considered together. They then claim that they found "for the first time" a "universal approach" to address these questions.

I have a very hard time following the main arguments and conclusions of this paper. I have expertise in modeling and experimenting on similar systems discussed. Overall, the paper's figures

and equations appear well presented at a first look. The mathematical derivations appear correct as far as I checked.

But I am very confused on what the authors actually want to say – and how that aids the understanding actual biological systems.

For example, in Fig.2 they discuss two entities that are coupled with each other. Such bidirectional networks have been studied a lot in the past. The dynamic phenomena that can arise are very complex even for such a simple system that just have one self- and one cross reaction term for each entity. The authors now increase the complexity by doubling the number of interactions. Has their analysis really captured all possible complexity arising from that? How does that relate the specific neuron and cancer systems?

As another example, the authors motivate the importance of coordinating frequency vs. amplitude – I would expect some phase space depiction analyzing both jointly – but I do not recognize that. Also - what determines amplitude and frequency? For example, frequency is typically determined by the relevant time scales in the systems, e.g., degradation rates or signaling delays; amplitudes are typically given by ratios of activation and degradation [for example, Alon 2006 Introduction System Biology, CRC Press, or Glass 2021 Delay Equations, Nature Communications]. None of such features are discussed here.

Potentially this paper is geared towards the pure mathematical audience. In that case I wish to see at least two reviewers with the corresponding background to attest the relevance and correctness of this work. Furthermore, the authors need to make the relevance for the discussed biological systems significantly more transparent so that it can inform the research of experimentalists working in those fields.

Reviewer #4:

Remarks to the Author:

This paper deals with the problem of orthogonal control of amplitude and frequency of biological oscillators. The authors propose a number of feedback controllers that can achieve these goals for specific classes of bifurcations. The problem is interesting and the paper well written, but I found the scope and results too narrow for Nature Communications - the paper is better suited for a specialised journal in control or nonlinear dynamics, as the main results are mathematical in nature and their applicability to biological systems is unclear.

There is also a concern on novelty with respect to this paper

<https://www.sciencedirect.com/science/article/pii/S2405471218301108>

Which is cited lightly, but in my opinion quite directly comparable to the results on the protein-mRNA interaction model. I missed some more in depth discussions on how their results compare to that previous work.

Specific points:

- It is shown that the feedback controllers can indeed achieve the goal, but there is no guarantee that any such controllers can be implemented by a real biochemical mechanism. There is plenty of recent literature on equivalences between PID controllers and biochemical implementations (particularly in Cell Systems journal), but the natural implementation of a general linear feedback controller as obtained in this work is in general not possible.
- The authors show the utility of this approach in two model biological systems. But because of the impossibility of finding biological circuits that implement the controller in both case studies, the examples somewhat diminish the extent of the theoretical contribution.
- The paper touches on energy considerations for each controller, but I found these results quite disconnected from the way energy consumption is defined in neural models and the protein-mRNA model.

I do not wish to discourage the authors in the value of their theoretical contribution, but given the considerations above I believe the paper is better suited for a control theory or nonlinear dynamics journal.

Responses to the reviewers' comments and changes made in the revised manuscript

First of all, we would like to express our great appreciation to all the constructive comments and questions made by the reviewers.

We apologize that the previous manuscript was not clear enough for the audiences to follow our approach, especially we have not explained clearly its differences with the existing work in the literature. Also, some figures and text confused the audiences and left an undesirable impression. For instance, in the previous manuscript, it seems that our approach complicates the oscillator which diminished practical usefulness of our work. With the help of reviewers' comments, in addition to addressing the technical issues, we have also expressed our idea and approach in a clearer way, and emphasized the novelty (compared with the existing work in the literature) and practical relevance of our work.

The idea of frequency and amplitude modulations in biological systems was proposed several decades ago. However, the progress in this area has been very slow. As far as we know, there is no solid experimentally proved research in this area so far. Also, there are very few theoretical works. Especially the one with a rigorous computational framework is still lacking. The reason is that we must need mathematical theories to deal with this kind of research systematically, but the two areas (applied mathematics and theoretical biology) are somehow separated. As stated in very recent papers [JJ Tyson & B Novak, Trends in Cell Biology, 2020; AJ Lopatkin & JJ Collins, Nature Reviews Microbiology, 2020], the dynamical systems tools including bifurcation theories are very significant for solving biological problems. For the biologists, it is not easy to find what sort of mathematical tools can be used to solve their problems. For the mathematicians, they have feasible tools but may not realize what kind of tools can be applied in the area of biology. Therefore, we believe that it is a good opportunity to communicate with scientists in both areas through our work if it is published in Nature Communications. If it was published in a specialized journal such as: control theory and nonlinear dynamics, we think less experimentalists and biologists would notice this work. As a consequence, the progress of this area may be delayed once again. Via this work, we want to inform not only the biologists that the problem can be solved using mathematical tools but also the mathematicians that they can use their abilities to conquer the biological difficulties in a systematical way. We think that this one small step is significant for improving the future studies in frequency and amplitude modulations. Therefore, we would greatly appreciate it if the reviewers are willing to support our work. In our opinion, their support will be beneficial to the area of frequency and amplitude modulations.

We list in this document our point-by-point responses (blue) to all the reviewers as well as the corresponding changes made in the manuscript (orange). In the revised manuscript, the changes are coloured with blue. The major revisions are also provided here point-by-point.

Content

Responses to reviewer #1	2
Responses to reviewer #2	10
Responses to reviewer #3	12
Responses to reviewer #4	17

Responses to reviewer #1

1. At the Introduction level, I think you could rephrase your reasoning, emphasizing first practical examples of frequency vs. amplitude regulation matter. And also give some ideas of where their combined regulation could be of practical importance. Then, develop the theoretical canvas, linking clearly different positive/negative feedback mechanisms and their interplay with frequency/amplitude regulation. For example, in the current version of the manuscript, it is not clear where you take inspiration for distinction from endogenous vs. exogenous mechanisms.

Thanks for the very constructive suggestions. We apologize that we have not expressed our idea clearly.

In the revised manuscript, we have rewritten Introduction section (mainly in its first several paragraphs) and some sentences in the legend of Figure 1. We have deleted the common background of circadian rhythm and emphasizing the significance of frequency and amplitude in the beginning so that the audiences can follow up the topic immediately. Then, we reviewed the related works on frequency and amplitude modulations in the literature and emphasizing the differences between their ideas and ours which further leads the audiences to our method. Note that references [8], [34], [35] are newly added.

Introduction

Plenty of living organisms as well as synthetic biological networks have their capacity to generate rhythmic processes involving molecules, cells, and tissues¹⁻⁵. For instance, neuron spiking, cell reproduction, hormone secretion, protein synthesis, and heartbeat are common periodic activities in the human body. These processes are usually regulated by single or multiple biochemical oscillators exhibiting various frequencies and amplitudes (Fig. 1). The former ones would result in distinct functional consequences⁶, while the latter ones abstract expression levels meeting body demands for protein, hormone, energy to name a few⁷. More importantly, they control the identity and intensity of a signal being critical to internal information transduction⁸. Therefore, their coordinations are essential to physiological behaviors, such as sleeping, feeding and mood⁹⁻¹³.

Oscillators with anomalous frequency or amplitude may disrupt biorhythm and lead to sleep and metabolism disorders or even diseases¹⁴⁻¹⁶. Fairly recent perspectives also addressed that circadian rhythmicity in our body can be leveraged to develop chronotherapy and to account for the mechanism behind it^{17,18}. Harmonizing administration of agent with associated biological target can improve its efficacy and reduce toxicity. Thus, flexible biochemical oscillators have a wide range of benefits for healthy sustainability. As a consequence, designing practical coordinators to acquire desirable frequency or/and amplitude becomes a growing and significant issue.

* **Correspondence** and requests for materials should be addressed to B.W.Q. (boweiqin@fudan.edu.cn) or W.L. (wlin@fudan.edu.cn).

The theoretical study on biorhythm dates back to 1960s when Winfree studied frequency exclusively in his pioneering work¹⁹. Since then, the adjustment of frequency via synchronization were thoroughly studied using reformulated Kuramoto model²⁰. In contrast, there are few studies focusing particularly on amplitude. Over past decades, frequency and amplitude modulations of biochemical oscillators attract more and more attentions, among which coordinating frequency or amplitude independently (Fig. 1a, b) is particularly significant.

For biological models, the dynamics depends highly on their network motifs (pattern of positive/negative interactions) and significantly attract plenty of attentions²¹⁻²³. Therefore, investigating the mechanisms between the motif and frequency-amplitude coordination becomes a natural direction. Auto-regulations via endogenous interactions were ubiquitously studied²⁴⁻²⁸. Prior works also suggested that the tunability of frequency and amplitude can be enhanced by positive feedback loops, while the independent coordinations may not be achieved if only negative feedback loops appear²⁹⁻³¹. Besides, a very recent work³² studied the dual-feedback oscillator and the repressilator and the modifications of their network architectures. The frequency and amplitude of a gene outside the main oscillator are “physically” decoupled by re-designing the oscillator, thus, they can be modulated independently in extended ranges.

Actually, the regulations of frequency and amplitude of the components inside an oscillator are also significant. Moreover, besides the network topology, the intensities of endogenous interactions affect frequency and amplitude as well, but there are few related works³³. The two quantities can be regulated by making a slight intervention on these intensities. It can be realized either

internally by varying the system parameters (e.g., degradation rate, synthesis rate, strength of a stimulus, etc.); or externally by a feedback controller (e.g., computer-based microfluidic devices^{34,35}). More importantly, the underlying mechanisms for coordinations as well as their energy consumptions also depend highly on the intensities, which have not yet been uncovered. In this article, we present a universal computational framework for precisely designing such an intervention coordinator. We will try to answer the questions: What specific form of an intervention should be made (i.e., enhancing or mitigating the original interactions)? How is its intensity and energy consumption? Additionally, with our well-designed coordinator, both frequency and amplitude can be regulated precisely and concurrently (Fig. 1c).

FIG. 1. Independent and hybrid coordinations in representative biological models. **a**, *Frequency coordination.* Periodic biological event is generated due to neuronal excitability exhibiting diverse frequencies under distinct circumstances. They are sketched by heterogeneous wave-like patterns (i)-(iii) on the right. On the rightmost, the projections of the waves are shown to represent their amplitudes. Under (independent) frequency coordinations, the rates of occurrence can be modulated in both ways (acceleration or deceleration) while the amplitude (intensity of the event) remains invariant. **b**, *Amplitude coordination.* The rhythmic biological processes also exist in genome-scale transcriptional and translational processes such as protein-miRNA interaction. The protein concentration in a specific gene regulatory network changes throughout the day. It is represented by the amplitude of the wave-like pattern (right) and is varied under (independent) amplitude coordinations. In the meanwhile, the frequency of these processes are not changed. **c**, *Hybrid coordination.* A biorhythm disorder may include both frequency and amplitude disruption which further influences internal signal transduction. With hybrid coordination, the two quantities are modulated simultaneously, and they are interchangeable with one another. Compared with pattern (vii), (viii) has a higher frequency and a lower amplitude. An opposite situation is illustrated as pattern (ix). Biological events in the space may occur under different boundary conditions. The amplitude projections shown in the rightmost of **a**, **b** and **c** are typical diagrams for the Dirichlet, Robin and Neumann boundary condition, respectively.

- [8] Hansen, A.S. & O’Shea E. K. Limits on information transduction through amplitude and frequency regulation of transcription factor activity. *eLife* **4**, e06559 (2015).
- [34] Lugagne, J.-B. et al. Balancing a genetic toggle switch by real-time feedback control and periodic forcing. *Nat. Commun.* **8**, 1671 (2017).
- [35] Perrino, G. et al. Automatic synchronisation of the cell cycle in budding yeast through closed-loop feedback control. *Nat. Commun.* **12**, 2452 (2021).

2. Finally, I found it hard to link what energy is in your framework. I assume, of course, that you are referring to the concept from a classical Systems’ point of view. Still, it would be much more helpful to clearly relate what you define as energy with what is measurable in practice in the examples of biological systems you consider.

Thanks for the reviewer’s comment. Indeed, we referred to the concept from dynamical systems and control theory. When studying a biochemical system and its energy consumption, the conventional mathematical or physical concepts are usually used. Of course, there are more appropriate definitions for the energy consumption of the entire system (especially for some neuronal models). But we focused on the energy consumed by the designed coordinator instead of the entire system. The energy defined in our work is only a part of the system. Moreover, as our computational framework is universal, we desire to define the energy consumption that can be applied in the generic case. Therefore, as far as we have searched in the literature, we think that the L2-norm is the most suitable definition for our purposes. It is commonly used to assess a control policy. For instance, in a recent Science article [A. Li et al., The fundamental advantages of temporal networks, *Science*, 2017], the authors also used L2-norm to define the energy consumption of the control policy and they also considered a protein-protein interaction network. Also, we studied the frequency and amplitude of the periodic oscillation in this work. Such an oscillation (both in neuronal and protein-miRNA model) can be viewed as a cyclic signal for information processing. Then, it is reasonable to use L2-norm to define the energy of an oscillating signal. Once the signal is measurable (such as: action potential or expression levels), our defined energy can be readily computed. We are sorry that, in the previous version, the reason for using the L2-norm was not explained clearly.

We have made some changes in the revised manuscript to clarify the reason that we choose L2-norm and to link it closely with biorhythm. We have rewritten the second sentence in section “The optimal energy consumption of a coordinator” and included a new reference [61]. Also, we added one more paragraph and more references [61], [70], [71] in Methods, section “The index for assessing energy consumption”, to clarify the reason.

The optimal energy consumption of a coordinator. In practice, the consumption of a coordinating policy for sustaining an oscillation is of great significance. Therefore, an index \mathcal{E} is always introduced to measure the performance of a specific control policy⁶¹, see Methods for the one used in our problem. Fig. 4e shows the

The index for assessing energy consumption. We evaluate the average L_2 -norm of the designed coordinator over one period, which yields the following index

$$\mathcal{E} = \frac{1}{T} \int_0^T \|\mathbf{F}\mathbf{u}\| dt, \quad (10)$$

where T is the period of oscillation and $\|\cdot\|$ is the L_2 -norm induced by the inner product [Supplementary equation (S4)].

Note that the index \mathcal{E} evaluates only the energy consumed by the coordinator rather than the entire system, because we are more interested in the added interventions. There could be other definitions for assessing the energy consumption, especially for a specific model. Here, we use the L_2 -norm because it is a generic definition for the assessment of a control policy^{61,70}. Thus, it is more appropriate for our universal framework. More importantly, The biorhythm can be regarded as a cyclic signal for information processing (such as: neuron-neuron or cell-cell) whose energy is conventionally defined as the integral of L_2 -norm⁷¹. Therefore, our definition can be computed in practice once the signal of a biorhythm (e.g., action-potential or expression level) is observed.

[61] Li, A., Cornelius, S. P., Liu, Y.-Y., Wang, L. & Barabási, A.-L. The fundamental advantages of temporal networks. *Science* **358**,

[70] Lewis, F. L. & Syrmos, V. L. *Optimal Control*, Second Edn. (Wiley, New York, 1995).

[71] Grami, A. *Introduction to Digital Communications*. (Academic Press, 2016).

3. Concerning the exposition of your procedure in “A Universal policy for the coordinator,” I would consider rewriting it entirely. You chose to omit the essential equations and use symbols that are not clearly understandable if not framed into equations. Moreover, you present critical results without explaining all the logical steps. That is detrimental to your text readability.

Thanks for the reviewer’s comment. We agree that the procedure was not clear enough in the previous manuscript because we placed the essential equations into “Methods” section. To improve the readability, we have made changes accordingly.

We have rewritten the first and the second section in “Results” and moved the essential equations into their right place so that the audiences can easily follow our ideas and procedures. In the second section, when introducing our computational framework, we separate it into four steps. We have tried to use a list, but it made the article looks bad. Finally, we divided the procedure into several paragraphs, each of which starts with an indication.

Endogenous linear interactions and their interventions. We illustrate our computational framework for the frequency-amplitude coordinator using the two-component diffusive system which can always be modeled as (see Methods for a detailed description):

$$\underbrace{\frac{\partial \mathbf{u}(x, t)}{\partial t}}_{\text{Time evolution}} = \underbrace{D(\epsilon) \frac{\partial^2 \mathbf{u}(x, t)}{\partial x^2}}_{\text{Spatial diffusion}} + \underbrace{\mathbf{A}(\epsilon) \mathbf{u}(x, t)}_{\text{Linear interaction}} + \underbrace{\mathbf{g}(\mathbf{u}(x, t), \epsilon)}_{\text{Nonlinear interaction}}. \quad (1)$$

Such a system possesses two self-interactions and two cross-interactions, all of which can be divided into two parts: the linear interactions $\mathbf{A}\mathbf{u}$ and the nonlinear ones \mathbf{g} . The matrix \mathbf{A} is associated with the Jacobian matrix and can be acquired in any computational model²¹. It represents the endogenous linear interactions, which consists of four coefficients a_{ij} ($i, j = 1, 2$) representing associated interaction (u_j to u_i). The sign of a_{ij} indicates the role played by the component j (activator or inhibitor), see Fig. 3a, b. The absolute values represent their intensities which are determined by system parameters, such as: degradation rate, synthesis rate, Michaelis constant, etc. These parameters are usually different from system to system (see Supplementary Section 9 for more details on the linear interactions).

For a given biochemical system, the endogenous linear interactions always exist in its computational model and have dominant impacts on the behavior of an oscillation near the quiescent state. Specifically, the intensities of these interactions determine the frequency and amplitude decisively. Therefore, our aim is to design a coordinator to intervene those interactions which further regulate frequency and amplitude precisely. Our coordinator is designed as:

$$\begin{bmatrix} u_1 \rightarrow u_1 & u_2 \rightarrow u_1 \\ u_1 \rightarrow u_2 & u_2 \rightarrow u_2 \end{bmatrix} := \mathbf{F}\mathbf{u} = \begin{bmatrix} f_{11} & f_{12} \\ f_{21} & f_{22} \end{bmatrix} \begin{bmatrix} u_1 \\ u_2 \end{bmatrix}, \quad (2)$$

which has the same form as $\mathbf{A}\mathbf{u}$ because it is a perturbation of the endogenous linear interactions. The coordinator can be implemented either internally by varying the system parameters or externally as a feedback controller (e.g., the computer-based microfluidic devices^{34,35}). After implementing the coordinator, the resulting linear interactions become $(\mathbf{A} + \mathbf{F})\mathbf{u}$ (see Fig. 3a). Now, the key question is: How to determine the intensities of those interventions for desired frequency and amplitude? Later,

we will show that they depend highly on the associated endogenous ones.

A universal design policy for the coordinator. To design a feasible coordinator, we need to know how its intensity relates to the frequency and amplitude. Here, we apply the center manifold and the normal form theories to accomplish the task (Fig. 2). Our framework includes following steps.

We first perform a persistence analysis (see Methods) before moving into detailed computations. It guarantees the existence of the oscillation after intervention.

This gives us Theorem 1 which provides the first criteria ($f_{22} = -f_{11}$) for designing the coordinator. It indicates that the two self-interactions of a feasible coordinator must be opposite to one another. Thus, a negative feedback loop is established providing the potential to produce rhythmic oscillation in a biological system²⁹. Reasoning that it ensures the occurrence of the Hopf bifurcation leading to a cyclic oscillation. In practical applications, a stable oscillation is of great significance and interest.

The second step is to link the intensities f_{ii} in a coordinator with the frequency and amplitude. Therefore, we exploit the normal form of the Hopf bifurcation⁵⁷ utilizing the center manifold and the normal form theories. For a periodic oscillation, the Poincaré normal form is always found as (see Methods for more details)

$$\dot{w} = \lambda w + \eta w^2 \bar{w} + \mathcal{O}(|w|^4), \quad w \in \mathbb{C}, \quad (3)$$

where $\lambda = \mu + i\omega$ is the complex eigenvalue corresponding to the Hopf bifurcation, and η is the normal form coefficient whose real part, denoted by χ , is the first Lyapunov coefficient. It is the simplest form preserving the information, and is usually applied to study qualitative behaviors. Here, we extract the quantitative information (frequency and amplitude) from the normal form. It serves as a bridge from the original oscillation to the coordinated one. Therefore, those information are helpful for designing the coordinator. The quantifying procedure is illustrated in Fig. 2.

The third step is to modulate both components consistently. We notice that they always share the same frequency, whereas their amplitudes could be distinct. As a consequence, the variations made by the coordinator may be inconsistent, in the sense that, the amplitude of u_1 (e.g., protein concentration) is increased or decreased while that of u_2 (e.g., miRNA concentration) is invariant, or vice versa. Such a result may be unexpected for a given biological system. To avoid this circumstance, we analyze both components when computing the normal form (see Methods). We surprisingly find that the modulations on both amplitudes are commensurate with one another if we have:

$$\frac{f_{21}}{f_{12}} = \frac{a_{21}}{a_{12}}. \quad (4)$$

This becomes the second criteria for the design policy. Reasonably, it incorporates the endogenous cross-interactions and their interventions. The ratio of two

amplitudes is affected by the cross-intensities. To keep this ratio unvarying, the intensities of the interventions must also follow a fixed ratio determined by the intrinsic ones (i.e., a_{21}/a_{12}).

In the final step, we will derive two algebraic equations for the coordinator. From equation (3), the frequency and amplitude are extracted and approximated as $\omega/(2\pi)$ and $2\sqrt{-\mu/\chi}$, respectively. Note that μ is independent of the intervention coefficients, whereas ω and χ are expressed in terms of f_{11} and f_{12} , i.e., $\omega = \omega(f_{11}, f_{12})$ and $\chi = \chi(f_{11}, f_{12})$. Before we go any further, it is worth pointing out that the approximated amplitude is independent of x (i.e., the spatial variable). In spite of this, the coordinator designed here still works for the entire space, because the spatially heterogeneous effect is eliminated during the computation (see Supplementary Sections 5 and 6 for more details) and it does not affect the final coordination.

4. Personally, I would envisage a classic structure: 1. Introduction, 2. Procedure outline: (potentially broken down into 2-3 steps, based on figure 3, with the essence of your analytical results. 3. Energy characterization, including better definitions and explanations. 4. Applications: again divided into three subsections: 4.1. FHN (frequency regulation only); 4.2. Amplitude regulation in the gene network; 4.3. Hybrid regulation in the cancer network. 5. Discussion/Conclusions.

Thanks for the reviewer's comment. In mathematical viewpoint, we agree that the classical structure you mentioned is a better and a more logical way to write the manuscript. The reason why we use an atypical mathematical structure is that the potential audiences come from different disciplines including biologists and experimentalists, we hope that putting the "Results" section right after introduction is a friendlier way because it is the conventional way in their area. Our work and its findings have the potential to inform them for possible studies in future. We focus not only on the mathematical theory and computational methods but also on the potential applications in practical biological systems. Therefore, we finally chose to put the manuscript in its current structure to increase the readability of our manuscript. We have thought about using hierarchical section and finally abandoned it because the number cannot appear in the section title for a Nature Communications article. Without a number, too much hierarchy would make the manuscript unclear. Therefore, we finally separate them into several sections, each of which represents one subtopic. We are sorry that we cannot satisfy all the audiences, but we have tried our best to make a balance to improve the readability.

As mentioned before, in the new manuscript, we have divided our procedure into four steps and rewritten the first and second section for a better presentation. We hope that it becomes clearer than the previous version. Also, regarding the energy, we have explained the reason and provided more details and references in "Methods" section. We have not put these details in the main text because there is a word-number limitation.

5. Figure 1. Why not fix the figure shape on the right and instead use figure size for amplitude and color-coding for frequency. What is the application of hybrid frequency-amplitude regulation?

Thanks for the reviewer's comment and questions. When we were preparing this figure, we found that it is difficult to use figure size to distinguish the amplitude of a 3D wave due to inevitable visual error. Therefore, we chose to show the amplitude of each wave in a two-dimensional space which provides clearer comparisons. As for the frequency, we thought it is intuitive to show different cases with the same period of time. Then, the audiences just need to count the number of peaks for each case. If we used color-coding for frequency, the information would be redundant and the whole figure could be more complicated yielding low readability. If we chose color-coding frequency and showed one period for each wave, then the waves would have different length which might diminish the elegance of the figure. Therefore, we finally decided to keep the right part of Figure 1 as it is. Of course, we may misunderstand your comment. Please let us know if so.

A biorhythm disorder includes both frequency disruption and amplitude disruption and they may occur simultaneously in a biological system. Also, the two quantities play significant roles in internal signal transduction (the frequency represents the identity and the amplitude represents the intensity). Therefore, how to calibrate them concurrently is a significant question to be answered. In the area of frequency and amplitude modulations, the two quantities are usually considered independently. In this work, we showed that our approach is able to coordinate both frequency and amplitude simultaneously. Therefore, we include a third case in our work to emphasize this feature and call it the hybrid coordination. It can be applied to reset an anomalous biorhythm in both frequency and amplitude to a normal one.

In the revised manuscript, we have included one more illustration in Figure 1c to present the possible application of the hybrid coordination. Accordingly, we have changed the corresponding legend. Also, in the introduction part, we have emphasized its application and practical usefulness, and included one more reference [8] to support it.

under (independent) amplitude coordinations. In the meanwhile, the frequency of these processes are not changed. **c**, *Hybrid coordination*. A biorhythm disorder may include both frequency and amplitude disruption which further influences internal signal transduction. With

[8] Hansen, A.S. & O'Shea E. K. Limits on information transduction through amplitude and frequency regulation of transcription factor activity. *eLife* 4, e06559 (2015).

6. Figure 2. This figure could actually be elaborated, showing examples of frequency vs. amplitude regulation in all possible four configurations if available (F-A ++/ +- / -+ / --).

Thanks for the reviewer's constructive suggestion.

In the new manuscript, we have redrawn the Figure 2 with more information. Note that it becomes Figure 3 in the revised version. In Figure 3a, to avoid possible confusion of our approach, we use two "motif equations" to clarify the mechanisms behind independent frequency and amplitude coordinations. Furthermore, we show the concrete examples visually in Figure 3b and Figure 3c for non-intervened oscillation and four coordinated ones, respectively. Please refer to the new manuscript for more details.

FIG. 3. The underlying mechanisms and energy consumptions of independent frequency and amplitude coordinations. **a** The optimal coordinator [the one with minimum energy consumption \mathcal{E} , see equation (10)] we designed for independent frequency and amplitude coordination follows the two rules. The four linear interactions a_{ij} (arrow: activation; I-shaped: inhibition) between two components (blue and red circles) are intervened by the coordinator f_{ij} . For both independent frequency and amplitude coordination, all the four interactions are either enhanced (thicker curves) or mitigated (thinner curves) simultaneously. Thus, the coordinator can be viewed as either an amplifier or a damper. **b** The non-intervened oscillation in the neuronal model equation (8) (blue: V ; red: W). All four configurations for independent frequency and amplitude coordination are tabulated below. Only the signs of the linear interactions and interventions are taken into account. If f_{ij} (or a_{ij}) is positive, then component j can be regarded as an activator to i , otherwise, an inhibitor. It can be observed that all the intervention coefficients follow the same sign as that of the corresponding endogenous interaction, if an increased frequency or amplitude is desired (the case of enhancement in **a**). On the contrary, for acquiring a decreased frequency or amplitude, the designed intervention exhibits as a reversible version of the endogenous one (the case of mitigation in **a**). **c** The four intervened examples for each case given in **b**. The energy consumption for each case is sketched by arrows on the right of the motif. A higher and darker arrow suggests a greater energy consumption.

7. Figure 3. Should precede Figure 2.

Thanks for the suggestion.

In the revised manuscript, we have placed the old Figure 3 before the old Figure 2.

8. Figure 4 vs. Figure 5: I do not understand why you don't keep the same order of exposition between the two figures (akin to Figure 6 w.r.t. Figure 4).

Thanks for the suggestion.

In the revised manuscript, we have rearranged the order of subfigures in Figure 5.

9. Figure 6. Why not consider plotting panel C in symlog scale or in a way that does not require zooming in but rather shows both regions on the left and the right sufficiently well? It would be very nice to extend this figure with graphs accounting for the possible configurations just tabulated in Figure 2.

Thanks for the reviewer's constructive comments and suggestions. Regarding the zoom-in figure, we have tried the symlog scale, but we were still not able to separate the two curves far away enough. Please see the figures given below. We have tried to use symlog scale in only horizontal axis (the first case), in only the vertical axis (the second case) and in both axes (the third case). To have a clearer presentation, we may need to design a new scale transformation. At this moment, we are sorry that we have no idea on how to do it in a proper and intuitive way. Therefore, we finally decided to keep the two subfigures as their original forms. If there is any better way that comes into your mind, we would greatly appreciate it. Regarding the second question, we agree that it would be better to extend Figure 6 and link it with the table in previous figure.

We have added, in the new manuscript, two more subfigures to account for the table in Figure 2b. With these two subfigures, the mechanisms behind independent frequency and amplitude coordination become more transparent and easier to understand.

curve. The policies applied as time goes on are highlighted by circles from left to right. **f, g** The optimal configurations for the four cases tabulated in Fig. 3b (the curves for $ij = 11$ and $ij = 22$ are not visually distinguishable). For an increased (resp., decreased) frequency or amplitude, we have $a_{ij} \cdot f_{ij} > 0$ (resp., < 0). **h, i** The frequency and amplitude variation (r_F and r_A) together with the average energy

neuronal model with zero flux at the boundary). Note that the configurations for independent frequency and amplitude coordinations (Fig. 6f, g) obey the mechanisms presented in Fig. 3a. For an oscillation moving in the hy-

10.A final remark concerns your Online Methods. Propositions (sometime, more appropriately, Lemmas) should be demonstrated accordingly. Conclusions are more properly Theorems or Corollaries, and once again should be properly proved. I could not find clear proofs of your statements, except for Corollary 3, neither in your Online Methods section, nor in your Supplementary Material.

Thanks for the constructive comments with which we agree. Regarding the propositions, we thought they state some properties of the system that we investigated. Therefore, we would like to keep it instead of replacing it with "Lemmas". According to the above comments and suggestions, we have made the following changes.

In the revised manuscript, we have replaced all the "Conclusion" with "Theorem". We have provided more information by writing the proofs of all conclusions and explaining the propositions with more details in the Supplementary Information. Also, we have made some changes accordingly in the main text (mainly in Methods) and Supplementary Information.

3 More details on Propositions

3.1 Proposition 1

Remember that we study the periodic oscillation arising from the Hopf bifurcation. Therefore, for the original system, there exists a critical value for the bifurcation parameter (i.e., $\epsilon = \epsilon^*$) where the Hopf bifurcation occurs. When we make an intervention F on the original system, we expect that the Hopf bifurcation remains the same up to its qualitative behavior. In other words, our intervention only alter the frequency and amplitude. Therefore, there exists the same critical value for the Hopf bifurcation in the intervened system. Otherwise, the comparison between original and coordinated periodic oscillations would be unfair. Finally, the intervened system should satisfy the condition that it possesses a critical value $\epsilon = \epsilon^*$ for the Hopf bifurcation.

3.2 Proposition 2

Only the most generic case (i.e., the generic Hopf bifurcation) is considered in this work, therefore, we need to avoid the complicated and degenerate (which is not likely to occur in real applications) case for the system. Consequently, at the critical bifurcation parameter, the system possesses only a pair of purely imaginary eigenvalues. Otherwise, more complicated case/bifurcation would arise. Moreover, to guarantee the stability of the investigated oscillation, the real part of any eigenvalue should not be positive. Any eigenvalue located in the right-half of the complex plane would lead an unstable periodic oscillation.

4 Proofs

4.1 Proof of Theorem 1

The Hopf bifurcation occurs when the characteristic equation has a pair of purely imaginary roots. Thus, according to Eq. (6), the critical value for the Hopf bifurcation is determined by the equation $c_1(k_i) = 0$. We then denote the nonintervention one and the coordinated one as c_1 and \hat{c}_1 , respectively. Now, we have $\hat{c}_1 = c_1 - f_{11} - f_{22}$. It is stated in Proposition 1 that the Hopf bifurcation we consider in this work always occurs at the same bifurcation parameter, therefore, we have $c_1 = \hat{c}_1 = 0$ for all real f_{11} and f_{22} . This equation is satisfied if and only if $f_{11} + f_{12} = 0$.

4.2 Proof of Theorem 2

According to Proposition 2, there are only two cases for the roots of Eq. (6) (in the main text) for each i . The equation possesses either two negative real roots or a pair of imaginary roots whose real part is non-positive (≤ 0). For the first case, we assume that the two real solutions are: $r_1 < 0$ and $r_2 < 0$. Then, the characteristic equation has the form $\lambda_{k_i}^2 - (r_1 + r_2)\lambda_{k_i} + r_1 r_2 = 0$. Consequently, we deduce that $c_0(k_i) = r_1 r_2 > 0$. For the second case, we assume that the real part and the imaginary part are $a \leq 0$ and $b \in \mathbb{R}$, respectively. Analogously, we deduce that $c_0(k_i) = a^2 + b^2 > 0$. Finally, we conclude that $c_0(k_i)$ is strictly positive for all i .

4.3 Proof of Theorem 3

Assume that there exists $i = i^* > 0$ such that Eq. (6) possesses a pair of pure imaginary roots. Then, we deduce that $c_1(k_{i^*}) = 0$. Since $c_1(k_i)$ is strictly increasing with respect to k_i (provided that $d_1 + d_2$ is positive), we must have $c_1(k_0) < 0$. Moreover, it follows from Theorem 2 that $c_0(k_0) > 0$. Therefore, when $i = 0$, one of the roots of Eq. (6) has positive real part, and, therefore, Proposition 2 is violated. Consequently, i must be 0. This completes the proof.

4.4 Proof of Theorem 4

The Hopf bifurcation occurs when the characteristic equation possesses a pair of purely imaginary roots, which yields the equation $c_1(k_i) = 0$ [see Eq. (6) in the main text]. According to Theorem 2, we have $i = 0$ for the stable oscillation. Consequently, the bifurcation parameter for the Hopf bifurcation is determined by the equation $c_1(k_0) = 0$.

Responses to reviewer #2

1. The author's figures and proofs are clear and intuitive. However, their sentences are often incomplete. Many sentences begin with conjunctions (such as "because" and "whereas") and should be joined with the previous sentence.

Thanks for the reviewer's comment.

In the revised manuscript, we have rephrased the inappropriate sentences and improved the expositions of the manuscript.

pendent of x (i.e., the spatial variable). In spite of this, the coordinator designed here still works for the entire space, **because** the spatially heterogeneous effect is eliminated during the computation (see Supplementary Sections 5 and 6 for more details) and it does not affect the final coordination.

Given a fixed fold change r_F , the computed feedback coefficients may not be unique, **because** the two algebraic equations are nonlinear. Some of the choices should be

tions. In Fig. 5d, it represents the fold change r_A of the amplitude, **whereas** in Fig. 4e, it is the reciprocal of the fold change of frequency (i.e., $1/r_F$). We then deduce

2. The legend for Figure 1b is confusing. They illustrate a network where a miRNA regulates a protein (by regulating its mRNA abundance), and that protein regulates both itself and the miRNA. They describe this as a "protein-mRNA interaction"; perhaps "protein-miRNA feedback" or "protein-miRNA circuitry" would be clearer.

Thanks for the reviewer's suggestion, with which we totally agree.

In the revised manuscript Figure 1b, we have changed "protein-mRNA interaction" to "protein-miRNA circuitry". We have also made some corrections (mRNA \rightarrow miRNA) in the main text and Supplementary Information accordingly.

deceleration) while the amplitude (intensity of the event) remains invariant. **b**, *Amplitude coordination*. The rhythmic biological processes also exist in **genome-scale transcriptional and translational processes such as protein-miRNA circuitry**. The protein concentration in a

3. In Figure 5g legend the authors incorrectly cite reference 9 instead of 59 (cancer network). They refer to protein-mRNA oscillation in the cancer network, but reference 59 describes a cancer network consisting of a miRNA cluster and proteins (E2F and MYC). The authors should clarify if the reported quantity represents this miRNA cluster.

Thanks for the reviewer's comment. We would like to clarify that the number 9 in parenthesis represents the equation number instead of a cited reference. All references are cited as superscripts in the main text. To avoid possible misunderstanding, we have made the following changes in the legend of Figure 5g.

We changed "equation (9)" to "[see equation (9) in Methods]".

The reported quantity in Figure 5g and 5h represents the protein level. We have emphasized this fact by making the following change.

We changed "Independent amplitude coordination for the protein-mRNA oscillation..." to "Independent amplitude coordination for the protein level of the protein-miRNA oscillation...". The time course for the quantity of the miRNA cluster is shown in the Supplementary Information.

arrows. The smaller one corresponds to $x = 0$ while the larger one to the dark circle. **g** Independent amplitude coordination for the **protein level of the protein-miRNA oscillation in the "cancer network" [see equation (9) in Methods]**. The amplitude is gradually and independently suppressed (the color becomes lighter) by varying the coordinator at each time label ($t = 200, 400, 600, 800$ and 1000). The blue area is the cancer zone with high probability of oncogenesis classified by the level of protein (> 2.75). The amplitude suppression prevents the oscillation from the entry into the cancer zone. For the corresponding time course of the **miRNA cluster**, see Supplementary Fig. 14. **h** The time course at $x^* = \pi/2$.

4. We note that the authors focus on two relatively simple endogenous biorhythm circuits. Could the authors comment on more complex oscillators that control important biorhythms (CLOCK/BMAL, COOLAIR/COLDAIR)? Can the authors predict whether their coordinator can be applied to such networks?

Thanks for the reviewer's questions. As we mentioned in "Discussion", our method can be applied to more complicated systems with more components. There are computational models describing the circadian rhythm regulated by BMAL1:CLOCK transcription factor and PER:CRY complex. For example, see the Kim-Forger model introduced in [JJ Tyson & B Novak, A dynamical paradigm for molecular cell biology, Trends in Cell Biology, 2020]. In this model, the cyclic oscillation also arises from the Hopf bifurcation. Therefore, it is possible to use our framework to study the frequency-amplitude coordination in that model.

In the 4th paragraph of "Discussion" section, we have mentioned the possibility to apply our framework to the above model.

manifold and the normal form theories. For instance, the Kim-Forger model describing the circadian rhythm regulated by BMAL1:CLOCK transcription factor and PER:CRY complex⁵³. In this model, the cyclic oscillation also arises from the Hopf bifurcation. More compo-

5. We recommend the authors make their Matlab and Maple code public upon publication. This would greatly facilitate application of their design process to other biorhythm circuits.

Thanks for the suggestion. We agree. If the manuscript is accepted and published, we will provide the Matlab and Maple codes for one case publicly. Other codes can be provided upon reasonable request.

6. Units of time are not specified in any figures or legends.

Thanks for the reviewer's comment. We are sorry that the time units are missing in the previous version.

In the revised manuscript, we specified the time unit for all the time courses corresponding to the FitzHugh-Nagumo model. Please see Figure 4a, 4f, 4g, 6a in the main text. For instance, see the figure provided below.

On the other hand, the "cancer network" studied in our work is a nondimensionalized computational model. Therefore, the time of this model does not possess a unit. For instance, in a very recent article published in Nature Communications [Glass et al. 2021], the authors studied nondimensionalized delay differential equations for biological network motifs. In that article, the time does not possess a unit.

Responses to reviewer #3

- For example, in Fig.2 they discuss two entities that are coupled with each other. Such bidirectional networks have been studied a lot in the past. The dynamic phenomena that can arise are very complex even for such a simple system that just have one self- and one cross reaction term for each entity. The authors now increase the complexity by doubling the number of interactions. Has their analysis really captured all possible complexity arising from that? How does that relate the specific neuron and cancer systems?

Thanks for the reviewer's comments and questions. First of all, we want to apologize that we have not made our arguments clear enough for the audiences. Some text and figures may mislead and confuse the readers. Please let us answer the questions and clarify our arguments one by one.

(i) The authors now increase the complexity by doubling the number of interactions.

We are sorry that the old Figure 2 misled what we actually desired to state. Though we showed 8 curves in Figure 2a for each motif, the actual number of interactions in the network motif are not increased. For a biochemical system with two components, there are at most four interactions: two self-interactions and two cross-interactions. All those interactions are determined (or interfered) either by the system parameters or by a set of interventions. Consequently, every interaction can be divided into two parts: an endogenous one and an intervention. For instance, a negative self-interaction in a biochemical computational model usually describes the degradation rate of the corresponding component. This rate is the intrinsic property for a given component in a specific system. But it is also possible to change this rate by an intervention. Therefore, when we consider one interaction, there are two ways to coordinate its intensity (enhancement or mitigation). Using our coordinator, we did not change the network motif or the number of interactions. We just perturb the intensity of the original interactions like a perturbation control. Moreover, this control may be implemented either internally or externally. As a metaphor, the internal intervening coordinator acts like the pedal of a car which accelerates or slows down the system, while the external interventions like some artificial controller steering the four wheels (four linear interactions) of the car. Consequently, the complexity of the system (in other words, the qualitative behaviour of the computational model) has not been increased. We only alter its quantitative behaviour leading to successful frequency and amplitude coordinations.

In order to make our statements clearer to understand, in the new manuscript, we have redrawn the old Figure 2 (please note that it is the Figure 3 now). In this new figure, we separate the 8 curves. As mentioned before, the linear interactions are divided into two parts: the endogenous one and our intervention. We think the new figure will be more suitable for explaining our purposes. We have also changed some text accordingly. Especially, we abandoned the word "exogenous feedback" because it left an impression that the number of interactions is increased. We have rewritten the first section of "Results" so that our ideas can be explained more clearly.

FIG. 3. The underlying mechanisms and energy consumptions of independent frequency and amplitude coordinations. **a** The optimal coordinator [the one with minimum energy consumption \mathcal{E} , see equation (10)] we designed for independent frequency and amplitude coordination follows the two rules. The four linear interactions a_{ij} (arrow: activation; I-shaped: inhibition) between two components (blue and red circles) are intervened by the coordinator f_{ij} . For both independent frequency and amplitude coordination, all the four interactions are either enhanced (thicker curves) or mitigated (thinner curves) simultaneously. Thus, the coordinator can be viewed as either an amplifier or a damper. **b** The non-intervened oscillation in the neuronal model equation (8) (blue: V ; red: W). All four configurations for independent frequency and amplitude coordination are tabulated below. Only the signs of the linear interactions and interventions are taken into account. If f_{ij} (or a_{ij}) is positive, then component j can be regarded as an activator to i , otherwise, an inhibitor. It can be observed that all the intervention coefficients follow the same sign as that of the corresponding endogenous interaction, if an increased frequency or amplitude is desired (the case of enhancement in **a**). On the contrary, for acquiring a decreased frequency or amplitude, the designed intervention exhibits as a reversible version of the endogenous one (the case of mitigation in **a**). **c** The four intervened examples for each case given in **b**. The energy consumption for each case is sketched by arrows on the right of the motif. A higher and darker arrow suggests a greater energy consumption.

Endogenous linear interactions and their interventions. We illustrate our computational framework for the frequency-amplitude coordinator using the two-component diffusive system which can always be modeled as (see Methods for a detailed description):

$$\underbrace{\frac{\partial \mathbf{u}(x,t)}{\partial t}}_{\text{Time evolution}} = \underbrace{\mathbf{D}(\epsilon) \frac{\partial^2 \mathbf{u}(x,t)}{\partial x^2}}_{\text{Spatial diffusion}} + \underbrace{\mathbf{A}(\epsilon) \mathbf{u}(x,t)}_{\text{Linear interaction}} + \underbrace{\mathbf{g}(\mathbf{u}(x,t), \epsilon)}_{\text{Nonlinear interaction}}. \quad (1)$$

Such a system possesses two self-interactions and two cross-interactions, all of which can be divided into two parts: the linear interactions $\mathbf{A}\mathbf{u}$ and the nonlinear ones \mathbf{g} . The matrix \mathbf{A} is associated with the Jacobian matrix and can be acquired in any computational model²¹. It represents the endogenous linear interactions, which consists of four coefficients a_{ij} ($i, j = 1, 2$) representing associated interaction (u_j to u_i). The sign of a_{ij} indicates the role played by the component j (activator or inhibitor), see Fig. 3a, b. The absolute values represent their intensities which are determined by system parameters, such as: degradation rate, synthesis rate, Michaelis constant, etc. These parameters are usually different from system to system (see Supplementary Section 9 for more details on the linear interactions).

For a given biochemical system, the endogenous linear interactions always exist in its computational model and have dominant impacts on the behavior of an oscillation near the quiescent state. Specifically, the intensities of these interactions determine the frequency and amplitude decisively. Therefore, our aim is to design a coordinator to intervene those interactions which further regulate frequency and amplitude precisely. Our coordinator is designed as:

(ii) *Has their analysis really captured all possible complexity arising from that?*

Thanks for the questions. We acknowledge that there are other complex dynamic phenomena can arise from the simple system, even if it possesses only two components. As our main purpose is to find a universal approach to achieve frequency and amplitude coordinations, we focus only on the cyclic oscillation in this work. Indeed, other complex phenomena such as homoclinic or heteroclinic orbits also exist in such simple computational models, they are beyond the scope of our study because they may divert attention from the main topic.

To study the cyclic oscillation, the Hopf bifurcation is possibly the most significant and intuitive starting point. Therefore, in our work, we pay our attentions to the oscillation arising from the Hopf bifurcation and conduct a systematic investigation on its frequency and amplitude coordinations. We do not intend to change the qualitative behaviour of the oscillation. To this end, we made a thorough stability analysis to avoid the occurrence of other complexities. As mentioned in the last question, our designed coordinator only changes the quantitative behaviour (frequency and amplitude) of the cyclic oscillation rather than the qualitative behaviour.

$$\begin{bmatrix} u_1 \rightarrow u_1 & u_2 \rightarrow u_1 \\ u_1 \rightarrow u_2 & u_2 \rightarrow u_2 \end{bmatrix} := \mathbf{F}\mathbf{u} = \begin{bmatrix} f_{11} & f_{12} \\ f_{21} & f_{22} \end{bmatrix} \begin{bmatrix} u_1 \\ u_2 \end{bmatrix}, \quad (2)$$

which has the same form as $\mathbf{A}\mathbf{u}$ because it is a perturbation of the endogenous linear interactions. The coordinator can be implemented either internally by varying the system parameters or externally as a feedback controller (e.g., the computer-based microfluidic devices^{34,35}). After implementing the coordinator, the resulting linear interactions become $(\mathbf{A} + \mathbf{F})\mathbf{u}$ (see Fig. 3a). Now, the key question is: How to determine the intensities of those interventions for desired frequency and amplitude? Later, we will show that they depend highly on the associated endogenous ones.

1

It follows from the above answer that we considered all the four interactions included in a system with two components. To achieve our aims, when designing a coordinator, all the interactions are allowed to be enhanced or mitigated. Also, we considered all the possible coordinating cases for a given oscillation including: increasing or decreasing its frequency or amplitude. Please let us know if we misunderstand your question.

(iii) How does that relate the specific neuron and cancer systems?

As stated before, in order to coordinate frequency and amplitude we only need to enhance or mitigate the endogenous interaction loops. In this work, we designed a universal computational framework to accomplish the task for the cyclic oscillation born from the Hopf bifurcation. For any biochemical system including the specific neuronal model and "cancer network" studied in our work, the computational model may be highly nonlinear and the four interactions can be very complicated. Nonetheless, these interactions can always be split into two parts: the linear one and the nonlinear one (please see equation (1) in the revised version). The former one is referred as endogenous linear interaction in our article which plays a dominant role for quantitative behaviour of the studied cyclic oscillation. For any biological model including the specific neuronal and cancer system, the strength of each linear interaction can be calculated by finding the Jacobian matrix, and their values are determined by the system parameters. Usually, it is highly dependent on the degradation rate, synthesis rate, concentration of enzyme, etc. (could be different from system to system), thus, it is able to be changed either internally or externally. Our previous manuscript may mislead what we actually desired to say. We did not double the interactions which seems hard to be related to the specific systems. Our intervention can be viewed as a perturbation of the original interaction, just like a perturbation control.

In the revised manuscript, we have emphasized that the endogenous interactions always exist and can be found in the computational model. For any biological system, it is related to the Jacobian matrix and may be different from system to system. In the first paragraphs in "Results" section, we linked the interaction matrix \mathbf{A} with the Jacobian matrix. Also, we have written one more section (Section 9) in Supplementary Information to illustrate it using a computational model of an RNA-protein negative-feedback loop. We hope that this section can make the endogenous linear interactions become more transparent.

Endogenous linear interactions and their interventions. We illustrate our computational framework for the frequency-amplitude coordinator using the two-component diffusive system which can always be modeled as (see Methods for a detailed description):

$$\underbrace{\frac{\partial \mathbf{u}(x, t)}{\partial t}}_{\text{Time evolution}} = \underbrace{\mathbf{D}(\epsilon) \frac{\partial^2 \mathbf{u}(x, t)}{\partial x^2}}_{\text{Spatial diffusion}} + \underbrace{\mathbf{A}(\epsilon) \mathbf{u}(x, t)}_{\text{Linear interaction}} + \underbrace{\mathbf{g}(\mathbf{u}(x, t), \epsilon)}_{\text{Nonlinear interaction}}. \quad (1)$$

Such a system possesses two self-interactions and two cross-interactions, all of which can be divided into two parts: the linear interactions $\mathbf{A}\mathbf{u}$ and the nonlinear ones \mathbf{g} . The matrix \mathbf{A} is associated with the Jacobian matrix and can be acquired in any computational model²¹. It

9 Endogenous linear interactions in a computational model

For a given biochemical oscillator, the endogenous linear interactions always exist, and they are extractable from the corresponding computational model. Moreover, the intensities of these linear interactions are determined (or interfered) by the system parameters, such as: the intensity of an upstream-regulating signal, synthesis rate, degradation rate, etc. Here, we illustrate the kinetic model of an RNA-protein negative-feedback loop as an example [see Ref. (6)]. Its reaction-diffusion version is written as:

$$\begin{aligned} \frac{\partial X(x, t)}{\partial t} &= d_x \frac{\partial^2 X(x, t)}{\partial x^2} + k_1 S \frac{K_d^p}{K_d^p + Y^p} - k_{dx} X, \\ \frac{\partial Y(x, t)}{\partial t} &= d_y \frac{\partial^2 Y(x, t)}{\partial x^2} + k_{sy} X - k_{dy} Y - k_2 E_T \frac{Y}{K_m + Y + K_I Y^2}. \end{aligned} \quad (S58)$$

In this model, X and Y represent the expression level of mRNA and protein, respectively. There are 13 parameters involved in this system whose physical meanings are listed in Table 1.

Table 1: The parameters and their physical meanings.

Parameter	Physical meaning	Parameter	Physical meaning
S	intensity of an upstream-regulating signal	k_1	synthesis rate of mRNA
p	integer indicating monomer, dimer, trimer, etc.	k_d	dissociation constant
k_{dx}	degradation rate of mRNA	k_{sy}	synthesis rate of protein
k_{dy}	(linear) degradation rate of protein	k_2	enzymatic degradation rate of protein
E_T	concentration of enzyme	K_m	Michaelis constant
K_I	inhibition constant of enzymatic degradation	d_x	diffusion constant of mRNA
d_y	diffusion constant of protein		

This generic model describes a simple oscillator involving one mRNA and its corresponding protein. The Hopf bifurcation and a periodic oscillation exist under certain parameter values. As mentioned in the main text, the linear interaction matrix \mathbf{A} is the Jacobian matrix at the equilibrium $(X, Y) = (X^*, Y^*)$, where $X^* = k_1 S K_d^p / [k_{dx}(K_d^p + Y^{*p})]$ and Y^* is the real root of the following equation

$$-k_{dy}Y^* + \frac{k_{sy}k_1 S k_d^p}{(k_d^p + Y^{*p})k_{dx}} + \frac{k_2 E_T Y^*}{K_m + Y^* + K_I Y^{*2}} = 0. \quad (\text{S59})$$

Then, by simple manipulation, the interaction matrix is found as

$$\mathbf{A} = \begin{bmatrix} a_{11} & a_{12} \\ a_{21} & a_{22} \end{bmatrix} = \begin{bmatrix} -k_{dx} & -\frac{k_1 S k_d^p p}{(k_d^p + Y^{*p})^2} Y^{*p-1} \\ k_{sy} & -k_{dy} - \frac{k_2 E_T}{K_m + Y^* + K_I Y^{*2}} + \frac{k_2 E_T Y^* (1 + 2K_I Y^*)}{(K_m + Y^* + K_I Y^{*2})^2} \end{bmatrix}. \quad (\text{S60})$$

Accordingly, the endogenous linear interactions are indeed determined by the system parameters. The relation between the linear interactions a_{ij} and system parameters are given in Table 2. If a small perturbation is added on system parameter, then the intensity of the associated linear interaction is also changed. For instance, a small variation made on the concentration of enzyme E_T yields the change of a_{22} . Such a variation can be regarded as an intervention which further affects the frequency and amplitude of the oscillation arising from the Hopf bifurcation. In Fig. 1, the diagram of the linear interactions at the Hopf bifurcation is depicted. The parameters are chosen following the values provided in Ref. (6).

Table 2: The linear interactions and their related system parameters.

Linear interaction	System parameters
a_{11}	k_{dx}
a_{12}	k_1, S, k_d, p, Y^*
a_{21}	k_{sy}
a_{22}	$k_{dy}, k_2, E_T, K_m, K_I, Y^*$

Figure 1: The linear interactions a_{ij} in the computational model Eq. (S58). The parameters are set as: $k_1 = k_{dx} = k_{dy} = 0.05$, $p = 4$, $K_m = 0.1$, $k_d = 1$, $K_I = 2$, $k_{sy} = 1$, $k_2 = 1$ and $E_T = 1$. Positive and negative interactions are indicated by arrows and I-shaped curves, respectively.

- As another example, the authors motivate the importance of coordinating frequency vs. amplitude – I would expect some phase space depiction analyzing both jointly – but I do not recognize that. Also - what determines amplitude and frequency? For example, frequency is typically determined by the relevant time scales in the systems, e.g., degradation rates or signaling delays; amplitudes are typically given by ratios of activation and degradation [for example, Alon 2006 Introduction System Biology, CRC Press, or Glass 2021 Delay Equations, Nature Communications]. None of such features are discussed here.

Thanks for the reviewer's comments and questions. We agree that analysing frequency and amplitude in a phase space jointly is a good idea.

In the revised manuscript, we included one more subfigure in Figure 6 (Fig. 6c) which shows the trajectory corresponding to the temporal profile in Figure 6a and 6b. In this subfigure, we can intuitively compare the amplitude from the size of the circle. The frequency is coded by different colour-scales, because, in a phase space, the frequency cannot be intuitively presented. We also provide one more movie to present more information on this figure. In the new movie, the differences of the trajectories in each time interval for both frequency and amplitude can be clearly observed.

Movie 6

This animation shows the phase space evolution of the time course in Fig. 6b. Six panels show the evolution in different time intervals. Obviously, due to the hybrid coordination, the amplitude (size of the circle) and the frequency are different in these panels.

Regarding the second question, we agree that the relevant time scales or delays influence the frequency and that the ratio of activation and degradation influence the amplitude. But in our work, we are more interested in the relation between the slight variations of the four linear interactions and the two quantities. These interactions are determined by system parameters (such as: degradation rate, synthesis rate, concentration of enzyme, etc.) and can be different from system to system. We desired to examine how our slight interventions precisely influence the frequency and amplitude.

In fact, for a cyclic oscillation in a computational model, any slight perturbation on the parameters varies the amplitude and frequency. As far as we know, there is no certain universal mechanisms or rules for these variations. Otherwise, the frequency and amplitude modulation would be an easy task. Of course, the time scales or the ratio can determine roughly the scales of frequency or amplitude. But they cannot determine an accurate value of frequency and amplitude if only the scales of the parameters are provided. To obtain a precise value of frequency and amplitude, a rigorous computation has to be performed. In fact, this is what we have done in our work. We cared about the relation between the variations of frequency/amplitude and the variations of the linear interactions.

From the two specific models considered in our manuscript, we have only found the possible mechanisms on the four interactions for frequency and amplitude coordination. The four interventions f_{ij} must follow either the enhancement rule or the mitigation rule (see Figure 3 in the revised manuscript). Moreover, we have discussed the amplitudes of the two components. The ratio of the two amplitudes is determined by the ratio of cross-interactions, please see Eq. (4). Except for these two findings, we did not find any other universal rules. Of course, there could be some features that we have not found in this work. But they need an exhaustive investigation on many other systems which is beyond the topic of our article.

In summary, in our work, it is not easy to answer the generic question: What determines amplitude and frequency? Because any parameters can determine the two quantities. Also, the underlying rules may be different from system to system. There is no guarantee for a universal rule especially for the precise coordinations studied in our work. We can only provide information on the possible findings we found in our investigations. If there are some hidden rules, we believe that our computational framework can be applied to find them. But this needs a future study.

Besides the discussion given in Figure 3 and Eq. (4), we apologize that we cannot provide more universal rules for frequency and amplitude coordination at this moment. In the "Discussion" section of the revised manuscript, we have raised these questions which can be studied in future.

cal oscillations. Of course, it is a challenge to address the questions: Is our discovery a universal principle? Is there any other mechanism behind frequency and amplitude coordinations? They are beyond the scope of the present article which could be important directions for future study. The proposed coordinating strategy can be

Responses to reviewer #4

First of all, we greatly appreciate all the comments and questions made by this reviewer. Before making point-by-point responses, we would like to emphasize that our work has its novelty and advantages compared with the existing work in the literature. Moreover, although it is a theory-oriented work and has not been verified with an experiment presently, we believe that it still has potential contributions for the future studies in the area of frequency and amplitude modulations, in which there is no solid experiment-proved work and only a few pioneering works exist.

To improve the progress in a specific area, the theoretical research is always a good and significant starting point. There are important researches in the area of frequency and amplitude modulations including the two famous ones: [TYC Tsai et al., *Science*, 2008] and [M Tomazou et al., *Cell Systems*, 2018]. Though they provided important findings and ideas, they are all based on mathematical modelling and only numerical simulations were involved. There was a 10-year gap between the two articles. Apparently, lacking of a rigorous theoretical work, the area of frequency and amplitude modulations progressed very slowly. In our opinions, to further promote the development of the area of frequency and amplitude modulations, a rigorous theory-based computational framework is needed. It is also in accordance with the perspectives given in recent articles [JJ Tyson & B Novak, *Trends in Cell Biology*, 2020] and [AJ Lopatkin & JJ Collins, *Nature Reviews Microbiology*, 2020], where the importance of dynamical systems tools including bifurcation theories for solving biological problems are emphasized. We got inspirations from these perspectives and integrated them into our work. We hope that the publication of our work can be a good opportunity to make both biologists and mathematicians pay their attentions to this area. Actually, we have already discussed with some experimentalists and biologists with our work and they showed their interests. We also hope that our work can promote the development of this area.

We understand that our work has not been verified experimentally at this moment. But the computational framework we designed and the mechanisms we found here still have their great contributions. In our opinions, these results can guide the biologists to design the experiments for frequency and amplitude coordinations. Also, it can be useful for synthetic biologists to design new biological models with flexible frequency and amplitude. If it was published in a specialized journal such as: control theory and nonlinear dynamics, very few biologists would notice it. As a consequence, its potential contribution in the area of frequency and amplitude modulations would be buried. Therefore, we would greatly appreciate it if the reviewer supports our work in its publication in an interdisciplinary journal like *Nature Communications*, which has broad audiences from diverse disciplines.

1. There is also a concern on novelty with respect to this paper

<https://www.sciencedirect.com/science/article/pii/S2405471218301108>

Which is cited lightly, but in my opinion quite directly comparable to the results on the protein-mRNA interaction model. I missed some more in depth discussions on how their results compare to that previous work.

Thank the reviewer for the comment. We totally agree that it is very significant to compare our approach with the existing one in the literature, especially, to make the novelty of our work clear to the audiences. We apologize that we have not made an in-depth discussion and detailed comparisons on the two works. In the revised manuscript, we have discussed the existing work and compared ours with it in both "Introduction" and "Discussion" sections. We have emphasized the differences between the two works as well as the advantages and disadvantages of our works compared with the existing work. Please see the detailed responses below.

The paper mentioned in this comment is entitled "Computational Re-design of Synthetic Genetic Oscillators for Independent Amplitude and Frequency Modulation", and it was published in *Cell Systems* 2018 (abbreviated as [CS18] hereafter).

Gene of interest G versus the main oscillator

The two works lay emphasize on different aspects. In [CS18], the authors are interested in controlling the frequency and amplitude of a gene G. According to their work and the figures therein, this gene is a non-autonomous gene, in the sense that, there is no bidirectional interactions between it and the main oscillator. It is only repressed by the main oscillator and does not have an impact back on the oscillator. Therefore, it is apparent that the frequency of the gene G must be the same as that of the main oscillator. That is also the reason why the two controllers "Dial 1" and "Dial 2" can decouple the frequency and amplitude. In such a way, the main oscillator can be somehow regarded as a controller for modulating the frequency and amplitude of the outside gene G.

In our work, we are interested in coordinating the frequency and amplitude of the oscillator itself. That is, there are two components (e.g., repressor and activator, or mRNA and protein) in one oscillator. We did not consider another gene of interest outside the oscillator. The components inside the oscillator have their own frequencies and amplitudes, which are also significant and need to be regulated precisely. Therefore, we want to design a coordinator to modulate their frequencies and amplitudes rather than the one outside the oscillator.

Metaphorically, [CS18] used a car (the main oscillator) to pull and control a motorcycle (the gene G), while our

work regulated the four wheels (linear interactions) of the car and controlled the car (the oscillator) itself. Consequently, the focuses of the two works are different. We are not saying which one is better. What we want to emphasize here is that, though the concerns are distinct, both works have their practical usefulness under different circumstances. If a non-autonomous gene (like gene G in [CS18]) needs to be modulated, the method proposed in [CS18] is obviously more suitable for this purpose. If one or more components inside the oscillator need to be coordinated, then we think our method is able to accomplish the task.

Network architecture versus intensities of linear interactions

Besides the difference in the focus of the two works, the two approaches are also different. In [CS18], the authors investigated the modifications of the network architecture. It was verified in their work that this re-designing approach can successfully decouple the frequency and amplitude “physically”. In our work, because we want to modulate the frequency and amplitude of the components inside the main oscillator, the re-designing approach seems not to be suitable for this purpose. Also, in our opinion, in addition to the network architecture, the intensities of the linear interactions inside the oscillator are also significant for determining the frequency and amplitude. Therefore, we want to see how their variations affect the two quantities and also see whether they can be leveraged to accomplish our task. Thus, we pay our attentions to those values instead of modifying the motif. We apologize that we have not presented our idea and method clearly in the previous manuscript. Especially, we misused the word “exogenous feedback”. It left an impression that our designed coordinator can be only implemented as a feedback controller which seems to complicate the original oscillator. In fact, our designed coordinator can also be regarded as a small perturbation made on the linear interactions. Therefore, it can be regarded as a perturbative intervention as well. As a metaphor, our designed coordinator can be viewed as the pedals of a car, which accelerate or slow down the system. In such a case, the oscillator is intervened internally. Therefore, our framework may guide the synthetic biologists and provide significant information for designing flexible biological models. Moreover, our coordinator can also be regarded as an external controller steering the four wheels (linear interactions) of a car. In this case, the computational framework can afford a feasible and precise way for the biologists to realize frequency and amplitude modulations in traditional biochemical model with some artificial equipment such as the computer-based microfluidic device.

With the inspirations learnt from control strategies in engineering, we think that, accomplishing a control task using resources as minimum as possible is sometimes the basic requirement. For instance, in some circumstances of vehicle control, using another car to control the key target is not acceptable. Analogously, it may not be the best choice to control the frequency and amplitude of a gene using an additional oscillator. In other words, for a given two-component oscillator (mRNA-protein interaction), can we modulate their frequencies and amplitudes without using another oscillator? That becomes the reason why we pay our attention to the oscillator itself and on the intensities of its endogenous interactions.

Again, we are not trying to say which method is better. The two methods can achieve different tasks under distinct requirements. The one proposed in [CS18] is more appropriate for controlling a non-autonomous gene because the motif of the main oscillator is not restricted, while our approach can be used to modulate the frequency and amplitude of the components inside the oscillator. Moreover, our work aims at the precise relation between the four intensities and the frequency and amplitude.

Advantages and disadvantages

As mentioned before, the concern and approach in our work are different from those in [CS18]. Under distinct circumstances, our method may have advantages and disadvantages. On the one hand, our method is able to control the frequency and amplitude of both components inside the main oscillator without changing its motif. Moreover, it is a rigorous computational method. Therefore, we can achieve precise frequency and amplitude coordinations, either independently or concurrently, while the method proposed in [CS18] needs numerical simulation to decide the values of the “dials” when an accurate frequency or amplitude is required. Also, the orthogonality of frequency and amplitude modulation may not be fully guaranteed in [CS18]. On the other hand, the method in [CS18] seems to be more easily to be implemented practically. To apply our computational framework in practice such as using the microfluidic devices, a careful future study must be needed. Also, to modulate a non-autonomous gene, it is obvious that the method in [CS18] have a better performance.

Briefly, in our opinion, both approaches have their practical usefulness in distinct circumstances. The method in [CS18] is more appropriate when the non-autonomous gene is considered and the network architecture is allowed to be modified. As a comparison, our method is suitable for modulating the frequency and amplitude of the main oscillator itself, and it can also provide precise coordination if the accuracy is significant for modulations. In addition, our discovering revealed the possible mechanisms behind frequency and amplitude modulations providing significant information for synthetic biologists to design flexible biological models.

We apologize again that we have not made an in-depth and detailed discussion on [CS18] in the previous manuscript. In the revised version, we have made the following changes to emphasize this work as well as the differences between it and ours:

In the introduction section, we have rewritten several paragraphs so that the work [CS18] are discussed a bit more. Moreover, we have clarified the differences between our method and those in prior works.

For biological models, the dynamics depends highly on their network motifs (pattern of positive/negative interactions) and significantly attract plenty of attentions^{21–23}. Therefore, investigating the mechanisms between the motif and frequency-amplitude coordination becomes a natural direction. Auto-regulations via endogenous interactions were ubiquitously studied^{24–28}. Prior works also suggested that the tunability of frequency and amplitude can be enhanced by positive feedback loops, while the independent coordinations may not be achieved if only negative feedback loops appear^{29–31}. Besides, a very recent work³² studied the dual-feedback oscillator and the repressilator and the modifications of their network architectures. The frequency and amplitude of a gene outside the main oscillator are “physically” decoupled by re-designing the oscillator, thus, they can be modulated independently in extended ranges.

Actually, the regulations of frequency and amplitude of the components inside an oscillator are also significant. Moreover, besides the network topology, the intensities of endogenous interactions affect frequency and amplitude as well, but there are few related works³³. The two quantities can be regulated by making a slight intervention on these intensities. It can be realized either internally by varying the system parameters (e.g., degradation rate, synthesis rate, strength of a stimulus, etc.), or externally by a feedback controller (e.g., computer-based microfluidic devices^{34,35}). More importantly, the underlying mechanisms for coordinations as well as their energy consumptions also depend highly on the intensities, which have not yet been uncovered. In this article, we present a universal computational framework for precisely designing such an intervention coordinator. We will try to answer the questions: What specific form of an intervention should be made (i.e., enhancing or mitigating the original interactions)? How is its intensity, and energy consumption? Additionally, with our well-designed coordinator, both frequency and amplitude can be regulated precisely and concurrently (Fig. 1c).

In the discussion section, we have written one more paragraph to compare the two so that both the novelty of our work will be clearer to the audiences. We are sorry that we cannot compare the two works as detail as our responses because of the word-limitation of the manuscript.

Compared with the recent work focusing on the re-design of specific oscillators³², our approach pays attention to the oscillator itself and the intensities of its endogenous linear interactions. On the one hand, we focus on coordinating the frequency and amplitude of the components inside the oscillator rather than using a main oscillator to control an outside gene³². Therefore, the two works have practical usefulness under distinct circumstances. If the main oscillator is more important and needs to be modulated, then our framework is appropriate to accomplish the task. On the other hand, our designed coordinator is able to precisely acquire desirable frequency and amplitude. It provides the possi-

bility of precise coordinations. Though it has not been verified experimentally in this work, it may be realized in future by a well-designed computer-based microfluidic device. For various biological systems, such as the neuronal model and the gene regulatory network considered in this work, the endogenous interactions are always extractable, if its corresponding computational model is given [or discovered via machine learning from data^{63,64}]. Therefore, the proposed strategy can be applied to those systems straightforwardly. This work serves as a necessary foundation for precise biorhythm regulation. It may fill the gap between experimental data and theoretical coordinating strategies making a potential contribution for precision medicine.

Finally, to emphasize that we focus on the oscillator and its components’ frequency and amplitude, we have changed one word in the title of our work.

A frequency-amplitude coordinator and its optimal energy consumption for biological oscillators

2. It is shown that the feedback controllers can indeed achieve the goal, but there is no guarantee that any such controllers can be implemented by a real biochemical mechanism. There is plenty of recent literature on equivalences between PID controllers and biochemical implementations (particularly in Cell Systems journal), but the natural implementation of a general linear feedback controller as obtained in this work is in general not possible.

Thanks for the reviewer’s comments. We agree that it may be difficult to implement naturally our designed coordinator as a general linear feedback controller because of the restricted techniques for real biochemical systems presently. But it may be realized using artificial devices. For instance, in the future, our designed coordinator may be implemented using computer-based microfluidic devices, which was recently used to control the toggle-switch system in *E. coli* [J.-B. Lugagne et al., Nature Communications 2017] and to control the synchronization of cell cycle in budding yeast [G. Perrino et al., Nature Communications 2021]. The designed computational framework is able to be written into a microfluidic chip. Together with other experimental equipment, a closed-feedback loop control device can be formed. Besides, for modulating frequency and amplitude in neuroscience, the so-called Deep Brain Stimulation (DBS) [RM Pluta et al., JAMA, 2011] technique

may be considered. It implants electrodes into the brain and send signals to intervene the abnormal signals and also affect cellular and chemical behaviours in the brain. It has already been used to treat psychiatric disorders including Parkinson disease. Moreover, closed-loop controls using DBS have also been designed [B Rosin et al., Neuron, 2011; OV Popovych et al., Scientific Reports, 2016]. We apologize that, presently, we are not able to perform a solid experiment to justify our computational method due to the restriction of devices. But we are preparing for such an experiment and have had detailed discussions with experimentalists. We believe that our computational framework can be applied experimentally in the near future.

On the other hand, as mentioned before, our control policy can be regarded not only as an external feedback controller but also an amendment on the endogenous linear interactions (refer to new Figure 3). That is also the reason that we call the control policy a “coordinator” instead of a “controller”. As mentioned in previous responses, we pay our attention to the intensities of the linear interactions. For any biological model, these endogenous interactions a_{ij} always exist whose intensities are determined by several parameters, such as: degradation rate, synthesis rate, concentration of enzyme, etc. Therefore, we may also achieve our coordination policy by varying the system parameters as a perturbation. Our designed intervention f_{ij} is only a slight variation on the internal one and the outcome is $a_{ij} + f_{ij}$. For instance, a degradation rate can be interpreted as the strength of a negative self-interaction, thus, we can perturb this interaction by altering this rate. Our computation shows that the endogenous linear interactions dominate the quantitative behaviour of the periodic oscillation and a slight intervention is sufficient to coordinate its frequency and amplitude. When studying the biological systems, their motifs always attract plenty of attention. However, the intensity of the interaction is sometimes ignored though it is as significant as the network architecture, especially for the quantitative information, like frequency and amplitude. Therefore, we think our study will be helpful in the area of frequency and amplitude modulation, akin to the pioneering study in [CS18].

We apologize that we have not stated our control policy clearly in previous manuscript. Especially, some text mentioning the exogenous feedback and the old Figure 2 where each motif possesses 8 interactions may confuse the audiences and left an undesirable impression. In the revised manuscript, we have redrawn this figure (it becomes Figure 3 in the new manuscript) to emphasize that our control policy is an intervention on the endogenous interactions which can be added either externally or internally as a slight perturbation.

FIG. 3. The underlying mechanisms and energy consumptions of independent frequency and amplitude coordinations. **a** The optimal coordinator [the one with minimum energy consumption \mathcal{E} , see equation (10)] we designed for independent frequency and amplitude coordination follows the two rules. The four linear interactions a_{ij} (arrow: activation; I-shaped: inhibition) between two components (blue and red circles) are intervened by the coordinator f_{ij} . For both independent frequency and amplitude coordination, all the four interactions are either enhanced (thicker curves) or mitigated (thinner curves) simultaneously. Thus, the coordinator can be viewed as either an amplifier or a damper. **b** The non-interventive oscillation in the neuronal model equation (8) (blue: V ; red: W). All four configurations for independent frequency and amplitude coordination are tabulated below. Only the signs of the linear interactions and interventions are taken into account. If f_{ij} (or a_{ij}) is positive, then component j can be regarded as an activator to i , otherwise, an inhibitor. It can be observed that all the intervention coefficients follow the same sign as that of the corresponding endogenous interaction, if an increased frequency or amplitude is desired (the case of enhancement in **a**). On the contrary, for acquiring a decreased frequency or amplitude, the designed intervention exhibits a reversible version of the endogenous one (the case of mitigation in **a**). **c** The four intervened examples for each case given in **b**. The energy consumption for each case is sketched by arrows on the right of the motif. A higher and darker arrow suggests a greater energy consumption.

Also, we have changed some text accordingly. Especially, we have abandoned the word "exogenous feedbacks" which may mislead the audiences. In addition, we have rewritten the Introduction section (mainly in the first several paragraphs) and rewritten the first and second sections in "Results" to rephrase our ideas so that it is more understandable. We have emphasized that our method pays attention to the intensities of the linear interactions and their interventions.

Results

Endogenous linear interactions and their interventions. We illustrate our computational framework for the frequency-amplitude coordinator using the two-component diffusive system which can always be modeled as (see Methods for a detailed description):

$$\underbrace{\frac{\partial \mathbf{u}(x, t)}{\partial t}}_{\text{Time evolution}} = \underbrace{D(\epsilon) \frac{\partial^2 \mathbf{u}(x, t)}{\partial x^2}}_{\text{Spatial diffusion}} + \underbrace{\mathbf{A}(\epsilon) \mathbf{u}(x, t)}_{\text{Linear interaction}} + \underbrace{g(\mathbf{u}(x, t), \epsilon)}_{\text{Nonlinear interaction}}. \quad (1)$$

Such a system possesses two self-interactions and two cross-interactions, all of which can be divided into two parts: the linear interactions $\mathbf{A}\mathbf{u}$ and the nonlinear ones g . The matrix \mathbf{A} is associated with the Jacobian matrix and can be acquired in any computational model²¹. It represents the endogenous linear interactions, which consists of four coefficients a_{ij} ($i, j = 1, 2$) representing associated interaction (u_j to u_i). The sign of a_{ij} indicates the role played by the component j (activator or inhibitor), see Fig. 3a, b. The absolute values represent their intensities which are determined by system parameters, such as: degradation rate, synthesis rate, Michaelis constant, etc. These parameters are usually different from system to system (see Supplementary Section 9 for more details on the linear interactions).

For a given biochemical system, the endogenous linear interactions always exist in its computational model and have dominant impacts on the behavior of an oscillation near the quiescent state. Specifically, the intensities of these interactions determine the frequency and amplitude decisively. Therefore, our aim is to design a coordinator to intervene those interactions which further regulate frequency and amplitude precisely. Our coordinator is designed as:

$$\begin{bmatrix} u_1 \rightarrow u_1 & u_2 \rightarrow u_1 \\ u_1 \rightarrow u_2 & u_2 \rightarrow u_2 \end{bmatrix} := \mathbf{F}\mathbf{u} = \begin{bmatrix} f_{11} & f_{12} \\ f_{21} & f_{22} \end{bmatrix} \begin{bmatrix} u_1 \\ u_2 \end{bmatrix}, \quad (2)$$

which has the same form as $\mathbf{A}\mathbf{u}$ because it is a perturbation of the endogenous linear interactions. The coordinator can be implemented either internally by varying the system parameters or externally as a feedback controller (e.g., the computer-based microfluidic devices^{34,35}). After implementing the coordinator, the resulting linear interactions become $(\mathbf{A} + \mathbf{F})\mathbf{u}$ (see Fig. 3a). Now, the key question is: How to determine the intensities of those interventions for desired frequency and amplitude? Later,

we will show that they depend highly on the associated endogenous ones.

A universal design policy for the coordinator. To design a feasible coordinator, we need to know how its intensity relates to the frequency and amplitude. Here, we apply the center manifold and the normal form theories to accomplish the task (Fig. 2). Our framework includes following steps.

We first perform a persistence analysis (see Methods) before moving into detailed computations. It guarantees the existence of the oscillation after intervention.

This gives us Theorem 1 which provides the first criteria ($f_{22} = -f_{11}$) for designing the coordinator. It indicates that the two self-interactions of a feasible coordinator must be opposite to one another. Thus, a negative feedback loop is established providing the potential to produce rhythmic oscillation in a biological system²⁹. Reasoning that it ensures the occurrence of the Hopf bifurcation leading to a cyclic oscillation. In practical applications, a stable oscillation is of great significance and interest.

The second step is to link the intensities f_{ij} in a coordinator with the frequency and amplitude. Therefore, we exploit the normal form of the Hopf bifurcation⁵⁷ utilizing the center manifold and the normal form theories. For a periodic oscillation, the Poincaré normal form is always found as (see Methods for more details)

$$\dot{w} = \lambda w + \eta w^2 \bar{w} + \mathcal{O}(|w|^4), \quad w \in \mathbb{C}, \quad (3)$$

where $\lambda = \mu + i\omega$ is the complex eigenvalue corresponding to the Hopf bifurcation, and η is the normal form coefficient whose real part, denoted by χ , is the first Lyapunov coefficient. It is the simplest form preserving the information, and is usually applied to study qualitative behaviors. Here, we extract the quantitative information (frequency and amplitude) from the normal form. It serves as a bridge from the original oscillation to the coordinated one. Therefore, those information are helpful for designing the coordinator. The quantifying procedure is illustrated in Fig. 2.

The third step is to modulate both components consistently. We notice that they always share the same frequency, whereas their amplitudes could be distinct. As a consequence, the variations made by the coordinator may be inconsistent, in the sense that, the amplitude of u_1 (e.g., protein concentration) is increased or decreased while that of u_2 (e.g., miRNA concentration) is invariant, or vice versa. Such a result may be unexpected for a given biological system. To avoid this circumstance, we analyze both components when computing the normal form (see Methods). We surprisingly find that the modulations on both amplitudes are commensurate with one another if we have:

$$\frac{f_{21}}{f_{12}} = \frac{a_{21}}{a_{12}}. \quad (4)$$

This becomes the second criteria for the design policy. Reasonably, it incorporates the endogenous cross-interactions and their interventions. The ratio of two amplitudes is affected by the cross-intensities. To keep this ratio unvarying, the intensities of the interventions must also follow a fixed ratio determined by the intrinsic ones (i.e., a_{21}/a_{12}).

In the final step, we will derive two algebraic equations for the coordinator. From equation (3), the frequency and amplitude are extracted and approximated as $\omega/(2\pi)$ and $2\sqrt{-\mu/\chi}$, respectively. Note that μ is independent of the intervention coefficients, whereas ω and χ are expressed in terms of f_{11} and f_{12} , i.e., $\omega = \omega(f_{11}, f_{12})$ and $\chi = \chi(f_{11}, f_{12})$. Before we go any further, it is worth pointing out that the approximated amplitude is independent of x (i.e., the spatial variable). In spite of this, the coordinator designed here still works for the entire space, because the spatially heterogeneous effect is eliminated during the computation (see Supplementary Sections 5 and 6 for more details) and it does not affect the final coordination.

To make a clear illustration on the relation between endogenous linear interaction and system parameters, we have written one more section (Section 9) in Supplementary Information. In this new section, we use a generic computational model to explain how the endogenous linear interactions can be found and how they are determined by the system parameters.

9 Endogenous linear interactions in a computational model

For a given biochemical oscillator, the endogenous linear interactions always exist, and they are extractable from the corresponding computational model. Moreover, the intensities of these linear interactions are determined (or interfered) by the system parameters, such as: the intensity of an upstream-regulating signal, synthesis rate, degradation rate, etc. Here, we illustrate the kinetic model of an RNA-protein negative-feedback loop as an example [see Ref. (6)]. Its reaction-diffusion version is written as:

$$\begin{aligned}\frac{\partial X(x, t)}{\partial t} &= d_x \frac{\partial^2 X(x, t)}{\partial x^2} + k_1 S \frac{K_d^p}{K_d^p + Y^p} - k_{dx} X, \\ \frac{\partial Y(x, t)}{\partial t} &= d_y \frac{\partial^2 Y(x, t)}{\partial x^2} + k_{sy} X - k_{dy} Y - k_2 E_T \frac{Y}{K_m + Y + K_I Y^2}.\end{aligned}\quad (S58)$$

In this model, X and Y represent the expression level of mRNA and protein, respectively. There are 13 parameters involved in this system whose physical meanings are listed in Table 1.

Table 1: The parameters and their physical meanings.

Parameter	Physical meaning	Parameter	Physical meaning
S	intensity of an upstream-regulating signal	k_1	synthesis rate of mRNA
p	integer indicating monomer, dimer, trimer, etc.	k_d	dissociation constant
k_{dx}	degradation rate of mRNA	k_{sy}	synthesis rate of protein
k_{dy}	(linear) degradation rate of protein	k_2	enzymatic degradation rate of protein
E_T	concentration of enzyme	K_m	Michaelis constant
K_I	inhibition constant of enzymatic degradation	d_x	diffusion constant of mRNA
d_y	diffusion constant of protein		

This generic model describes a simple oscillator involving one mRNA and its corresponding protein. The Hopf bifurcation and a periodic oscillation exist under certain parameter values. As mentioned in the main text, the linear interaction matrix \mathbf{A} is the Jacobian matrix at the equilibrium $(X, Y) = (X^*, Y^*)$, where $X^* = k_1 S K_d^p / [k_{dx} (K_d^p + Y^{*p})]$ and Y^* is the real root of the following equation

$$-k_{dy} Y^* + \frac{k_{sy} k_1 S k_d^p}{(k_d^p + Y^{*p}) k_{dx}} + \frac{k_2 E_T Y^*}{K_m + Y^* + K_I Y^{*2}} = 0. \quad (S59)$$

Then, by simple manipulation, the interaction matrix is found as

$$\mathbf{A} = \begin{bmatrix} a_{11} & a_{12} \\ a_{21} & a_{22} \end{bmatrix} = \begin{bmatrix} -k_{dx} & -\frac{k_1 S k_d^p}{(k_d^p + Y^{*p})^2} Y^{*p-1} \\ k_{sy} & -k_{dy} - \frac{k_2 E_T}{K_m + Y^* + K_I Y^{*2}} + \frac{k_2 E_T Y^* (1 + 2K_I Y^*)}{(K_m + Y^* + K_I Y^{*2})^2} \end{bmatrix}. \quad (\text{S60})$$

Accordingly, the endogenous linear interactions are indeed determined by the system parameters. The relation between the linear interactions a_{ij} and system parameters are given in Table 2. If a small perturbation is added on system parameter, then the intensity of the associated linear interaction is also changed. For instance, a small variation made on the concentration of enzyme E_T yields the change of a_{22} . Such a variation can be regarded as an intervention which further affects the frequency and amplitude of the oscillation arising from the Hopf bifurcation. In Fig. 1, the diagram of the linear interactions at the Hopf bifurcation is depicted. The parameters are chosen following the values provided in Ref. (6).

Table 2: The linear interactions and their related system parameters.

Linear interaction	System parameters
a_{11}	k_{dx}
a_{12}	k_1, S, k_d, p, Y^*
a_{21}	k_{sy}
a_{22}	$k_{dy}, k_2, E_T, K_m, K_I, Y^*$

Figure 1: The linear interactions a_{ij} in the computational model Eq. (S58). The parameters are set as: $k_1 = k_{dx} = k_{dy} = 0.05$, $p = 4$, $K_m = 0.1$, $k_d = 1$, $K_I = 2$, $k_{sy} = 1$, $k_2 = 1$ and $E_T = 1$. Positive and negative interactions are indicated by arrows and I-shaped curves, respectively.

- The authors show the utility of this approach in two model biological systems. But because of the impossibility of finding biological circuits that implement the controller in both case studies, the examples somewhat diminish the extent of the theoretical contribution.

Thanks for the reviewer's comments. We agree that our theoretical contribution is diminished to some extent because it has not been proved by a solid experiment. Also, due to the unclear explanation of the approach given in the previous manuscript, the biological circuits seem to be complicated by our coordinator (the number of interactions was increased from four to eight). We are sorry to leave such an impression to the audiences. As mentioned before, our approach focuses on the intensities of the linear interactions in an unchanged motif. We also investigated their relation with the frequency and amplitude. Indeed, if we apply our designed coordinator as an external controller, it may be difficult to realize it currently. But we may realize it in future by computer-based microfluidic devices. Moreover, we can also regard the designed coordinator as a slight variation on the endogenous interaction loop which can be realized by varying the system parameters. Although our work is just a theory-oriented one, we think that it has the potential contribution for understanding the biological models, especially in the area of quantitative biology. We provide the important information on the relation between the intensities of linear interactions and the frequency and amplitude of a periodic oscillation. In our opinion, it will be useful for future studies on frequency and amplitude modulation, both theoretically and experimentally.

In the revised manuscript, we have emphasized that the designed coordinator is an intervention on the linear interactions. It can be implemented either internally or externally. Also, for a possible future application of our framework, we have cited two more references [34], [35].

Actually, the intensities of interactions in a motif are also significant to frequency and amplitude coordinations, but there are few related works³³. The two quantities can be regulated by making a slight intervention on these intensities. It can be realized either internally by varying the system parameters (e.g., degradation rate, synthesis rate, strength of a stimulus, etc.) or externally by a feedback controller (e.g., computer-based microfluidic devices^{34,35}). More importantly, the underlying mechanisms for coordinations as well as their energy consumptions also depend highly on the intensities, which have not yet been uncovered. In this article, we present a universal computational framework for precisely designing such an intervention coordinator. We will try to answer the questions: What specific form of an intervention should be made (i.e., enhancing or mitigating the original interactions)? How is its intensity and energy consumption? Additionally, with our well-designed coordinator, both frequency and amplitude can be regulated precisely and concurrently (Fig. 1c).

For a given biochemical system, the endogenous linear interactions always exist in its computational model and have dominant impacts on the behavior of an oscillation near the quiescent state. Specifically, the intensities of these interactions determine the frequency and amplitude decisively. Therefore, our aim is to design a coordinator to intervene those interactions which further regulate frequency and amplitude precisely. Our coordinator is designed as:

$$\begin{bmatrix} u_1 \rightarrow u_1 & u_2 \rightarrow u_1 \\ u_1 \rightarrow u_2 & u_2 \rightarrow u_2 \end{bmatrix} := \mathbf{F}\mathbf{u} = \begin{bmatrix} f_{11} & f_{12} \\ f_{21} & f_{22} \end{bmatrix} \begin{bmatrix} u_1 \\ u_2 \end{bmatrix}, \quad (2)$$

which has the same form as $\mathbf{A}\mathbf{u}$ because it is a perturbation of the endogenous linear interactions. The coordinator can be implemented either internally by varying the system parameters or externally as a feedback controller (e.g., the computer-based microfluidic devices^{34,35}). After implementing the coordinator, the resulting linear interactions become $(\mathbf{A} + \mathbf{F})\mathbf{u}$ (see Fig. 3a). Now, the key question is: How to determine the intensities of those interventions for desired frequency and amplitude? Later, we will show that they depend highly on the associated endogenous ones.

[34] Lugagne, J.-B. et al. Balancing a genetic toggle switch by real-time feedback control and periodic forcing. *Nat. Commun.* **8**, 1671 (2017).

[35] Perrino, G. et al. Automatic synchronisation of the cell cycle in budding yeast through closed-loop feedback control. *Nat. Commun.* **12**, 2452 (2021).

4. The paper touches on energy considerations for each controller, but I found these results quite disconnected from the way energy consumption is defined in neural models and the protein-mRNA model.

Thanks for the comment. We agree that the energy consumption defined in our work is quite different from the traditional way in neural models. We apologize that we have not explained clearly the reason why we use L2-norm as the energy index. Usually, for a given biological system, the total energy consumption is considered and defined. In our work, we focused on a control (or intervention) policy to make a little shift on the original oscillation. Therefore, we are more interested in the efficiency of the designed policy instead of the energy consumed by the entire system. Consequently, the energy we defined can be viewed as a small part of the entire system.

When defining the energy consumed by the coordinator, we have searched similar definition in the literature. We found that the L2-norm is the most common way to represent such an energy consumption. When studying biochemical systems, some conventional definitions from physics or engineering are usually applied. Specifically, the periodic oscillation studied in our work can be regarded as a signal in both neuronal and protein-miRNA models. Therefore, it is reasonable to use L2-norm to represent its energy. Moreover, to assess the efficiency of a controller (the coordinator in our work), the L2-norm is still used in the literature. For instance, in a recent Science article [A. Li et al., The fundamental advantages of temporal networks, *Science*, 2017], the authors also used L2-norm to define the energy consumption of the control policy and they also considered a protein-protein interaction network.

Indeed, for each specific system, there could be a better way to define the energy consumption, especially for the energy consumed by the entire system. But we think that, as far as we have searched, the L2-norm is the most appropriate definition for our universal framework, because we desire to define the energy consumption that can be applied in the generic case.

We have made some changes in the revised manuscript to clarify the reason that we choose L_2 -norm and to link it closely with biorhythm. We have rewritten the second sentence in section "The optimal energy consumption of a coordinator" and included a new reference [61]. Also, we added one more paragraph and more references [61], [70], [71] in Methods, section "The index for assessing energy consumption", to clarify the reason.

The optimal energy consumption of a coordinator. In practice, the consumption of a coordinating policy for sustaining an oscillation is of great significance. Therefore, an index \mathcal{E} is always introduced to measure the performance of a specific control policy⁶¹, see Methods for the one used in our problem. Fig. 4e shows the

The index for assessing energy consumption. We evaluate the average L_2 -norm of the designed coordinator over one period, which yields the following index

$$\mathcal{E} = \frac{1}{T} \int_0^T \|\mathbf{F}\mathbf{u}\| dt, \quad (10)$$

where T is the period of oscillation and $\|\cdot\|$ is the L_2 -norm induced by the inner product [Supplementary equation (S4)].

Note that the index \mathcal{E} evaluates only the energy consumed by the coordinator rather than the entire system, because we are more interested in the added interventions. There could be other definitions for assessing the energy consumption, especially for a specific model. Here, we use the L_2 -norm because it is a generic definition for the assessment of a control policy^{61,70}. Thus, it is more appropriate for our universal framework. More importantly, the biorhythm can be regarded as a cyclic signal for information processing (such as: neuron-neuron or cell-cell) whose energy is conventionally defined as the integral of L_2 -norm⁷¹. Therefore, our definition can be computed in practice once the signal of a biorhythm (e.g., action-potential or expression level) is observed.

[61] Li, A., Cornelius, S. P., Liu, Y.-Y., Wang, L. & Barabási, A.-L. The fundamental advantages of temporal networks. *Science* **358**,

[70] Lewis, F. L. & Syrmos, V. L. *Optimal Control*, Second Edn. (Wiley, New York, 1995).

[71] Grami, A. *Introduction to Digital Communications*. (Academic Press, 2016).

5. I do not wish to discourage the authors in the value of their theoretical contribution, but given the considerations above I believe the paper is better suited for a control theory or nonlinear dynamics journal.

Thank the reviewer for his/her support in our theoretical contribution. Though it is suited for a control theory or nonlinear dynamics journal, we think that it is more appropriate for a publication in an interdisciplinary journal with broad audiences from diverse disciplines, like Nature Communications. Based on above responses, we think our work have potential contributions to the future studies in the area of frequency and amplitude modulations. Although it is the theoretical work, it can guide or inform the experimentalists and biologists to pay their attention to this topic and on our computational framework. We also hope that this work can bring the biologists and mathematicians closer and promote the progress in this area.

Both experiment-based and theory-based research or their combination can make progress in an area. However, for the area of frequency and amplitude modulations, there is no solid experimental work as far as we know, even for a system with only two components. Therefore, we believe that our theoretical work is significant for the development of this area. For a problem being difficult to conduct an experiment, a theoretical work is always a good start. For instance, in Nature Communications, there are plenty of theoretical works using mathematical tools without an experiment to solve rigorously the control strategy, biological problem, etc. (We list below some very recent ones.) Thus, we think that the publication of our work in Nature Communications can reflect its fullest value for frequency and amplitude modulations. Finally, we sincerely hope that the reviewer can support our work.

1. Wang L.-Z. *et al.* A geometrical approach to control and controllability of nonlinear dynamical networks. *Nat. Commun.* **7**, 11323 (2016).
2. Murugan A. & Vaikuntanathan S. Topologically protected modes in non-equilibrium stochastic systems. *Nat. Commun.* **8**, 13881(2017).
3. Angulo M. T., Moog C. H. & Liu Y.-Y. A theoretical framework or controlling complex microbial communities. *Nat. Commun.* **10**, 1045 (2019).
4. Heltberg M. L., Krishna S. & Jensen M. H. On Chaotic dynamics in transcription factors and the associated effects in differential gene regulation. *Nat. Commun.* **10**, 71 (2019).
5. Gambuzza L. V. *et al.* Stability of synchronization in simplicial complexes. *Nat. Commun* **12**, 1255 (2021).
6. Glass D. S., J. X. & Riedel-Kruse I. H. Nonlinear delay differential equations and their application to modelling biological network motifs. *Nat. Commun.* **12**, 1788 (2021).
7. Zhang Y. & Strogatz S. H. Designing temporal networks that synchronize under resource constraints. *Nat. Commun.* **12**, 3273 (2021).

Reviewers' Comments:

Reviewer #1:

Remarks to the Author:

I would like to thank the authors for their commitment to addressing my comments. I found the new version of the manuscript remarkably improved both in terms of the strength of its message and the quality of the presentation. The only downside of the present version is the fact that it appears disseminated by typos and grammar issues, for which I am advising professional English writing support in the production stage. Nevertheless, I believe that the manuscript can now be promoted to publication. Congratulations.

Reviewer #2:

Remarks to the Author:

The authors have done a good job in addressing the points raised in my original review. I recommend the revised paper for publication in Nature Communications.

Reviewer #3:

Remarks to the Author:

I appreciate the changes and detailed responses by the authors. The paper has improved. Nevertheless, I think that a publication in a more specialized might be more suitable.

Reviewer #4:

Remarks to the Author:

The authors have substantially improved the presentation of the results, and the addition of the first Results section with the model details is very welcome.

However, my main points remain unaddressed and I do think the paper is better suited for a specialised maths/nonlinear dynamics journal, as also suggested by other referees.

[quote from my first review]

1) It is shown that the feedback controllers can indeed achieve the control, but there is no guarantee that any such controllers can be implemented by a real biochemical mechanism. There is plenty of recent literature on equivalences between PID controllers and biochemical implementations (particularly in Cell Systems journal), but the natural implementation of a general linear feedback controller as obtained in this work is in general not possible.

2) The authors show the utility of this approach in two model biological systems. But because of the impossibility of finding biological circuits that implement the controller in both case studies, the examples somewhat diminish the extent of the theoretical contribution.

I did not mean that authors need to provide experimental validation of the results (which is what the rebuttal says), and I am fully supportive of theoretical contributions in this type of journal. My point is about the lack of molecular circuits/systems that can produce the required control input $u(t)$. It would be great if the authors could provide simulations of circuits that can actually achieve the input $u(t)$ they derived. One example of this type of exercise is this paper <https://www.sciencedirect.com/science/article/pii/S2405471219303084> where authors discovered molecular motifs that behave as PID controllers. I think this would be a great contribution and would not need experimental validation to be influential in its own right.

As an alternative, the revision discusses the use of external control via a microfluidic device, and it claims that would be 'straightforward'. Unfortunately this statement is not accurate, and the many papers published in the past 3 years on microfluidic-based control of cell cultures will attest for the multiple challenges of the problem, see e.g. the works of Khammash, Hersen, Di Bernardo, and

other leading labs working in this space. The issue here is that the microfluidic actuators are quite limited, both in the range of outputs they can produce and the sampling time.

[quote from my first review]

3) The paper touches on energy considerations for each controller, but I found these results quite disconnected from the way energy consumption is defined in neural models and the protein-mRNA model.

This comment remains completely unaddressed; in neural dynamics particularly (like their first example), the problem of energy consumption and homeostasis has been addressed by countless works (a quick search for 'energy consumption neural dynamics' will reveal this). The use of L2 norm of the input as an energy metric is quite disconnected from the metrics that are standard in the neural dynamics literature. This would be completely fine for a maths/nonlinear dynamics journal, but for the interdisciplinary readership of Nature Comms, which certainly includes neural dynamics researchers, this assumption will be a red flag or at least deserved a thorough discussion and comparison with the neural dynamics literature.

As I said in my first review, and also suggested by other reviewers, this is a well executed theoretical work but too disconnected from the specific domains it touches upon (neural dynamics, and molecular circuits) and hence better suited for a technical journal.

Responses to the reviewers' comments and changes made in the revised manuscript

First of all, we greatly appreciate the supportive and constructive comments made by all the reviewers and apologize that there are still issues remain unaddressed. Especially, we sincerely thank the reviewer #4 for his/her significant suggestions. We provide below our new responses and corresponding changes made to these two remaining concerns made by the reviewer #4.

In summary, we have re-designed the controller in a more biological manner to coordinate the frequency and amplitude in the "cancer network" example. We now use the Michaelis-Menten regulations to successfully modulate the two quantities. We then re-performed all the numerical simulations using the new coordinator and explained why the Michaelis-Menten regulations are feasible to accomplish the tasks. Also, for the energy consumption in the neuronal model, we have searched related works in the literature and redefined it in a classical way. We think that these two major revisions would connect our work tightly to the biological discipline and would be more friendly to the audience in this direction. Please see below the detailed responses.

1. Coordinating the frequency and amplitude in the gene regulatory network using the molecular circuit or molecular function.

[quote from my first review]

- 1) It is shown that the feedback controllers can indeed achieve the control, but there is no guarantee that any such controllers can be implemented by a real biochemical mechanism. There is plenty of recent literature on equivalences between PID controllers and biochemical implementations (particularly in Cell Systems journal), but the natural implementation of a general linear feedback controller as obtained in this work is in general not possible.*
- 2) The authors show the utility of this approach in two model biological systems. But because of the impossibility of finding biological circuits that implement the controller in both case studies, the examples somewhat diminish the extent of the theoretical contribution.*

I did not mean that authors need to provide experimental validation of the results (which is what the rebuttal says), and I am fully supportive of theoretical contributions in this type of journal. My point is about the lack of molecular circuits/systems that can produce the required control input $u(t)$. It would be great if the authors could provide simulations of circuits that can actually achieve the input $u(t)$ they derived. One example of this type of exercise is this paper

<https://www.sciencedirect.com/science/article/pii/S2405471219303084>

where authors discovered molecular motifs that behave as PID controllers. I think this would be a great contribution and would not need experimental validation to be influential in its own right.

As an alternative, the revision discusses the use of external control via a microfluidic device, and it claims that would be 'straightforward'. Unfortunately this statement is not accurate, and the many papers published in the past 3 years on microfluidic-based control of cell cultures will attest for the multiple challenges of the problem, see e.g. the works of Khammash, Hersen, Di Bernardo, and other leading labs working in this space. The issue here is that the microfluidic actuators are quite limited, both in the range of outputs they can produce and the sampling time.

Response:

Thanks for the reviewer's comments and suggestions with which we totally agree. We also apologize for our misunderstanding on some comments provided by the reviewer #4 previously. When preparing the previous manuscript, we think that the linear terms designed in the controller is the simplest way to achieve our goal. Also,

from a mathematical viewpoint, considering the leading term is a natural way. That is why we only investigate the linear terms both analytically and numerically in the old version. We agree that this configuration is somehow disconnected from the specific molecular situations.

To address this issue, we re-designed the coordinator for the “cancer network” example. The new coordinator comprises the Michaelis-Menten regulations. Such regulations were shown to be useful and practical for realizing the proportional control in biological molecules in the work mentioned by the reviewer #4 [Michael Chevalier et al., Cell Systems, 9, 338-353]. In our case, we consider all four possible regulations: two-self regulations and two cross-regulations.

$$\begin{aligned} F_1(\phi, \mu) &= f_{11}M(\phi - \phi_0, K_{11}) + f_{12}M(\mu - \mu_0, K_{12}), \\ F_2(\phi, \mu) &= f_{21}M(\phi - \phi_0, K_{21}) + f_{22}M(\mu - \mu_0, K_{22}), \end{aligned} \quad (2)$$

where $M(x, K) = x/(x + K)$ is a MM regulation and f_{ij} indicate its intensities. The MM regulation was recently

We have reperformed all the numerical simulations for this example using the new coordinator. It is verified that the independent frequency and amplitude coordinations are successfully attained by the new coordinator. Also, for the oscillation relatively far from the Hopf bifurcation, the new coordinator is still feasible to accomplish the task (suppressing the amplitude from the cancer zone).

To explain the reason why the Michaelis-Menten regulations works, we consider the leading terms of them. Because we investigate the oscillation near the Hopf bifurcation, the leading terms are linearly dependent on the state variables as follows

$$M(u_j, K_{ij}) = \frac{u_j}{u_j + K_{ij}} = \frac{1}{K_{ij}}u_j - \frac{1}{K_{ij}^2}u_j^2 + \dots \approx \frac{1}{K_{ij}}u_j.$$

In fact, near the Hopf bifurcation, we take the advantage of the linear regime of the Michaelis-Menten function. In such a region, the new designed regulations are regarded as a kind of proportional (or linear) feedback control. Therefore, near the Hopf bifurcation, the Michaelis-Menten regulations act as interventions on the linear endogenous interactions. Consequently, it is possible to connect our previous results with the new coordinator. In such a way, we are able to explain the mechanisms and also provide a computational framework to estimating the regulating intensities. We have also verified that the effects of the Michaelis-Menten regulation and its linear approximation are almost identical.

In the theoretical part, we only investigate the linear term because it is sufficient to estimate accurately the coefficients in the nonlinear coordinator. Also, the investigation on the linear term can be applied straightforwardly in other cases. For instance, in a neuronal model, we can simply integrate the linear terms as an additional stimulus current or power supply. The linear results are also possible to be applied in other generalized biological functions possessing linear regime. For more details, please see the “changes made” below or the new manuscript. We hope that, with this new designed coordinator using biological functions, our work is now connected tightly to the molecular dynamics and it becomes more reasonable for the audience from biological disciplines.

Changes made:

Because we have designed a new coordinator for the “cancer network” example and it is significant to show that the frequency and amplitude coordinations can be attained using biological functions, we have changed the logic and structure of the article. In the “Results” section of the revised manuscript, we first show the “cancer network” model and how we use the Michaelis-Menten regulations to coordinate the frequency and amplitude by presenting all the associated numerical results. In this section, we have included one more reference, the work on PID control published in Cell Systems.

Results

Coordinating a “cancer network” by Michaelis-Menten regulations. As a first example, we consider the cyclic dynamics in a gene regulatory network involving a microRNA (miRNA) cluster and a protein module⁵⁹. It is called a “cancer network” because the miRNA behaves as an oncogene or tumor suppressor depending on the protein concentration. Its mathematical model abstracts the interaction among the transcription factors and a miRNA cluster⁵⁹ whose dimensionless diffusive model is given as (see Supplementary Section 8 for more details on the model)

$$\begin{aligned}\frac{\partial\phi}{\partial t} &= d_p \frac{\partial^2\phi}{\partial x^2} + \frac{1}{\epsilon} \left[\alpha' + \left(\frac{\kappa\phi^2}{\Gamma_1' + \phi^2 + \Gamma_2'\mu} \right) - \phi \right], \\ \frac{\partial\mu}{\partial t} &= d_m \frac{\partial^2\mu}{\partial x^2} + 1 + \phi - \mu,\end{aligned}\quad (1)$$

where $\phi = \phi(x, t)$ and $\mu = \mu(x, t)$ represent the dimensionless level of protein and miRNA, respectively. Here, we include two diffusive terms (second order differentiation with respect to the spatial variable x) into the equations to describe the unbiased molecular diffusion. With appropriate parameters (Supplementary Table 3), the model possesses a constant quiescent state (ϕ_0, μ_0) which undergoes the Hopf bifurcation at $\epsilon = \epsilon^*$ yielding rhythmic oscillation when $\epsilon < \epsilon^*$ (Fig. 2).

To achieve frequency and amplitude coordinations, we add two terms $F_1(\phi, \mu)$ and $F_2(\phi, \mu)$ into the first and second equation to regulate the dynamics of protein and miRNA, respectively. They act as instantaneous regulations on ϕ and μ as follows

$$\begin{aligned}F_1(\phi, \mu) &= f_{11}M(\phi - \phi_0, K_{11}) + f_{12}M(\mu - \mu_0, K_{12}), \\ F_2(\phi, \mu) &= f_{21}M(\phi - \phi_0, K_{21}) + f_{22}M(\mu - \mu_0, K_{22}),\end{aligned}\quad (2)$$

where $M(x, K) = x/(x + K)$ is a MM regulation and f_{ij} indicate its intensities. The MM regulation was recently

shown to be practical and useful in implementing the proportional control in biological molecules³⁶. Here, we take into account all possible regulations: two self-regulations ($\phi \rightarrow \phi$ and $\mu \rightarrow \mu$) and two cross-regulations ($\phi \rightarrow \mu$ and $\mu \rightarrow \phi$). We need to point out that the regulations in equation (2) incorporate the coordinate translations because the dynamics considered in our case oscillates around the quiescent state (ϕ_0, μ_0) and we desire to leverage the information on their displacement from the quiescent state. Actually, such a translated regulation can be divided into two typical MM functions

$$M(x - x_0, K) = \frac{x - x_0}{x - x_0 + K} = \frac{x}{x + \bar{K}} - \frac{x_0}{\bar{K}} \cdot \frac{\bar{K}}{x + \bar{K}}, \quad (3)$$

with $\bar{K} = K - x_0$.

For the given Michaelis constants ($K_{ij} = 5$ in our case), we select appropriate regulation intensities f_{ij} to coordinate independently the frequency or amplitude of the oscillation near the Hopf bifurcation ($\epsilon = 0.08 < \epsilon^*$). By varying the intensities at distinct time, we eventually accelerate the oscillation with a doubled frequency. Concurrently, both amplitudes of protein and miRNA concentrations are kept near constants (Fig. 2a-d). Moreover, with other well-designed intensities, the independent amplitude coordination is also successfully performed (Supplementary Fig. 3 and Movie 5).

Depending on the levels of protein module ϕ , the miRNA cluster is classified as oncogenes or tumor suppressors⁵⁹. When ϕ approximately lies between 2.75 and 3.75, the region is labeled as a cancer zone where hyper-proliferation occurs. Accordingly, the probability of oncogenesis is increased. We find that the oscillation stays away from the cancer zone (i.e., $\phi < 2.75$) if it is close to the quiescent state (Fig. 2a, b). But, when it keeps growing, its amplitude increases and the peak enters the cancer zone. For instance, the peak of the oscillation (when $\epsilon = 0.05$ without a coordinator) shown in Fig. 2e, f ($t < 50$) lies in the cancer zone. To prevent the oscillation from the entry of the cancer zone, it is possible to use our coordinator with appropriate intensities f_{ij} to suppress its amplitude. For this purpose, we design appropriate coordinators and apply them at different time to decrease the amplitude gradually ($50 < t < 300$ in Fig. 2e, f). Eventually, the protein concentration stays away from the cancer zone. Thus, the miRNA cluster is no longer classified as oncogenes. Evidently, our designed coordinator is feasible for coordinating both frequency and amplitude. Let us now introduce its theoretical background and a universal computational framework for determining the intensities.

[36] Chevalier, M. et al. Design and analysis of a proportional-integral-derivative controller with biological molecules. *Cell Syst.* 9, 338–353 (2019).

Correspondingly, we have redrawn the corresponding figures according to the new numerical results. We have integrated some examples into new Fig. 2. The first four of them are previously presented in Supplementary Information and the rest of them are previously presented in the old Fig. 5g, h.

FIG. 2. Independent frequency and amplitude coordinations in the “cancer network” using Michaelis-Menten regulations. **a** The time course of protein concentration in the “cancer network” [equation (1)] when $\epsilon = 0.08$. When $t < 50$, no coordinator is applied to the system. Starting from $t = 50$, equation (2) with $K_{ij} = 5$ is applied as a coordinator. At time labels $t = 50, 100, 150, 200$ and 250 , the intensities f_{ij} are varied. **b** The time course of protein concentration at $x^* = \pi/2$. **c** The time course of miRNA level corresponding to **a**. **d** The time course of miRNA level at $x^* = \pi/2$. **e** The time course of protein concentration in the “cancer network” when $\epsilon = 0.05$. The amplitude is gradually and independently suppressed (the color becomes lighter) by varying the coordinator at distinct time labels. The brown area is the cancer zone with high probability of oncogenesis classified by the level of protein (> 2.75). The amplitude suppression prevents the oscillation from the entry into the cancer zone. For the corresponding time course of the miRNA cluster, see Supplementary Fig. 2. **f** The time course at $x^* = \pi/2$. (For the parameters and intensities of the coordinator used in **a-f** see, respectively, Supplementary Table 3 and Supplementary Table 4.)

Because we presented the “cancer network” as the first example, we have swapped the order of Fig. 1a, b.

After presenting all the numerical results in the “cancer network”, we explained in the second section of “Results” why the Michaelis-Menten regulations works by introducing its linear approximation. We also connect the “cancer network” with the general biological model.

Endogenous linear interactions and their interventions. To investigate the dynamics near the Hopf bifurcation of every biological diffusive model akin to equation (1), we can always translate the quiescent state to the origin and write the equations into a generic form (see Methods for a detailed description):

$$\underbrace{\frac{\partial \mathbf{u}(x, t)}{\partial t}}_{\text{Time evolution}} = \underbrace{D(\epsilon) \frac{\partial^2 \mathbf{u}(x, t)}{\partial x^2}}_{\text{Spatial diffusion}} + \underbrace{\mathbf{A}(\epsilon) \mathbf{u}(x, t)}_{\text{Linear interaction}} + \underbrace{\mathbf{g}(\mathbf{u}(x, t), \epsilon)}_{\text{Nonlinear interaction}}. \quad (4)$$

Such a system possesses two self-interactions and two cross-interactions, all of which can be divided into two parts: the linear interactions $\mathbf{A}\mathbf{u}$ and the nonlinear ones \mathbf{g} . The matrix \mathbf{A} is associated with the Jacobian matrix and can be acquired in any computational model²¹. It represents the endogenous linear interactions, which consists of four coefficients a_{ij} ($i, j = 1, 2$) representing associated interaction (u_j to u_i). The sign of a_{ij} indicates the role played by the component j (activator or inhibitor), see Fig. 6a, b. Their magnitudes represent the intensities which are determined by system parameters, such as: degradation rate, synthesis rate, Michaelis constant, etc. These parameters are usually different from system to system (see Supplementary Section 9 for more details on the linear interactions).

In the third section of “Results”, we introduced the computational framework by considering the linear terms. We explained the reason why we only take the linear terms into account. Also, we showed some nonlinear biological functions possessing a linear regime, which may also be used to achieve frequency and amplitude coordinations near the Hopf bifurcation. Note that the computational framework for determining the linear coefficient is the same as that in the previous manuscript.

A universal design policy for the coordinator. To design a feasible coordinator, we need to know how the intensities of the MM regulations relate to the frequency and amplitude. Here, we apply the center manifold and the normal form theories to accomplish the task (Fig. 3). In the following computational framework, we consider the linear regime of the MM function where the feedback coordinator for equation (4) is written as

$$\mathbf{F}\mathbf{u} = \begin{bmatrix} f_{11} & f_{12} \\ f_{21} & f_{22} \end{bmatrix} = \begin{bmatrix} u_1 \\ u_2 \end{bmatrix}. \quad (6)$$

Note that the intensities f_{ij} in equation (6) and equation (2) differ by a given Michaelis constant K_{ij} . Considering solely the linear regime is sufficient to provide accurate estimation of the intensities in equation (2). Comparing with the nonlinear MM functions, the linear terms

For a given biological system, the endogenous linear interactions always exist in its computational model and have dominant impacts on the behavior of the oscillation near the quiescent state. Specifically, the intensities of these interactions determine the frequency and amplitude decisively. When translating the coordinated system, our proposed MM regulations [equation (2)] are also shifted to a regular MM function. For instance, $M(\phi - \phi_0, K_{11})$ is translated to $M(\Phi, K_{11})$ with $\Phi = \phi - \phi_0$ representing the displacement of the oscillation from the quiescent state. Near the Hopf bifurcation, the quantity Φ is relatively small. The MM function is therefore in its linear regime, that is,

$$M(\Phi, K) = \frac{\Phi}{\Phi + K} \approx \frac{1}{K} \Phi \quad \text{when } |\Phi| \ll 1. \quad (5)$$

Consequently, near the Hopf bifurcation, the introduced MM regulation is analogous to a linear proportional control making interventions on the original linear interactions, thereby our proposed coordinator is feasible for coordinating the frequency and amplitude.

significantly simplify the subsequent calculations. Another advantage of investigating the linear terms is that it is applicable for other cases where the MM regulations may not be used. For instance, when coordinating a neuronal model, it is possible to integrate the linear terms as additional stimulus current or power supply. The result for equation (6) is also suitable for other nonlinear functions possessing a linear regime, such as: the sinusoidal signal/current

$$\sin(au_1 + bu_2) \approx au_1 + bu_2 \quad \text{when } |u_1| \text{ and } |u_2| \ll 1, \quad (7)$$

and other generalized Michaelis-Menten functions

$$\frac{u_1}{u_1 + K_1 u_2 + K_2} \approx \frac{1}{K_2} u_1 \quad \text{when } |u_1| \text{ and } |u_2| \ll 1, \\ \frac{u_1}{u_1 + K_1 u_1^2 + K_2} \approx \frac{1}{K_2} u_1 \quad \text{when } |u_1| \ll 1. \quad (8)$$

We now present the computational framework which includes following steps.

In the last paragraph of the third section of “Results”, we again mentioned that the Michaelis-Menten regulations can be approximated by the linear coordinator. We also compared the results for both coordinators (Supplementary Fig. 4).

Following the above procedure, we design feasible coordinators for the independent frequency and amplitude coordinations in the “cancer network” (see Supplementary Section 8 for computational details). As mentioned before, equation (6) provides accurate estimation of equation (2). To verify this fact, we also compare the efficacy of the linear coordinator and that of the MM regula-

tions. Our proposed nonlinear coordinator is indeed well approximated by the linear coordinator even if the oscillation is relatively far from the quiescent state (Supplementary Fig. 4).

Supplementary Fig. 4:

Figure 4: Reduce the amplitude of the “bigger” oscillation in the “cancer network” using distinct coordinators. The orange curve is the same as that shown in the main text Fig. 2f representing the time course coordinated by the MM functions, while the blue one is coordinated using linear terms alone. Apparently, the two coordinators have almost the same effect on coordinating the amplitude.

Besides the above major revision, we have also made some minor corrections (both in the main text and in the Supplementary Information) and redrawn two supplementary figures so that the entire article is consistent in its new logic and structure. Please refer to the new manuscript for these minor corrections which are highlighted with blue text. The new supplementary figures are shown below.

Figure 2: a The time course of miRNA cluster for independent amplitude coordination of the “bigger” oscillation in the “cancer network”, which corresponds to the time course of protein concentration given in the main text Fig. 2e, f. b The time course at $x^* = \pi/2$.

Figure 3: Independent amplitude coordinations for the periodic oscillation (close to the quiescent state) in the “cancer network”. The first row shows the original oscillation without a coordinator. In the second row, the amplitude is suppressed ($r_A = 0.6$) by the MM regulations $[(f_{11}, f_{12}) = (-8.133, 6.513)]$. In the last row, the amplitude is magnified ($r_A = 1.2$) by the MM regulations $[(f_{11}, f_{12}) = (10.465, -18.764)]$.

2. Energy consumption in the neuronal model

[quote from my first review]

3) The paper touches on energy considerations for each controller, but I found these results quite disconnected from the way energy consumption is defined in neural models and the protein-mRNA model.

This comment remains completely unaddressed; in neural dynamics particularly (like their first example), the problem of energy consumption and homeostasis has been addressed by countless works (a quick search for 'energy consumption neural dynamics' will reveal this). The use of L2 norm of the input as an energy metric is quite disconnected from the metrics that are standard in the neural dynamics literature. This would be completely fine for a maths/nonlinear dynamics journal, but for the interdisciplinary readership of Nature Comms, which certainly includes neural dynamics researchers, this assumption will be a red flag or at least deserved a thorough discussion and comparison with the neural dynamics literature.

Response:

Again, we apologize that we have not addressed this issue in the previous manuscript. To address this concern, we have read the related works on the energy consumption in the neuronal cable model (a reaction-diffusion model) [Huiwen Ju et al., *Cable energy function of cortical axons, Scientific Reports, 6, 29686, (2016)*] and the power consumption of an external controller for controlling the FitzHugh-Nagumo neuron [Fan Li et al., *Simulating the electric activity of FitzHugh-Nagumo neuron by using Josephson junction model, Nonlinear Dynamics, 69, 2169-2179, (2012)*]. Accordingly, we have redefined the index for assessing the energy consumption for the neuronal model in a classical way. Now, the analysis on the energy consumption is performed based on the new definition. Since the definition is changed, some associated conclusions are slightly different from the previous ones. Therefore, we have also changed the corresponding text in the new manuscript.

Changes made:

In the "Methods" of revised manuscript, we have provided detailed descriptions of the index for assessing the energy consumption. For the neuronal model, we now cite two new related references and use a classical definition to achieve our goal.

The index for assessing energy consumption. For different biological models, distinct definitions may be introduced to assess the energy consumption. For the diffusive neuronal model considered here, we adopt the classical definition used in the cable model⁶⁴. Accordingly, the energy at time t^* induced by the stimulus current I in a diffusive neuronal model is defined as

$$H = \int_0^{t^*} \int_0^\pi IV(x, t) dx dt. \quad (15)$$

Since we consider the periodic oscillation in this work, in order to make a fair comparison, we take the average energy consumption into account. It is defined as

$$\bar{H} = \frac{1}{T} \int_0^T \int_0^\pi IV(x, t) dx dt, \quad (16)$$

where T is the period of the considered oscillation. To implement our designed coordinator, the linear terms $f_{11}(V - V_0) + f_{12}(W - W_0)$ (Supplementary Section 7) can be integrated as an additional stimulus current I_c . Our aim is to assess the energy consumption of the coordinator alone rather than the entire system. We therefore refer to an analogous power consumption index used in controlling the activity of the FN neuron⁶⁵. Consequently, we finally define the index for our purpose as follows

$$\mathcal{E} = |\bar{H}_c - \bar{H}_o| = \frac{1}{T} \left| \int_0^T \int_0^\pi [(I_c + I_o)V_c - I_oV_o] dx dt \right|, \quad (17)$$

where I_o is the original constant stimulus current, V_o and V_c are the membrane potential in the original and controlled system, respectively. This index evaluates the average energy consumption of the designed coordinator.

As for the other biological models such as the "cancer network" considered in this work, the energy can be evaluated by the average L_2 -norm of the designed coordinator over one period, which yields the following index

$$\mathcal{E} = \frac{1}{T} \int_0^T \|F\mathbf{u}\| dt, \quad (18)$$

where T is the period of oscillation and $\|\cdot\|$ is the L_2 -norm induced by the inner product [Supplementary equation (S4)].

We introduce L_2 -norm here because it is a generic definition for the assessment of a control policy⁷³ including the one used in a biological network with protein-protein interactions⁶³. Moreover, the biorhythm can be regarded as a cyclic signal for information processing whose energy is also conventionally defined as the integral of L_2 -norm⁷⁴. Therefore, it is appropriate for the "cancer network". Of course, there could be other definition for a different model. It is worth pointing out that the two definitions introduced here can be computed in practice once the signal of a biorhythm (e.g., action-potential or expression level) is measured.

[64] Ju, H., Hines, M. L. & Yu, Y. Cable energy function of cortical axons. *Sci. Rep.* **6**, 29686 (2016).

[65] Li, F., et. al. Simulating the electric activity of FitzHugh-Nagumo neuron by using Josephson junction model. *Nonlinear Dyn.* **69**, 2169-2179 (2012).

We have recomputed the energy consumption in the neuronal model using the new definition. Accordingly, we have changed the corresponding figures (Figs. 4e, 5d, 7h and 7i).

A zoom-in view of Fig. 4e is provided in Supplementary Fig. 11.

Figure 11: The average energy consumption of the optimal coordinator for independent frequency coordination. It is a zoom-in view of the blue curve shown in Fig. 4e in the main text.

Since the definition has been changed, the associated results and conclusions are different from the previous ones. We therefore changed some text (in the sections “The optimal energy consumption of a coordinator” and “Hybrid coordination and energy transfer”) accordingly. The Fig. 6c (the old Fig. 3c) where the energy consumptions are indicated has also been changed.

The optimal energy consumption of a coordinator. In practice, the consumption of a coordinating policy for sustaining an oscillation is of great significance. Therefore, an index \mathcal{E} is always introduced to measure the performance of a specific control policy⁶³, see Methods for the one used in our problem. Fig. 4e shows the energy consumption of independent frequency coordination in the F-N system with N.B.C. It is clear that there is a big difference between the two policies. Usually, the blue one is the optimal one that we may use because its consumption is lower. It is worth pointing out that, throughout this work, we only illustrate the coordinations attained by the optimal coordinator. A clearer energy consumption of the optimal frequency coordinator

for the F-N model is provided in Supplementary Fig. 11 from which we deduce that the coordinator consumes more energy for acquiring a decreased frequency (i.e., $\omega_0/\omega_c > 1$). In Fig. 5d, the energy consumption for the optimal independent amplitude coordination is also given. Apparently, the magnitudes of the index \mathcal{E} are exceedingly different for the two cases in the sense that the amplitude coordination consumes more energy than the frequency coordination. Moreover, increasing and decreasing the amplitude yield analogous energy consumptions. In Fig. 6c, the optimal energy consumptions for the four cases are indicated according to their magnitudes.

consumed by the hybrid coordinators (Fig. 7h, i). Along the hybrid coordination, more energy are required from sustaining higher frequency to higher amplitude, because an oscillation with higher amplitude consumes more energy than the one with higher frequency. If the hybrid coordination is performed in an opposite way, some energy can be released. This finding is in accordance with the results shown in Fig. 6c.

Fig. 6c:

Reviewers' Comments:

Reviewer #4:

Remarks to the Author:

I would like to congratulate the authors for the effort and dedication put in the revision. Both of my remaining comments have been satisfactorily addressed - I think the paper is suitable for publication now and I hope the edits will increase the impact and broaden the readership of the paper.

Responses to the reviewers' comments and changes made in the revised manuscript

Reviewer #4 (Remarks to the Author):

I would like to congratulate the authors for the effort and dedication put in the revision. Both of my remaining comments have been satisfactorily addressed - I think the paper is suitable for publication now and I hope the edits will increase the impact and broaden the readership of the paper.

Response:

We thank again to reviewer #4 for his/her constructive comments according to which our work has been greatly improved.